# Vision and Language Synergy for Rehearsal Free Continual Learning

**M. Anwar Ma'sum**[1][*]**, Mahardhika Pratama**[1]**, Savitha Ramasamy**[2]**, Lin Liu**[1]**,
Habibullah Habibullah**[1]**, and Ryszard Kowalczyk**[1]

[1]University of South Australia, Mawson Lakes, SA, 5095, Australia
[2]Institute for Infocomm Research, A*STAR & IPAL, CNRS@CREATE
masmy039@mymail.unisa.edu.au, dhika.pratama@unisa.edu.au, ramasamysa@i2r.a-star.edu.sg,
lin.liu@unisa.edu.au, habibullah.habibullah@unisa.edu.au, ryszard.kowalczyk@unisa.edu.au

## Abstract

The prompt-based approach has demonstrated its success for continual learning problems. However, it still suffers from catastrophic forgetting due to inter-task vector similarity and unfitted new components of previously learned tasks. On the other hand, the language-guided approach falls short of its full potential due to minimum utilized knowledge and participation in the prompt tuning process. To correct this problem, we propose a novel prompt-based structure and algorithm that incorporate 4 key concepts (1) language as input for prompt generation (2) task-wise generators (3) limiting matching descriptors search space via soft task-id prediction (4) generated prompt as auxiliary data. Our experimental analysis shows the superiority of our method to existing SOTAs in CIFAR100, ImageNet-R, and CUB datasets with significant margins i.e. up to 30% final average accuracy, 24% cumulative average accuracy, 8% final forgetting measure, and 7% cumulative forgetting measure. Our historical analysis confirms our method successfully maintains the stability-plasticity trade-off in every task. Our robustness analysis shows the proposed method consistently achieves high performances in various prompt lengths, layer depths, and number of generators per task compared to the SOTAs. We provide a comprehensive theoretical analysis, and complete numerical results in appendix sections. The method code is available in https://github.com/anwarmaxsum/LEAPGEN for further study.

## 1 Introduction

Continual learning (CL) attracts tremendous interest in the world of artificial intelligence (AI) for dynamic environments. CL methods address the catastrophic forgetting when handling a sequence of tasks De Lange et al., 2021; Wang et al., 2024b. CL methods are proven promising in various applications e.g. computer vision, NLP, graph, and Automation (Liu et al., 2023; Biesialska et al., 2020; Tian et al., 2024; Shaheen et al., 2022). However, most of the CL methods require a small portion of previous task exemplars for a rehearsal process to maintain the knowledge from previously learned tasks (Rebuffi et al., 2017; Wu et al., 2019; Boschini et al., 2022). This approach isn't practical since old data can be unavailable anymore due to privacy constraints or data openness policy. Besides, the rehearsal approach incurs additional memory and computational expenses. To this end, a prompt-based approach comes as a breakthrough solution. Leveraging a frozen pre-trained backbone, the prompt-based approach demands only small added learnable parameters (prompt) to adapt to a new task. This approach leads to efficient yet accurate solutions for catastrophic forgetting.

Despite its promising performance and efficiency, the prompt-based approach suffers from the following dilemmas: First, the task-specific prompt approach e.g. DualPrompt(Wang et al., 2022b), SPrompt(Wang et al., 2022a), HiDE-Prompt(Wang et al., 2024a) and CPrompt (Gao et al., 2024) rely on task identifier where a misidentified task-id leads to misclassification e.g. in the case where the trained keys of two or more tasks have a similar vector. Second, the growing component approach e.g. CODA-P(Smith et al., 2023b), EvoPrompt(Kurniawan et al., 2024), and ConvPrompt(Roy et al., 2024) on the other hand increase prompt components to adapt with a new task, instead of the task identifier. However, the (new) extra components are optimal for the current task, but not for previous tasks. In addition, the old task components/generators that are trained side-by-side with the current

---

[*]Corresponding author

task components/generators lead to forgetting. Third, the shared learnable parameter approach such as G-Prompt in DualPrompt or semantic embedding in ConvPrompt has the risk of forgetting since it is adjusted in all tasks. This condition is similar to the pool-based approach i.e. L2P(Wang et al., 2022c) where any prompt in the pool may be trained in every task, as it is chosen by the input-to-prompt similarity mechanism. On the other hand, a language-guided approach e.g. LGCL (Khan et al., 2023) and GMM (Cao et al., 2024) do not fully reach the optimum potential of language assistance in prompt tuning since they only use the name of class as additional information.

In retrospect to the aforementioned drawbacks, we propose a novel approach for prompt-based continual learning. We design a new prompt generation and its learning mechanism to handle catastrophic forgetting. The key principles of our approach are (1) we utilize language as input for prompt generation instead of a tasks-shared learnable vector. The input is selected from the catalog of language descriptors based on their similarity to the input. (2) We utilize task-wise generators instead of growing generators to generate a prompt component, thus the current task generators are frozen in the upcoming tasks training. (3) We propose a new soft task-id prediction to limit the search space of matching descriptors. Our mechanism is different from the task-id prediction in the previous methods. (4) We utilize the generated language-based prompt as the auxiliary data to be appended to the input embedding. To our knowledge, these four principles are new in the prompt-based CL method and have not yet been explored in the previous State-of-the-art(s) (SOTAs).

Our contributions are: (1) We propose a novel rehearsal-free prompt-based method for CL problem named Language as Prompt Generator (LEAPGen) consisting of four main components as aforementioned. (2). We design a new task-id predictor and joint loss function for our method. (3). Our rigorous experiment and analysis prove the superiority of our method compared to the existing SOTAs in general and historical performance in terms of accuracy and forgetting index. Our extended analysis proves the robustness of the proposed method in various settings i.e., the number of layers, prompt length, and number of generators. (4). We provide rigorous analysis and discussion of our method theoretically and numerically, and complete numerical results, please see appendices.

## 2 RELATED WORK

**(a) Continual Learning:** DualPrompt (Wang et al., 2022b) and CODA-P (Smith et al., 2023b), offer a breakthrough solution for Class Incremental Learning (CIL) by training tiny task-aware parameters called **prompts** where the feature extractor e.g. ViT that contains far bigger parameters remains frozen. The prompt-based approach is proven to be more effective than the rehearsal approach e.g. ICARL (Rebuffi et al., 2017), GD (Prabhu et al., 2020), XDER(Boschini et al., 2022) that saves exemplars from the previous tasks and replays them along with current task samples, the bias correction approach e.g. BiC (Wu et al., 2019) and LUCIR (Hou et al., 2019) that trains an additional task-wise bias layer to balance the model's stability-plasticity dilemma, and the regularization approach e.g. EWC (Kirkpatrick et al., 2017), MAS (Aljundi et al., 2018), and DMC (Zhang et al., 2020) that tunes the learner parameters to accommodate the previous and current tasks. However, the prompt-based approach still has above-mentioned dilemmas leading to forgetting.

**(b) Language Guided Learning:** Inspired from CLIP (Radford et al., 2021), the language-guided approach becomes a new alternative method to assist rehearsal and regularization approaches (Ni et al., 2024), contrastive learning (Zhu et al., 2023), few-shot continual learning (Park et al., 2024), generative approachh (Cao et al., 2024), and prompt-based methods (Khan et al., 2023; Roy et al., 2024). However, it is still far from its optimum potential since it only utilizes small knowledge from the learned class such as the name of the class or inter-class text similarity to generate several new prompts. The discriminative and representative knowledge of language modality is not yet utilized in the learning process. Please see Appendix F for the detailed literature review.

## 3 PRELIMINARY

**(a) Problem Formulation :** In this study, we focus on Class incremental learning (CIL) since it is the most challenging sub-problem in CL. Class incremental learning (CIL) problem is defined as the problem of learning a sequence of fully supervised tasks $\{\mathcal{T}^t\}_{t=1}^T$, where after finishing learning a task $\mathcal{T}^t$, a model must recognize all learned tasks i.e. $\mathcal{T}^1, \mathcal{T}^2, ... \mathcal{T}^{t-1}, \mathcal{T}^t$. Symbol $T$ represents the number of consecutive tasks. Each task carries pairs of training samples i.e. $\mathcal{T}^t = \{(x_i^t, y_i^t)\}_{i=1}^{|\mathcal{T}^t|}$ where $x_i \in \mathcal{X}^t$ and $y_i \in \mathcal{Y}^t$ denotes input image and corresponding label, while $|.|$ denotes cardinality. $\mathcal{C}^t$ denotes the unique class labels in $\mathcal{Y}^t$ i.e. $\mathcal{C}^t = unique(\mathcal{Y}^t)$, and $|\mathcal{C}^t|$ denotes the number of classes in $\mathcal{T}^t$. Each task $t$ is disjoint from another task $t'$ i.e $\forall t, t' \neq t, (\mathcal{T}^t \cap \mathcal{T}^{t'} = \emptyset)$.

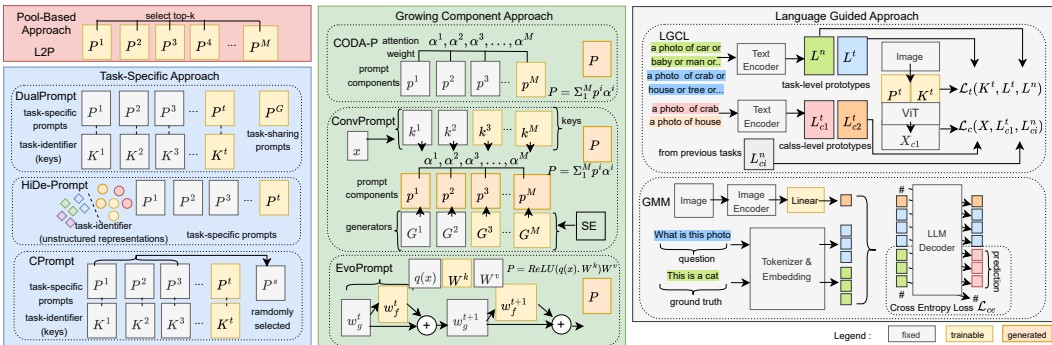

Figure 1: Visualization of Pool-Based Prompting (red area), Task-Specific Prompting (blue area), Growing Component Prompting (green area), and Language Guided Approach for Continual Learning (gray area).

**(b) Types of Prompt-based Approaches**: Figure 1 visualizes the topology of prompt-based approaches. **The pool-based approach** e.g. L2P(Wang et al., 2022c) selects top-k prompts from the prompt pool $[P^1, P^2, ...P^M]$ where $M$ is the pool size. Any prompt $P^m$ in the pool is trainable at any task $t \in [1..T]$. **The task-specific prompt approach** e.g. DualPrompt(Wang et al., 2022b) trains only the corresponding prompt $P^t$ during the training process on task $t$ while the prompt associated with the previous tasks i.e $[P^1, P^2, ...P^{(t-1)}]$ are frozen. Each task $t$ is accommodated with a learnable key $K^t$ to identify the task-id and select the prompt $P^t$ during the inference process. The prompt identification is based on the highest cosine similarity to the input. HiDe-Prompt(Wang et al., 2024a) utilize unstructured representation (centroids) produced by unsupervised learning instead of learnable keys. CPrompt(Gao et al., 2024) randomly selects a frozen prompt from $[P^1, P^2, ..., P^{t-1}]$ to support the tuning of $P^t$. **The growing component approach** dynamically increases the number of prompt components e.g. $p^i, p^{i+1}, ..p^M$ to adapt to a new task. The final prompt is generated by a weighted sum of the components with similarity coefficients e.g. $P = \Sigma_{i=1}^M p^i \alpha^i$. In CODA-P(Smith et al., 2023b), $p^i$ is a pre-defined learnable parameter, while $\alpha^i = \gamma(q(x), k^i)$ is attention similarity between input query $q(x)$ and learnable key $k^i$ associated to $p^i$. In ConvPrompt(Roy et al., 2024), $p^i$ is generated by respective generator network $G^i$ from shared embedding (SE). Similarly, $\alpha^i$ is the cosine similarity between input $x$ and $k^i$. EvoPrompt(Kurniawan et al., 2024) grows prompt memory $W$ that forms the fusion of referred (previously learned) memory $w_g^t$ and current working memory $w_f^t$. $W$ is implemented as a linear model with 2 parts i.e. $W^k$ and $W^v$, and the prompt $P$ is generated by the formula $P = ReLU(q(x).W^k)W^v$.

**(c) Language Guided Approaches**: The gray colored area of figure 1 shows how the role of language modality for the prompt-based method (LGCL) and generative method (GMM). LGCL(Khan et al., 2023) generates task level prototypes i.e. $L^1, L^2, ..L^t$ and class level prototypes e.g. $L_1^t, L_2^t, ..L_i^t$, then utilizes the prototypes for loss computation i.e. $\mathcal{L}_t(K^t, L^t, L^n)$ and $\mathcal{L}_c(X, L_c^T, L_{ci}^n)$ , where $\mathcal{L}_t$ and $\mathcal{L}_c$ are task-wise and class-wise losses respectively, $X$ and $L_c^t$ are the ViT output and language prototype for current learned class respectively, $K^t$ is the learnable key associated with prompt $P^t$, and $L^n$ and $L_c^n$ are task-level and class-level prototypes taken from the previously learned tasks. GMM(Cao et al., 2024) transforms question-and-answer (ground truth) sentences into embedding and then passes the embedding into the LLM decoder. The decoder generates predicted embedding associated with the ground truth embedding. The cross-entropy loss of predicted and ground truth tunes a trainable linear model.

**(d) Dilemma and Drawbacks**: The pool-based approach has a high risk of forgetting since all the prompts $[P^1, P^2, ...P^M]$ are possible to be trained in all tasks $t \in [1..T]$. A prompt $P^i$ is optimal for the $t^{th}$ task but is tuned again in the $t + 1^{th}$ task. Therefore, it is no longer optimal for $t^{th}$ task and leads to forgetting. In a task-specific approach, two different tasks e.g. $t \neq t'$ could produce similar key vectors i.e. $0.73 \leq cos(K^t, K^{t'}) \leq 0.96$ (Please see Appendix D.1) that leads to inaccurate task identification that leads to misclassification and forgetting. Note that $K^t$ is trained based on all samples of all classes in $\mathcal{T}^t$. $\mathcal{T}^t$ and $\mathcal{T}^{t'}$ are indeed disjoint so that $\mathcal{X}^t \cap \mathcal{X}^{t'} = \emptyset$, but they could have similar representation i.e. $f_\theta(\mathcal{X}^t) \approx f_\theta(\mathcal{X}^{t'})$ such as in CUB dataset where all the images are the photos of bird. Similarly, HiDe-Prompt suffers from the same drawback where two representations from different tasks have similar values $f_i \in \mathcal{T}^t \approx f_j \in \mathcal{T}^{t'}$. In the growing component approach, newly added components i.e. $p^i, p^{i+1}, ..p^M$ disrupts the previous components i.e. $p^1, p^2, ..p^{i+1}$

that already optimal for previous tasks. In addition, utilizing a continuously learned parameter e.g. shared embedding (SE)(Roy et al., 2024) or task-sharing prompt $P^G$(Wang et al., 2022b) increases the chance of forgetting. The existing language guidance is not fully explored. LGCL produces class prototype $L_c^n$ by encoding string "the photo of class name". However, the prototypes could be misleading due to high similarity between different classes, e.g. the prototype of class "Great White Shark" has 0.9 cosine similarity to the prototypes of class "Tree Frog" and "Iguana", please see Appendix D.2. With such different classes, we can't directly utilize language representations as references or guidance for model training.

**(e) Preliminary Analysis**: Table 1 presents the average accuracy and average forgetting of all learned classes in the first, second, and last tasks on ImageNet-R dataset with 10-tasks setting. The table shows that the existing prompt-based methods suffer from 9-15% average accuracy drop between the first and the final tasks. In the second task, the methods already suffer from up to 5% average forgetting and the amount tends to increase to up to 8% in the last tasks. Instead of reducing the average forgetting, almost all methods experience higher forgetting in latest tasks. HiDe-prompt manages to reduce the average forgetting with the

| Method | Avg. Accuracy | | | Avg. Forgetting | | |
|---|---|---|---|---|---|---|
| | T1 | T10 | Drop | T2 | T10 | Inc. |
| L2P | 76.89 | 62.50 | 14.39 | 5.52 | 5.01 | -0.51 |
| DualPrompt | 78.97 | 68.59 | 10.39 | 3.83 | 4.61 | 0.79 |
| CODA-P | 89.24 | 73.77 | 15.47 | 5.04 | 7.94 | 2.90 |
| LGCL | 78.59 | 68.65 | 9.93 | 2.91 | 4.75 | 1.85 |
| HiDe-Prompt | 85.22 | 75.75 | 9.47 | 3.29 | 2.29 | -1.00 |
| PGP | 78.97 | 68.62 | 10.35 | 3.83 | 4.53 | 0.70 |
| EvoPrompt | 89.27 | 76.00 | 13.27 | 5.17 | 4.22 | -0.95 |
| CPrompt | 90.91 | 76.32 | 14.59 | 4.55 | 6.10 | 1.54 |
| ConvPrompt | 89.53 | 77.08 | 12.45 | 2.57 | 4.17 | 1.61 |

Table 1: Preliminary results on Imagenet-R with 10 tasks setting.

highest amount i.e 1%, but in the last task, It achieves $1-2\%$ lower accuracy than the best achiever i.e. ConvPrompt. Despite its highest performance in the final tasks, ConvPrompt suffers from a fairly high accuracy drop i.e. 12%, and an increase of 1.5% forgetting. This preliminary numerical result confirms our aforementioned analysis.

## 4 PROPOSED METHOD

### 4.1 OVERVIEW

In this study, we propose a novel LanguagE As Prompt Generator (LEAPGen) accommodating our main principles that are emphasized in the introduction section. The structure and flow of LEAPGen are visualized in figure 2. LEAPGen generators produce prompts from top-k selected embedding as input. LEAPGen also produces auxiliary (aux) data from the top-k embedding. The prompts are prepended into ViT MSAs while the aux is appended into input patches, thus producing feature and final prediction by ViT layers and MLP head respectively. The top-k embedding is selected based on the cosine similarities between an input and the class-wise keys. LEAPGen limits the search space into task 1 to predicted task $t$, by performing soft task-id prediction. In each task of the training phase, LEAPGen updates task-associated learnable parameters i.e. generator, task-wise key, and class-wise keys. In the inference phase, LEAPGen selects the generators based on the predicted task $t$. LEAPGen utilizes cross-entropy loss and cosine similarity loss to optimize its parameters. Task-wise generators, language embedding for prompt generation, and soft task-id prediction are unique to recent SOTAs of evolving generator methods e.g. (Roy et al., 2024) and (Kurniawan et al., 2024), task-wise fixed prompt methods such as (Wang et al., 2024a) and (Gao et al., 2024), and pool-based method (Wang et al., 2022c) in terms of prompt generation/selection, task-prediction mechanism, and modality for prompt generation. The detailed architecture, flow, and learning mechanism are presented in sub-section 4.2 and 4.3.

### 4.2 ARCHITECTURE AND PROMPT GENERATION

**(a) Input and Learnable Parameters:** while the existing works use the language to compute loss as in LGCL(Khan et al., 2023) or calculate the number of generators per task as in ConvPrompt (Roy et al., 2024), we take the benefit of language modality as the input for prompt generation. Our method accommodates both language text i.e. generated descriptors by GPT as in ConvPrompt and class names (as utilized in LGCL) as descriptors. Descriptors list carries text describing the visual attributes of the classes, e.g. the descriptors for class "apple" are ["round shape", "smooth, glossy skin", "red, green, or yellow color", "stem and leaves", "five-pointed star shape when cut in half", "white flesh inside"]. The descriptors are more distinctive and representative than the words "the photo of apple" utilized in the existing works. The list of descriptors is then unified into a single long string and encoded into a numerical vector i.e. $E_c^t \in \mathbb{R}^D$ called a descriptor embedding. We store the descriptor embedding from the learned classes as the catalog for prompt generation. Each task $t$ contain $k$ generators $\{G_i^t\}_{i=1}^k$, a task-wise learnable key $K^t \in \mathbb{R}^D$, and a set of class-wise learnable parameters $\{L_c^t\}_{c=1}^{|C^t|}, L_c^t \in \mathbb{R}^D$, where $k$ is a predefined number, and $|C^t|$ is the number of classes on task $t$, each $L_c^t$ is associated to $E_c^t$. These learnable parameters are tuned only on task-$t$.

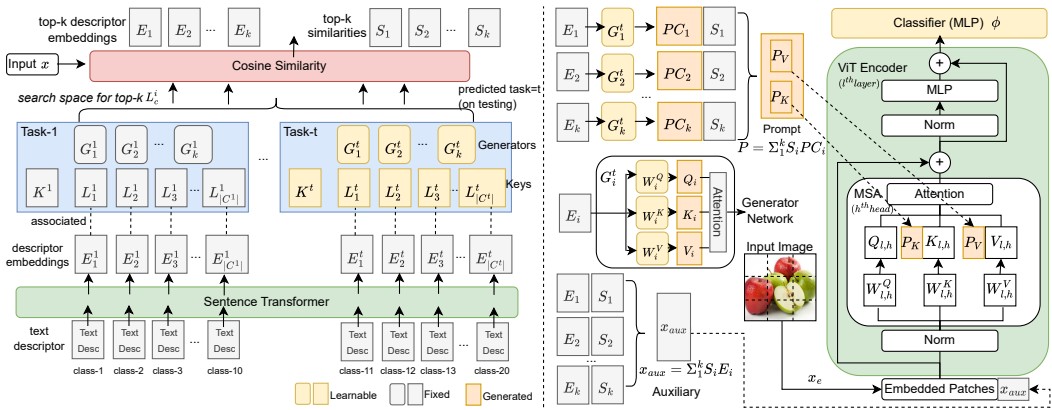

Figure 2: The LEAPGen architecture. Each task contains a set of generators, a task-wise learnable, and class-wise learnable parameters associated with the descriptor embedding. Given an input image, our method predicts the task-id and finds top-k descriptors matched to the input via cosine similarity afterward. The descriptors are dispatched to the generators producing the prompt components. LEAPGen produces the final prompt and an auxiliary embedding by the weighted sum of the prompt components and the descriptors respectively. The prompt is prepended into the ViT MSA layer, while the auxiliary is appended into the input patches.

**(b) Prompt Generation:** Before generating the prompt for an input image $x$, our method predicts the task-id via a soft task prediction (ref. point c). In the training process, the task-id is easily given in a supervised way instead of being predicted. Let the task-id be $t$, then our method selects $k$ descriptor embedding associated with top-$k$ matched $L$ by performing cosine similarity. Note that the search space is from the first task (task-1) until task-$t$ i.e. $\{E_c^1\}_{c=1}^{|C^1|}, \{E_c^2\}_{c=1}^{|C^2|}, ..., \{E_c^t\}_{c=1}^{|C^t|}$. Let the $k$ selected embedding be $\{E_i\}_{i=1}^k$ and associated similarity values $\{S_i\}_{i=1}^k$, then the embedding is inputted into the respective generators i.e. $\{G_i^t\}_{i=1}^k$ producing prompt components $\{PC_i\}_{i=1}^k$. Note that each generator is a trainable network, so it satisfies $PC_i = G_i^t(E_i)$. The final prompt is generated by a weighted sum of the prompt components and similarity i.e. $P = \Sigma_{i=1}^k S_i PC_i$. In a simplified expression, the prompt generation is formulated by equation 1.

$$P = \Sigma_{i=1}^k S_i G_i^t(E_i) \tag{1}$$

**(c) Generator Networks and Detailed Generation:** we design a single-head self-attention (SSA) network $G_i^t$ parameterized by $W_i^Q$, $W_i^K$ and $W_i^V$ without bias as visualized in figure 2. Following (Roy et al., 2024) and prefix tuning (Li & Liang, 2021), we implement $G_i^t$ as a key-value pair generators i.e. $(G_i^{tK}, G_i^{tV})$ and distribute them into corresponding layers and heads. Therefore, a $h^{th}$ head of MSA in $l^{th}$ layer corresponds to a pair of generators $(G_{i,l,h}^{tK}, G_{i,l,h}^{tV})$ parameterized by $(WK_{i,l,h}^Q, WKV_{i,l,h}^K, WK_{i,l,h}^V)$ and $(WV_{i,l,h}^Q, WKV_{i,l,h}^K, WV_{i,l,h}^V)$, where $WK_{i,l,h}^Q$ is $(D/H) \times (D/H)$ matrix and so do the other parameters. The language embedding $E_i$ is then divided into $H$ parts i.e. $E_i = [E_{i,l,1}, E_{i,l,2}, ..E_{i,l,H}]$, where a part $E_{i,l,h}, h \in [1..H]$ plays as the input for $(G_{i,l,h}^{tK}, G_{i,l,h}^{tV})$. Thus, the detailed formula of prompt generation for respective head $h$ on $l^{th}$ layer is expressed as in equation 2.

$$P_{l,h}^K = \Sigma_{i=1}^k S_i G_{i,l,h}^{tK}(E_{i,l,h}), \quad P_{l,h}^V = \Sigma_{i=1}^k S_i G_{i,l,h}^{tV}(E_{i,l,h}) \tag{2}$$

**(d) Soft Task-ID Prediction:** we design a soft task-id prediction to limit the search space to find top-k $E_i$. We use the term "soft" as the search of top-k $E_i$ is conducted not only within predicted task $t$ but also the previously learned tasks i.e. $[1, 2, ..t-1]$. The reason is that most similar language embedding comes from the respective class from within respective tasks, the second-ranked and the rest may come from classes from different (previous) tasks. Therefore, they will contribute to a better distinctive prompt than the embedding from the other classes within the same task. For instance, class "pear" has higher similar descriptors to class "apple" than to class "palm tree". But class "apple" is located in a different (previous) task, while class "palm tree" is within the same task as class "pear", thus selecting "apple" descriptors from the precious task is better than choosing "palm tree" descriptors from the same task. Different from the task-id prediction in previous works (Wang et al., 2022b; 2024a) we utilize both task-wise learnable parameter $K^t$ and class-wise learnable parameter $L_c^t$. We predict the task-id $t$ for an input $x$ by finding the maximum product of cosine

similarity between $x$ and $K^t$ and similarity between $x$ and $L_c^t$ as formulated in equation 3.

$$t = \underset{t\in[1..T],c\in C^t}{\arg\max} \ \{(x.K^t)(x.L_c^t)/((max(||x||_2.||K^t||_2,\epsilon))(max(||x||_2.||K^t||_2,\epsilon))\} \tag{3}$$

**(d) Learning with Auxiliary Data:** Our method generates auxiliary embedding based on $k$ selected language embedding used for prompt generation. The auxiliary embedding $x_{aux} \in \mathbb{R}^D$ helps our method to recognize the input image $x$ as it is generated based on the similarity between the input and the catalog of language embedding. The previous study(Shen et al., 2023) presents insightful knowledge of leveraging the auxiliary modality for deep learning training. The auxiliary embedding $x_{aux}$ is generated by the weighted sum of language embedding i.e. $x_{aux} = \Sigma_{i=1}^k s_i E_i$. The auxiliary embedding is concatenated into the input patches list, therefore the patches list that enters the ViT layers now becomes $[x_{e_1}, x_{e_2}, ..., x_{e_{N_p}}, x_{aux}]$, where $N_p$ is the number of patches.

### 4.3 FORMULATION OF LEARNING MECHANISM

**(a). Theoretical Foundation:** The goal of CIL is to learn a sequence of tasks i.e. $\mathcal{T}^1, \mathcal{T}^2, ..., \mathcal{T}^T$, where each task $\mathcal{T}^t, t \in [1..T]$ carries a pair of image and label set $(\mathcal{X}^t, \mathcal{Y}^t)$. Suppose that $C^t$ is the set of available classes in $\mathcal{Y}^t$, and $c \in C^t$ represents a class label $c$ available in task $t$. Suppose that ":" denotes the concatenation and $\mathcal{T}^{1:t} = \cup_{i=1}^t \mathcal{T}^i$ denotes the concatenation of task 1 to task $t$. Similarly, $\mathcal{X}^{1:t}$ and $\mathcal{Y}^{1:t}$ denotes the concatenation of input and label space respectively, from task 1 to task $t$. Given a sequence of experienced events $\mathcal{T} = \{\mathcal{T}^1, \mathcal{T}^2, ..., \mathcal{T}^T\}$, and a deep learning model $f_\theta(.)$. The objective of CIL is to recognize any input $x \in X^{1:T}$ by maximizing $P(x \in \mathcal{X}_c^t|\mathcal{T}, \theta)$. Following (Wang et al., 2024a), $P(x \in \mathcal{X}_c^t|\mathcal{T}, \theta)$ can be decomposed into task prediction and class prediction within task as in equation 4.

$$P(x \in \mathcal{X}_c^t|\mathcal{T}, \theta) = P(x \in \mathcal{X}_c^t|x \in \mathcal{X}^t, \mathcal{T}, \theta)P(x \in \mathcal{X}^t|\mathcal{T}, \theta) \tag{4}$$

Analogically, we can decompose the $P(x \in \mathcal{X}_c^t|\mathcal{T}, \theta)$ in a softer task-id prediction i.e. by decomposing the term $P(x \in \mathcal{X}_c^t|\mathcal{T}, \theta)$ w.r.t. $P(x \in \mathcal{X}^{1:t}|\mathcal{T}, \theta)$. Then, the decomposition can be derived into equation 5.

$$\begin{aligned} P(x \in \mathcal{X}_c^t|\mathcal{T}, \theta) &= P(x \in \mathcal{X}_c^t|x \in \mathcal{X}^t, \mathcal{T}, \theta)P(x \in \mathcal{X}^t|\mathcal{T}, \theta) \\ &= P(x \in \mathcal{X}_c^t|x \in \mathcal{X}^{1:t}, \mathcal{T}, \theta)P(x \in \mathcal{X}^{1:t}|\mathcal{T}, \theta) \end{aligned} \tag{5}$$

The first line of equation 5 expresses a hard decomposition where a classification is conducted into a strict two-step classification i.e. predict the task ID first, then predict the class label within the task. On the other hand, the second line of the equation 5 expresses a soft way of decomposed classification i.e. predict the possible task (1:t) of the sample then predict the class within the task range. In the first approach, a false task prediction directly leads to misclassification, while in the second approach, It still hold a possibility for correct classification as long as the predicted task ID is higher than the ground truth task. Since event $x \in \mathcal{X}^t$ is subset of event $x \in \mathcal{X}^{1:t}$, Then $P(x \in \mathcal{X}^t|\mathcal{T}, \theta) \le P(x \in \mathcal{X}^{1:t}|\mathcal{T}, \theta)$, that implies $P(x \in \mathcal{X}_x^t|x \in \mathcal{X}^t) \ge P(x \in \mathcal{X}_x^t|x \in \mathcal{X}^{1:t})$. The second approach has an easier way to predict the possible task but a harder step afterward, while the first approach has a more difficult task prediction but an easier way afterward. However, the equation above gives us insight that improving $P(x \in \mathcal{X}_c^t|\mathcal{T}, \theta)$ can be conducted by improving $P(x \in \mathcal{X}_c^t|x \in \mathcal{X}^t, \mathcal{T}, \theta), P(x \in \mathcal{X}^t|\mathcal{T}, \theta)$, $P(x \in \mathcal{X}_c^t|x \in \mathcal{X}^{1:t}, \mathcal{T}, \theta)$, $P(x \in \mathcal{X}^{1:t}|\mathcal{T}, \theta)$, pair to the first line, pair of the second line of equation 5, or all of them.

**(b). Learning Objective:** Based on the insights above, we develop a strategy to improve $P(x \in \mathcal{X}_c^t|\mathcal{T}, \theta)$ during the learning process. Note that $\psi$ is the frozen ViT parameter, $\theta = \{G_i^t, K^t, L_c^t\}_{t=1,c\in C^t}^{t=T}$ are our customized trainable parameters including generators, task-wise keys, and language class-wise keys, and $\phi$ is the parameter of MLP head (classifier) and the class label of an input $x$ is predicted by computing $h_\phi(f_{\psi,\theta}(x))$. First, utilizing cross-entropy function, we improve $P(x \in \mathcal{X}_c^t|x \in \mathcal{X}^t, \mathcal{T}, \theta)$ by intra-task loss $\mathcal{L}_{intra}$ as formulated in equation 6.

$$\mathcal{L}_{intra}(x, (\psi, \theta, \phi)) = -\sum_{c\in C^t} log \frac{exp(h_\phi(f_{\psi,\theta}(x))[c])}{\sum_{c'\in C^t} exp(h_\phi(f_{\psi,\theta}(x))[c'])} \tag{6}$$

where "[]" denotes the index filter for the respective class. Note that $h_\phi(f_{\psi,\theta}(x))$ produces a list of softmax values for all classes. Second, similar to the intra task loss, utilizing cross entropy function, we improve $P(x \in \mathcal{X}_c^t|x \in \mathcal{X}^{1:t}, \mathcal{T}, \theta)$ by inter-task loss $\mathcal{L}_{inter}$ as defined in equation 7.

$$\mathcal{L}_{inter}(x, (\psi, \theta, \phi)) = -\sum_{t=1}^{t=T}\sum_{c\in C^t} log \frac{exp(h_\phi(f_{\psi,\theta}(x))[c])}{\sum_{t=1}^{t=T}\sum_{c'\in C^t} exp(h_\phi(f_{\psi,\theta}(x))[c'])} \tag{7}$$

Then, we improve task prediction $P(x \in \mathcal{X}^t | \mathcal{T}, \theta)$ that leads to improve $P(x \in \mathcal{X}^{1:t} | \mathcal{T}, \theta)$ by computing negative cosine similarity between the input $x$ and task-wise and class wise learnable parameters $K^t$ and $L_c^t$ respectively as formulated in equations 8 and 9. Note that our designed task-id predictor utilize both $K^t$ and $L_c^t$ as explained in sub-section 4.1.c.

$$\mathcal{L}_t(x, K^t) = -(x.K^t)/(max(||x||_2.||K^t||_2, \epsilon)) \tag{8}$$

$$\mathcal{L}_c(x, L_c^t) = -(x.L_c^t)(max(||x||_2.||L_c^t||_2, \epsilon)) \tag{9}$$

Finally, we define a joint loss function as a total loss accommodating the four loss components above to train the learnable parameters, generators, and classifier head formulated in the equation 10. We add loss coefficients i.e. $\lambda_1, \lambda_2, \lambda_3$ to enhance the flexibility of our proposed method e.g. the higher number of learned classes needs higher capability on inter-task discrimination. We don't utilize any regularization, since all learned parameters on task-$t$ will be frozen after finishing the task.

$$\mathcal{L}_{total} = \mathcal{L}_{intra}(x, (\psi, \theta, \phi)) + \lambda_1 \mathcal{L}_{inter}(x, (\psi, \theta, \phi)) + \lambda_2 \mathcal{L}_t(x, K^t) + \lambda_3 \mathcal{L}_c(x, L_c^t) \tag{10}$$

**(c). Training and Inference** procedures for LEAPGen are presented in Appendix B.

### 4.4 THEORETICAL ANALYSIS

We state Theorem 1, 2, and 3 that prove the importance of the generator network, loss minimization, and better task prediction respectively as presented in Appendix A. Please see Appendix A for our detailed theorems and theoretical analysis.

## 5 EXPERIMENT AND ANALYSIS

### 5.1 EXPERIMENT SETTING

**Datasets:** We evaluate our method in three CIL benchmarks i.e. CIFAR100(Hendrycks et al., 2021), ImageNet-R(Belouadah & Popescu, 2019), and CUB(Wah et al., 2011). The CIFAR100 has 100 classes of small-scale images while ImageNet-R contains large-scale images that cover 200 classes. ImageNet-R contains art, cartoon, paint, and deviant images. The CUB dataset includes fine-grained images of 200 classes of birds. We follow 5, 10, and 20 task splits as in the SOTAs (Roy et al., 2024; Gao et al., 2024; Kurniawan et al., 2024).

**Baselines and Performance Metrics:** We compare our proposed method with the existing SO-TAs of prompt-based approach i.e. CIFAR100 L2P(Wang et al., 2022c), DualPrompt(Wang et al., 2022b), CODA-P(Smith et al., 2023b), LGCL(Khan et al., 2023), PGP(Qiao et al., 2023), HiDE-Prompt(Wang et al., 2024a), EvoPrompt(Kurniawan et al., 2024), CPrompt(Gao et al., 2024), and ConvPrompt(Roy et al., 2024). We also compare LEAPGen to low-Rank Adaptation (LoRA) approach i.e. CLoRA(Smith et al., 2023a), LAE(Gao et al., 2023), InfLoRA(Liang & Li, 2024), slow learner approach i.e. SLCA(Zhang et al., 2023), and generative approach i.e. GMM(Cao et al., 2024). Adapted from Hide-Prompt(Wang et al., 2024a), we measure final average accuracy (FAA) and final forgetting measure (FFM) that represent the final performance after learning all tasks, and cumulative average accuracy (CAA) and cumulative forgetting measure (CFM) that represent cumulative historical performance on each task. See Appendix E.

**Implementation Details:** We implemented LEAPGen and its lite version i.e. LEAPGen-lite in Pytorch utilizing the pre-trained ViT-B/16 backbone following the common practice. LEAPGen-lite utilizes lightweight Conv1d generators (note: different from ConvPrompt) without descriptors, it uses class names instead. We utilize Adam optimizer, set 128 batch-size, and cosine learning scheduler similar to ConvPrompt and CODA-P. The initial learning rate is set to 0.01, 0.05, and 0.005 for CIFAR100, ImageNet-R, and, CUB respectively, with 20 maximum epochs. All the methods are run under the same environment i.e. a single NVIDIA A100 GPU with 40 GB RAM. The hyper-parameter settings follow their official settings. See Appendix E.

### 5.2 RESULT AND ANALYSIS

**a) Overall Performance:** Table 2 shows the summarized numerical results of consolidated algorithms in split CIFAR100 and ImageNet-R dataset, while table 3 shows the summarized result in split CUB dataset. The detailed numerical results are presented in Appendix G. In the split CIFAR100 dataset, our proposed method (LEAPGen) achieves the highest accuracy with a 5-16% margin of FAA and a 5-11% margin of CAA. The table shows that a higher number of tasks results in a higher gap in performance. For the forgetting measure, our method consistently achieves the lowest FFM

| Method | Split CIFAR100 | | | | Split ImageNet-R | | | |
|---|---|---|---|---|---|---|---|---|
| | FAA | CAA | FFM | CFM | FAA | CAA | FFM | CFM |
| | 5 Task @20 classes/task | | | | 5 Task @40 classes/task | | | |
| L2P | $84.77 \pm 0.48$ | $88.67 \pm 0.30$ | $6.18 \pm 0.57$ | $5.99 \pm 0.29$ | $64.62 \pm 0.32$ | $68.01 \pm 0.42$ | $3.94 \pm 0.16$ | $3.55 \pm 0.20$ |
| DualPrompt | $86.41 \pm 0.21$ | $89.95 \pm 0.10$ | $5.37 \pm 0.21$ | $4.77 \pm 0.46$ | $69.71 \pm 0.11$ | $72.78 \pm 0.14$ | $3.32 \pm 0.16$ | $2.78 \pm 0.25$ |
| CODA-P | $88.22 \pm 1.06$ | $92.25 \pm 1.28$ | $7.05 \pm 2.18$ | $6.06 \pm 2.66$ | $74.89 \pm 0.36$ | $79.71 \pm 1.27$ | $8.89 \pm 0.65$ | $7.65 \pm 0.98$ |
| LGCL | $86.90 \pm 0.40$ | $90.45 \pm 0.18$ | $5.01 \pm 0.35$ | $4.36 \pm 0.13$ | $69.93 \pm 0.21$ | $72.91 \pm 0.19$ | $3.04 \pm 0.36$ | $2.50 \pm 0.38$ |
| HiDe-Prompt | $91.99 \pm 0.03$ | $93.95 \pm 0.09$ | $2.52 \pm 0.18$ | $2.33 \pm 0.15$ | $75.40 \pm 0.27$ | $78.88 \pm 0.04$ | $3.15 \pm 0.46$ | $2.64 \pm 0.16$ |
| PGP | $87.69 \pm 0.06$ | $91.26 \pm 0.13$ | $5.32 \pm 0.18$ | $4.60 \pm 0.15$ | $69.71 \pm 0.15$ | $72.77 \pm 0.07$ | $3.36 \pm 0.23$ | $2.85 \pm 0.25$ |
| EvoPrompt | $89.07 \pm 0.38$ | $92.32 \pm 0.26$ | $5.25 \pm 0.65$ | $5.39 \pm 0.24$ | $77.27 \pm 0.40$ | $81.67 \pm 0.18$ | $1.79 \pm 0.31$ | $1.41 \pm 0.32$ |
| CPrompt | $89.22 \pm 0.05$ | $93.09 \pm 0.06$ | $5.02 \pm 0.17$ | $4.31 \pm 0.35$ | $78.65 \pm 0.00^*$ | $82.44 \pm 0.00$ | $6.00 \pm 0.00$ | $5.49 \pm 0.00$ |
| ConvPrompt | $90.26 \pm 0.44$ | $93.49 \pm 0.19$ | $3.64 \pm 0.28$ | $3.25 \pm 0.16$ | $79.36 \pm 0.08$ | $82.93 \pm 0.24$ | $3.42 \pm 0.05$ | $2.36 \pm 0.16$ |
| **LEAPGen-lite** | $\mathbf{97.07 \pm 0.08}$ | $\mathbf{97.28 \pm 0.10}$ | $\mathbf{0.05 \pm 0.01}$ | $\mathbf{0.02 \pm 0.01}$ | $\mathbf{82.44 \pm 0.63}$ | $\mathbf{84.37 \pm 0.90}$ | $\mathbf{0.43 \pm 0.08}$ | $\mathbf{0.17 \pm 0.06}$ |
| **LEAPGen** | $\mathbf{96.84 \pm 0.12}$ | $\mathbf{96.85 \pm 0.26}$ | $\mathbf{0.08 \pm 0.07}$ | $\mathbf{0.06 \pm 0.04}$ | $\mathbf{82.79 \pm 0.32}$ | $\mathbf{85.06 \pm 0.29}$ | $\mathbf{0.51 \pm 0.04}$ | $\mathbf{0.18 \pm 0.07}$ |
| | 10 Task @10 classes/task | | | | 10 Task @20 classes/task | | | |
| L2P | $83.84 \pm 0.32$ | $88.67 \pm 0.16$ | $6.55 \pm 0.34$ | $5.16 \pm 0.14$ | $62.50 \pm 0.51$ | $67.05 \pm 0.47$ | $5.01 \pm 0.40$ | $4.41 \pm 0.43$ |
| DualPrompt | $85.36 \pm 0.20$ | $89.77 \pm 0.20$ | $5.41 \pm 0.33$ | $4.33 \pm 0.15$ | $68.59 \pm 0.24$ | $72.18 \pm 0.20$ | $4.61 \pm 0.07$ | $3.70 \pm 0.18$ |
| CODA-P | $86.44 \pm 0.16$ | $91.27 \pm 0.56$ | $6.38 \pm 1.46$ | $5.09 \pm 1.19$ | $73.77 \pm 0.50$ | $79.38 \pm 1.48$ | $7.94 \pm 0.08$ | $6.72 \pm 0.79$ |
| LGCL | $85.68 \pm 0.43$ | $90.16 \pm 0.29$ | $5.46 \pm 0.22$ | $4.25 \pm 0.32$ | $68.65 \pm 0.25$ | $72.57 \pm 0.19$ | $4.75 \pm 0.33$ | $3.38 \pm 0.58$ |
| HiDe-Prompt | $92.89 \pm 0.11$ | $95.01 \pm 0.08$ | $1.98 \pm 0.05$ | $1.56 \pm 0.15$ | $75.75 \pm 0.40$ | $79.27 \pm 0.17$ | $2.29 \pm 0.27$ | $2.33 \pm 0.17$ |
| PGP | $86.36 \pm 0.19$ | $90.83 \pm 0.17$ | $5.49 \pm 0.35$ | $4.28 \pm 0.27$ | $68.62 \pm 0.14$ | $72.19 \pm 0.20$ | $4.53 \pm 0.40$ | $3.63 \pm 0.35$ |
| EvoPrompt | $88.17 \pm 0.51$ | $92.18 \pm 0.49$ | $5.39 \pm 0.45$ | $3.97 \pm 0.73$ | $76.00 \pm 0.26$ | $80.97 \pm 0.30$ | $4.22 \pm 0.42$ | $3.59 \pm 0.52$ |
| CPrompt | $86.92 \pm 1.04$ | $91.73 \pm 0.66$ | $5.43 \pm 0.74$ | $4.01 \pm 0.81$ | $76.32 \pm 0.53$ | $81.50 \pm 0.30$ | $6.10 \pm 0.75$ | $5.60 \pm 1.35$ |
| ConvPrompt | $88.77 \pm 0.24$ | $92.71 \pm 0.04$ | $4.12 \pm 0.44$ | $2.67 \pm 0.11$ | $77.08 \pm 0.26$ | $81.47 \pm 0.10$ | $4.17 \pm 0.04$ | $3.11 \pm 0.17$ |
| **LEAPGen-lite** | $\mathbf{98.58 \pm 0.03}$ | $\mathbf{98.69 \pm 0.10}$ | $\mathbf{0.11 \pm 0.03}$ | $\mathbf{0.06 \pm 0.03}$ | $\mathbf{82.38 \pm 1.04}$ | $\mathbf{85.14 \pm 0.52}$ | $\mathbf{3.01 \pm 1.19}$ | $\mathbf{2.13 \pm 0.60}$ |
| **LEAPGen** | $\mathbf{98.38 \pm 0.15}$ | $\mathbf{98.15 \pm 0.39}$ | $\mathbf{0.10 \pm 0.03}$ | $\mathbf{0.05 \pm 0.00}$ | $\mathbf{84.09 \pm 0.93}$ | $\mathbf{85.54 \pm 0.65}$ | $\mathbf{1.46 \pm 1.25}$ | $\mathbf{2.11 \pm 1.21}$ |
| | 20 Task @5 classes/task | | | | 20 Task @10 classes/task | | | |
| L2P | $81.89 \pm 0.38$ | $87.16 \pm 0.33$ | $8.81 \pm 0.10$ | $6.79 \pm 0.33$ | $57.40 \pm 0.31$ | $63.33 \pm 0.21$ | $10.76 \pm 0.45$ | $7.88 \pm 0.17$ |
| DualPrompt | $82.32 \pm 0.22$ | $87.47 \pm 0.24$ | $6.88 \pm 0.35$ | $5.63 \pm 0.23$ | $65.19 \pm 0.17$ | $70.31 \pm 0.29$ | $7.30 \pm 0.18$ | $5.16 \pm 0.34$ |
| CODA-P | $81.29 \pm 0.16$ | $87.72 \pm 0.44$ | $6.82 \pm 1.60$ | $4.98 \pm 0.95$ | $70.55 \pm 0.71$ | $77.08 \pm 1.02$ | $8.23 \pm 0.86$ | $6.95 \pm 0.70$ |
| LGCL | $83.18 \pm 0.40$ | $88.63 \pm 0.18$ | $7.22 \pm 0.47$ | $4.91 \pm 0.56$ | $64.96 \pm 0.67$ | $70.18 \pm 0.37$ | $7.35 \pm 0.65$ | $5.05 \pm 0.32$ |
| HiDe-Prompt○ | - | $97.62 \pm 0.14$ | - | $0.74 \pm 0.03$ | - | $81.60 \pm 0.48$ | - | $2.23 \pm 0.38$ |
| PGP | $83.41 \pm 0.35$ | $89.23 \pm 0.13$ | $7.95 \pm 0.23$ | $5.66 \pm 0.22$ | $65.24 \pm 0.25$ | $70.36 \pm 0.26$ | $7.17 \pm 0.21$ | $5.09 \pm 0.25$ |
| EvoPrompt | $84.63 \pm 0.21$ | $89.47 \pm 0.21$ | $9.19 \pm 0.41$ | $7.39 \pm 0.65$ | $74.93 \pm 0.64$ | $79.92 \pm 0.13$ | $6.72 \pm 0.90$ | $5.67 \pm 0.26$ |
| CPrompt | $83.60 \pm 0.00^*$ | $90.10 \pm 0.00$ | $6.47 \pm 0.00$ | $4.78 \pm 0.00$ | $74.23 \pm 0.17$ | $79.82 \pm 0.51$ | $5.98 \pm 0.24$ | $5.54 \pm 0.48$ |
| ConvPrompt | $87.21 \pm 0.20$ | $91.60 \pm 0.36$ | $5.47 \pm 0.33$ | $3.92 \pm 0.33$ | $73.93 \pm 0.36$ | $78.92 \pm 0.37$ | $4.87 \pm 0.57$ | $3.57 \pm 0.25$ |
| **LEAPGen-lite** | $\mathbf{95.28 \pm 3.37}$ | $\mathbf{98.38 \pm 1.13}$ | $\mathbf{1.08 \pm 1.54}$ | $\mathbf{0.70 \pm 0.99}$ | $\mathbf{83.67 \pm 0.39}$ | $\mathbf{85.65 \pm 0.33}$ | $\mathbf{1.06 \pm 0.24}$ | $\mathbf{0.47 \pm 0.14}$ |
| **LEAPGen** | $\mathbf{96.51 \pm 2.16}$ | $\mathbf{98.73 \pm 0.26}$ | $\mathbf{0.66 \pm 0.75}$ | $\mathbf{0.52 \pm 0.39}$ | $\mathbf{87.03 \pm 0.12}$ | $\mathbf{87.81 \pm 0.48}$ | $\mathbf{2.17 \pm 0.17}$ | $\mathbf{2.54 \pm 0.77}$ |

Table 2: Summarized numerical result on split CIFAR100 and ImageNet-R in 5, 10, and 20 tasks settings, * and ○ denote the results obtained from 1 seeded run and the first 10 tasks respectively due to system crash.

and CFM in all settings of CIFAR100. In addition, the magnitude of its FFM and CFM is lower than 0.7%, while the best-of-competitor method i.e. HiDE-Prompt suffers more than 1.5 %, and more than 2% for the rest of the SOTAs . Note that HiDe-Prompt failed to finish the experiment in the 20 tasks setting, and the result is obtained from its first 10 tasks. However, our method still achieves higher performance in the setting by approximately 1% margin. In Imagenet-R dataset, the proposed method outperforms the existing methods with a higher margin i.e. 3-30% for FAA and 2-24% for CAA. Our method also attains the lowest forgetting in both FFM and CFM with $\geq 1\%$ margin to the competitors. Similar to in CIFAR100 dataset, the difference of accuracy and forgetting are higher in the higher number of tasks. In CUB dataset, our method keeps its superiority to the previous methods. For the final performance i.e. FAA and FFM, our method outperforms the SOTAs with 1-21% and 0.6-10% respectively, while in terms of cumulative performance i.e. CAA and CFM, LEAPGen gains 3-16% and 1.4-8.5% respectively.

**b) Comparison to Different Approaches:** Table 4 compares the performance of our method to SOTAs of different approaches i.e. LoRA, slow learner (SLCA), and generative (GMM) approaches. Compared to LoRA approach, our method achieves a better performance with a significant margin i.e. 5-22% and 3-17% for FAA and CAA respectively. These margins are even higher than the margins to the prompt-based approach. SLCA achieves a promising performance as its performance is on par to ConvPrompt and Hide-Prompt in ImageNet-R

| Method | Split CUB 10 Tasks @20 classes/task | | | |
|---|---|---|---|---|
| | FAA | CAA | FFM | CFM |
| L2P | $66.95 \pm 0.13$ | $74.03 \pm 0.32$ | $5.18 \pm 0.11$ | $7.39 \pm 0.23$ |
| DualPrompt | $73.95 \pm 0.73$ | $80.29 \pm 0.15$ | $7.87 \pm 0.76$ | $8.60 \pm 0.37$ |
| CODA-P | $72.99 \pm 0.30$ | $82.64 \pm 0.64$ | $11.71 \pm 1.49$ | $9.91 \pm 1.04$ |
| LGCL | $79.93 \pm 0.30$ | $83.07 \pm 0.23$ | $5.45 \pm 0.33$ | $5.58 \pm 0.39$ |
| HiDe-Prompt | $87.21 \pm 0.18$ | $87.66 \pm 0.01$ | $1.90 \pm 0.45$ | $2.66 \pm 0.13$ |
| PGP | $78.35 \pm 0.68$ | $82.21 \pm 0.56$ | $5.76 \pm 0.10$ | $6.10 \pm 0.26$ |
| EvoPrompt | $76.23 \pm 0.51$ | $81.00 \pm 0.40$ | $3.96 \pm 0.43$ | $3.57 \pm 0.83$ |
| CPrompt | $77.14 \pm 1.16$ | $85.67 \pm 0.56$ | $11.65 \pm 0.47$ | $8.69 \pm 0.31$ |
| ConvPrompt | $80.12 \pm 1.37$ | $84.70 \pm 0.64$ | $6.04 \pm 0.97$ | $4.61 \pm 0.45$ |
| **LEAPGen-lite** | $\mathbf{86.00 \pm 0.32}$ | $\mathbf{88.03 \pm 0.47}$ | $\mathbf{2.46 \pm 1.61}$ | $\mathbf{2.29 \pm 1.15}$ |
| **LEAPGen** | $\mathbf{88.45 \pm 0.58}$ | $\mathbf{90.90 \pm 0.81}$ | $\mathbf{1.32 \pm 0.52}$ | $\mathbf{1.29 \pm 0.82}$ |

Table 3: Summarized result on split CUB (10-tasks).

10 tasks and CIFAR100 10 tasks settings respectively. However, our method is still in the higher performance than SLCA with 4-7% margin. Our method also achieves higher performance than the GMM method with 3-10% margin.

| Method | S-ImageNet-R 5 Tasks | | S-ImageNet-R 10 Tasks | | S-ImageNet-R 20 Tasks | | S-CIFAR100 10 Tasks | |
|---|---|---|---|---|---|---|---|---|
| | FAA | CAA | FAA | CAA | FAA | CAA | FAA | CAA |
| C-LoRA | 75.85 ± 0.31 | 78.85 ± 0.34 | 71.89 ± 0.45 | 75.33 ± 0.28 | 65.71 ± 0.60 | 70.63 ± 0.85 | 82.97 ± 0.47 | 87.06 ± 0.25 |
| LAE | 73.84 ± 0.14 | 77.29 ± 0.45 | 71.70 ± 0.39 | 76.71 ± 0.10 | 66.98 ± 0.35 | 73.72 ± 0.05 | 88.81 ± 0.34 | 91.59 ± 0.13 |
| InfLoRA | 77.52 ± 0.37 | 82.01 ± 0.12 | 75.65 ± 0.14 | 80.82 ± 0.24 | 71.01 ± 0.45 | 77.28 ± 0.45 | 89.84 ± 0.03 | 91.70 ± 0.32 |
| SLCA | - | - | 77.00 ± 0.33 | 81.17 ± 0.64 | - | - | 91.53 ± 0.28 | 94.09 ± 0.87 |
| GMM | - | - | 80.72 | - | - | - | 87.59 | - |
| **LEAPGen-lite** | **82.44 ± 0.63** | **84.37 ± 0.90** | **82.38 ± 1.04** | **85.14 ± 0.52** | **83.67 ± 0.39** | **85.65 ± 0.33** | **98.58 ± 0.03** | **98.69 ± 0.10** |
| **LEAPGen** | **82.79 ± 0.32** | **85.06 ± 0.29** | **84.09 ± 0.93** | **85.54 ± 0.65** | **87.03 ± 0.12** | **87.81 ± 0.48** | **98.38 ± 0.15** | **98.15 ± 0.39** |

Table 4: Comparison of the proposed method to the LoRA approach, SLCA, and GMM. All the results of the competitor methods is taken from respective refrences.

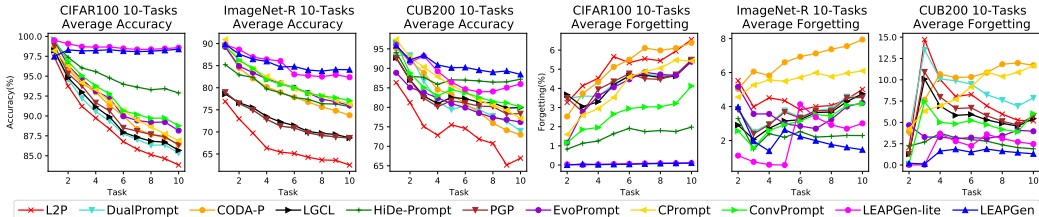

Figure 3: Historical performance plot of the consolidated methods in 3 CIL benchmark datasets.

**c) Historical Performance:** Figure 3 shows the historical average accuracy i.e. final average accuracy of all learned classes after finishing learning on each task $t$. LEAPGen's gentle slop shows that our proposed method experience the least performance degradation compared to the existing methods. In the CIFAR100 dataset, our method even manages to increase its performance in the later tasks. The existing methods suffer from a significant performance drop i.e up to 13%, 15%, 22% in CIFAR100, ImageNet-R, and CUB datasets respectively. For the average forgetting measure, our method once again demonstrates an excellent trend. Except on the first 5 tasks of ImageNet-R datasets, our method suffers from the lowest average forgetting in all tasks and all datasets. In CI-FAR100 datasets, our method experiences nearly zero forgetting from the first until the last tasks. In Imagenet-R and CUB datasets, our method manages to decrease its average forgetting in the 5 latest tasks, and consistently achieve the lowest magnitude i.e. $1.4 - 2.2\%$ and $1.2 - 1.8$ in CUB dataset. On the contrary, the competitor methods suffer from the increasing average forgetting during incremental learning, as shown by the rising curves. The methods suffer from a high average forgetting i.e. for up to 7%, 8%, and 15% in CIFAR100, ImageNet-R and CUB datasets respectively. Their increase of average forgetting is significant i.e. 4%, 3%, and 8% respectively. Overall, the historical analysis confirms how our method handles catastrophic forgetting better than the existing methods.

**d) LEAPGen-lite's Performance:** As shown in Table 2-4 and Figure 3 Despite utilizing far smaller (2.67% params) generators and without class descriptors generated by LLM, LEAPGen-lite still outperforms the existing methods significantly i.e. 4.7-25% FAA and 3-11% CAA in CIFAR100, and 3-30% FAA and 2-26% CAA in ImageNet-R dataset. LEAPGen-lite also achieves a low forgetting rate in these 2 datasets for all task-settings. In the CUB dataset, LEAPGen-lite archives a comparable performance to HiDe-Prompt and outperforms other SOTA with a significant margin i.e. 6-20% FAA and 3.3-14% CAA. This evidence proves our ideas i.e. language embedding as input for prompt generation, task-wise generators, soft task-id predictor, and learning with auxiliary data are not bounded by the generated descriptors and the size of generators.

**e) Ablation Study:** Table 5 shows the impacts of the proposed method's components on its performance. **(1) Descriptor Embedding** $E$ as one of LEAPGen's main components, elevates fine tuning (FT) baseline significantly i.e. $> 11\%$ FAA and $> 13\%$ CAA. It shows the promising impact of our idea . **(2) Task Key** $K^t$ **and Loss** $\mathcal{L}_t$ transform the method into task-wise prompt-

| Component | Loss | FAA | CAA | FFM | CFM |
|---|---|---|---|---|---|
| FT | $\mathcal{L}_{intra}$ | 61.5 | 65.3 | 5.9 | 5.6 |
| FT+$E$ | $\mathcal{L}_{intra}$ | 72.9 | 78.5 | 5.8 | 5.3 |
| FT+$E$+$K^t$ | $\mathcal{L}_{intra}, \mathcal{L}_t$ | 74.9 | 79.5 | 2.8 | 3.2 |
| FT+$G^t(E)$+$K^t$ | $\mathcal{L}_{intra}, \mathcal{L}_t$ | 76.6 | 80.5 | 1.8 | 2.4 |
| FT+$G^t(E)$+$K^t$+$L_c^t$ | $\mathcal{L}_{intra}, \mathcal{L}_t, \mathcal{L}_c$ | 77.2 | 81.0 | 0.5 | 1.3 |
| FT+$G^t(E)$+$K^t$+$L_c^t$+Aux | $\mathcal{L}_{intra}, \mathcal{L}_t, \mathcal{L}_c$ | 78.4 | 82.0 | 0.5 | 1.2 |
| FT+$G^t(E)$+$K^t$+$L_c^t$+Aux | $\mathcal{L}_{intra}, \mathcal{L}_{inter}, \mathcal{L}_t, \mathcal{L}_c$ | 83.7 | 86.0 | 2.5 | 1.9 |
| FT+$G^t(E)$+$K^t$+$L_c^t$+Aux+ Soft Task-ID Predictor | $\mathcal{L}_{intra}, \mathcal{L}_{inter}, \mathcal{L}_t, \mathcal{L}_c$ | 84.7 | 86.1 | 0.9 | 1.4 |

Table 5: Ablation study on the impact of LEAPGen components to its performance, run on 10 tasks Imagenet-R.

ing based on input to $K^t$ and $\mathcal{L}_t$ similarity. They produce a better model with 1-2% FAA and CAA improvement and reduce FFM and CFM by $> 2\%$. It shows the effectiveness of task-wise decomposition as applied in the task-wise prompt approach. **(3) Task-wise generator** $G^t()$ enhances the model recognition capability by 2% and 1% for FAA and CAA respectively. It also decreases its forgetting rate by 1%. It means the trainable generator produces a more discriminative prompt than

the descriptor embedding only. **(4)** $L_c^t$ **Class-wise Key** $L_c^t$ **and Loss** $\mathcal{L}_c$ for top-k embedding selection i.e. $E_i$ for $i \in [1...k]$ improves the model performance with 1% margin. It also reduces the model forgetting rate with up to 1% margin. Thus, measuring the input similarity to $L_c^t$ offers a better embedding selection than to $E_c^t$ directly. **(5) Auxiliary** embedding improves the LEAPGen's performance with a more than 1% margin. It shows the secondary contribution of descriptors embedding as an auxiliary modality along with its primary role in prompt generation. **(6) Inter-tasks loss** $\mathcal{L}_{inter}$ improves LEAPGen accuracy significantly i.e. 5.4% and 4.0% for FAA and CAA respectively even though it increases FFM and CFM by $0.5 - 2\%$. It contributes to balancing the knowledge of previously learned classes (stability) and currently learned classes (plasticity), and emphasizes the risk of forgetting previously learned classes. **(7) Soft task-id predictor** substitution to conventional task-id (Wang et al., 2022b) improves the accuracy with up to 1% and reduces FFM and CFM by 1.5% and 0.5% respectively. It implies that our designed task-id predictor outperforms the existing task-id predictor both in accuracy and forgetting.

**f) Sensitivity Analysis :** Figure 4 shows the sensitivity analysis of our proposed method in various settings i.e. different **prompt lengths (left), layer depths (center), and number of generators per task(right)**. LEAPGen maintains its stable FAA in any prompt length values i.e. within the range of $84.2 - 84.7\%$, while the existing methods i.e. ConvPrompt, CODA-P, DualPrompt, and L2P achieve significantly lower FAA i.e. $\leq 78\%$. LEAPGen with the

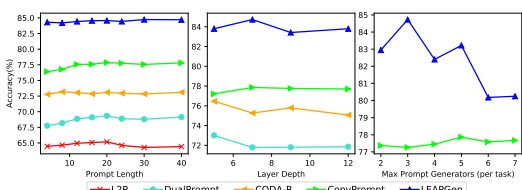

Figure 4: FAA of the consolidated methods under different prompt lengths (left), layer depth (middle), and maximum generators per task (right)

shortest prompt (4) has a higher FAA than the existing methods with the longest prompt (40), despite the 10 times smaller size. LEAPGen holds its stable performance at various depths of prompted VIT layers. LEAPGen consistently manages to achieve $83.5 - 84.7\%$ FAA, while other methods achieve far lower FAA i.e. $\leq 78\%$. The best layer depth for LEAPGen and ConvPrompt is 7, while the best depth for the others is 5. The number of generators turns out to be the most sensitive parameter to LEAPGen, as it is saturated after 4 generators per tasks. Considering the number of generators in LEAPGen is the same number as the selected descriptor embedding, it is quite logical that LEAPGen becomes counterproductive after some threshold $k$.

**g) Parameters, Running Time, and Storage:** Table 6 compares the number of executed parameters, running time (ImageNet-R), and storage of our methods and existing SOTAs. Despite having a higher number of parameters and storage, LEAPGen consumes less running time than existing SOTAs both training+inference and total running time. LEAPGen-lite consumes the least costs in total running time and

| Method | Descriptors | #Params(M) | Running Time (h) | | | | Storage (MB) |
|---|---|---|---|---|---|---|---|
| | | | T.Desc | Inf | Tr+Inf | Total | |
| HiDe-Prompt | - | 0.15 | - | 0.019 | 5.40 | 5.40 | 334 |
| ConvPrompt | ✓ | 1.28 | 1.07 | 0.033 | 1.04 | 2.11 | 346 |
| LEAPGen-lite | - | 0.16 | - | 0.028 | 0.53 | 0.53 | 332 |
| LEAPGen | ✓ | 6.35 | 1.07 | 0.025 | 0.72 | 1.79 | 567 |

Table 6: Cost comparison of the methods, T.Desc, Tr, and Inf denote time for generating descriptors, training, and inference respectively, detailed in Appendix D.3.

storage and requires relatively low parameters and inference time. LEAPGen and ConvPrompt require additional time to generate descriptors that increase their total simulation time. Despite having the least parameters, Hide-Prompt requires the longest training and total times since it needs extra operations to generate uninstructed class representations. Please see Appendix D for our detailed **extended analysis** on task-key similarity, language embedding similarity, performance-and-cost trade-off, class name as descriptor, and performance on various descriptors and types of generators.

## 6    CONCLUDING REMARKS

We identify the prompt-based approach dilemma and propose a novel prompt-based structure and algorithm that incorporates 4 key concepts (1) language as input for prompt generation (2) task-wise generators (3) limiting matching descriptors search space via soft task-id prediction (4) generated prompt as auxiliary data. Our experimental analysis shows the superiority of our method to existing SOTAs in CIFAR100, ImageNet-R, and CUB dataset with a significant margin i.e. up to 30% final average accuracy, 24% cumulative average accuracy, 8% final forgetting measure, and 7% cumulative forgetting measure. Our historical analysis confirms our method successfully maintains stability-plasticity in every task. Our robustness analysis shows our method consistently achieves higher performance in various prompt lengths, layer depths, and number of generators per task compared to the SOTAs.

ACKNOWLEDGMENTS

M. Anwar Ma'sum acknowledges the support of Tokopedia-UI Centre of Excellence for GPU access to run the experiments. Savitha Ramasamy acknowledges the programme DesCartes supported by the National Research Foundation, Prime Minister's Office, Singapore under its Campus for Research Excellence and Technological Enterprise (CREATE) programme.

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

# A  THEORETICAL ANALYSIS

We state the following theorem to prove the effectiveness of our proposed method in a theoretical analysis.

**Theorem 1:** *Let $f_\psi$ and $h_\psi$ are the ViT encoder and classifier head respectively, $\psi$ is frozen, $\theta = \{G_i^t, K^t, L_c^t\}_{t=1,c \in C^t}^{t=T}$ is the set of trainable parameters and generators, $\theta' = \{K^t, L_c^t\}_{t=1,c \in C^t}^{t=T}$ is the set of trainable parameters only, and $\mathcal{L}$ is loss function. Assume that $G$, $f$, $h$, and $\mathcal{L}$ are continues functions in $\mathbb{R}$, if $G_i^t$ can produce identity output, then there is $\epsilon \geq 0$ that satisfies $\mathcal{L}(h_\phi(f_{\psi,\theta'}(x))) - \mathcal{L}(h_\phi(f_{\psi,\theta}(x))) = \epsilon.$.*

**Theorem 2:** *Assuming a pre-trained ViT as the backbone, if the loss error $\mathcal{L} \leq \delta$ then exist $\mathcal{L}_{intra} \leq \delta$ and $\mathcal{L}_{inter} \leq \delta$.*

**Theorem 3:** *Let $F$ and $F'$ denote the task-pedictor functions parameterized by $(\{K^t\}, \{L_c^t\})$ and $\{K^t\}$ respectively, and $cos(,)$ denote cosine similarity function. If $K^t$ and $L_x^t$ are trained well enough so that $cos(x, K^t) \approx cos(x, L_c^t) \geq S$, and $cos(x, K^{t'}) \approx cos(x, L_{c'}^t) \approx cos(x, L_c^{t'}) \leq 2/3S$ for any $x \in \mathcal{X}_c^t$ then it guarantees $P(x \in \mathcal{X}^t | \mathcal{T}, \theta_F) \geq P(x \in \mathcal{X}^t | \mathcal{T}, \theta_{F'})$.*

Theorem 1 theoretically shows there is always a chance to achieve a smaller loss with generators than with descriptor embedding only. Theorem 2 shows that minimizing the loss error implies minimizing both inter-task and intra-task losses. Theorem 3 shows that if the learnable parameters are well-trained then it guarantees our task predictor as presented in equation 3 is more accurate than the existing task-id predictor (based on $\{K^t\}$ only). Please see in Appendix A for the detailed analysis.

## A.1  PROOF OF THEOREM 1

Recall that $f_\psi$ is pre-trained ViT encoder with a frozen parameter $\psi$, $h_\phi$ is classifier head parameterized by $\phi$ (learnable), $\theta = \{G_i^t, K^t, L_c^t\}_{t=1,c \in C^t}^{t=T}$ is the set of learnable parameters and generators, $\theta' = \{K^t, L_c^t\}_{t=1,c \in C^t}^{t=T}$ is the set of learnable parameters only. Note that $G_i^t$ produces a prompt component $PC_i = G_i^t(E_i)$. $\theta' = \{K^t, L_c^t\}_{t=1,c \in C^t}^{t=T}$ is the same as using identity function $I$ as generator. $\mathcal{L}$ is the loss function that in this case is cross-entropy. First we adopt the definition of continuity of $f(x)$ at $a$ i.e. for all $\eta_1 > 0$ there is some $\delta_1$ such that:

$$|x - a| < \delta_1 \Rightarrow |f(x) - f(a)| < \eta_1 \tag{A11}$$

Now, following the assumption that $f(.)$ is a continuous function in $\mathbb{R}$, now we substitute $f(.)$ with $f_{\psi,\theta}(.)$, subtitute $x$ with $G_i^t(E_i)$ and $a$ with $I(E_i)$. Not that $G_i^t$ is a trainable generator that can produce identity function $I$ as well as nearby vectors i.e. $G_i^t(E_i)$. Also note that $f_{\psi,\theta,(G=I)}(.) = f_{\psi,\theta'}(.)$ Therefore, the equation A11 can be derived into:

$$|G_i^t(E_i) - I(E_i)| < \delta_1 \Rightarrow |f_{\psi,\theta}(x) - f_{\psi,\theta'}(x)| < \eta_1 . \tag{A12}$$

Then, incorporating the continuity of $h(.)$ in $\mathbb{R}$ and the definition of conituinous function then we have

$$|f_{\psi,\theta}(x) - f_{\psi,\theta'}(x)| < \eta_1 \Rightarrow |h_\phi(f_{\psi,\theta}(x)) - h_\theta(f_{\psi,\theta'}(x))| < \eta_2 . \tag{A13}$$

Analogically, applying the continuity of $\mathcal{L}(.)$ in $\mathbb{R}$, then we have

$$|h_\phi(f_{\psi,\theta}(x)) - h_\theta(f_{\psi,\theta'}(x))| < \eta_2 \Rightarrow |\mathcal{L}(h_\phi(f_{\psi,\theta}(x))) - \mathcal{L}(h_\theta(f_{\psi,\theta'}(x)))| < \eta_3 \tag{A14}$$

Combining the equations A12, A13, A14, then we have

$$|G_i^t(E_i) - I(E_i)| < \delta_1 \Rightarrow |\mathcal{L}(h_\phi(f_{\psi,\theta}(x))) - \mathcal{L}(h_\theta(f_{\psi,\theta'}(x)))| < \eta_3 . \tag{A15}$$

In the equation above we can pick two constant $0 < \eta_3' < \eta_3$ and $0 < \delta_1' < \delta_1$ tat satisfies

$$|G_i^t(E_i) - I(E_i)| = \delta_1' \Rightarrow |\mathcal{L}(h_\phi(f_{\psi,\theta}(x))) - \mathcal{L}(h_\theta(f_{\psi,\theta'}(x)))| = \eta_3' . \tag{A16}$$

The term $|\mathcal{L}(h_\phi(f_{\psi,\theta}(x))) - \mathcal{L}(h_\theta(f_{\psi,\theta'}(x)))| = \eta_3$ can be satisfied by two conditions: (1) $\mathcal{L}(h_\phi(f_{\psi,\theta'}(x))) - \mathcal{L}(h_\theta(f_{\psi,\theta}(x))) = \eta_3$ if $\mathcal{L}(h_\phi(f_{\psi,\theta'}(x))) > \mathcal{L}(h_\theta(f_{\psi,\theta}(x)))$. In this condition, then we can state the following statement:

$$\mathcal{L}(h_\phi(f_{\psi,\theta'}(x))) - \mathcal{L}(h_\theta(f_{\psi,\theta}(x))) > 0 \tag{A17}$$

The other condition (2) $\mathcal{L}(h_\phi(f_{\psi,\theta}(x))) - \mathcal{L}(h_\theta(f_{\psi,\theta'}(x))) = \eta_3$ if $\mathcal{L}(h_\phi(f_{\psi,\theta}(x))) > \mathcal{L}(h_\theta(f_{\psi,\theta'}(x)))$. However, we didn't use this condition as the optimized $G_i^t$, instead, we select $G_i^t = I$ so that $\mathcal{L}(h_\phi(f_{\psi,\theta}(x))) = \mathcal{L}(h_\theta(f_{\psi,\theta'}(x)))$. The second condition shows the case where the best possible value of $G_i^t$ is the identity function itself. In the second condition, then we can state

$$\mathcal{L}(h_\phi(f_{\psi,\theta'}(x))) - \mathcal{L}(h_\theta(f_{\psi,\theta}(x))) = 0 \tag{A18}$$

Combining both inequations A17 and A18, then we prove **Theorem 1** that states : Assume that $G$, $f$, $h$, and $\mathcal{L}$ are continues functions in $\mathbb{R}$, if $G_i^t$ can produce identity output, then there is $\epsilon \geq 0$ that satisfies $\mathcal{L}(h_\phi(f_{\psi,\theta'}(x))) - \mathcal{L}(h_\phi(f_{\psi,\theta}(x))) = \epsilon$. Theorem 1 shows that a possibility to achieve better or at least similar learning performance with learnable generator networks than without them.

### A.2 PROOF OF THEOREM 2

Given a model with loss error $\mathcal{L} \leq \delta$, then

$$-\sum_{t=1}^{t=T} \sum_{c \in C^t} log \frac{exp(h_\phi(f_{\psi,\theta}(x))[c])}{\sum_{t=1}^{t=T} \sum_{c' \in C^t} exp(h_\phi(f_{\psi,\theta}(x))[c'])} \leq \delta \tag{A19}$$

$$\mathcal{L}_{inter}(x, (\psi,\theta,\phi)) = -\sum_{t=1}^{t=T} \sum_{c \in C^t} log \frac{exp(h_\phi(f_{\psi,\theta}(x))[c])}{\sum_{t=1}^{t=T} \sum_{c' \in C^t} exp(h_\phi(f_{\psi,\theta}(x))[c'])} \leq \delta \tag{A20}$$

Or simply

$$\mathcal{L}_{inter}(x, (\psi,\theta,\phi)) \leq \delta \tag{A21}$$

From equation 4 We get intra-task probability as

$$P(x \in \mathcal{X}_c^t | x \in \mathcal{X}^t, \mathcal{T}, \theta) = P(x \in \mathcal{X}_c^t | \mathcal{T}, \theta) / P(x \in \mathcal{X}^t | \mathcal{T}, \theta) \tag{A22}$$

Since $P(x \in \mathcal{X}_c^t | x \in \mathcal{X}^t, \mathcal{T}, \theta)$, $P(x \in \mathcal{X}_c^t | \mathcal{T}, \theta)$, and $P(x \in \mathcal{X}^t | \mathcal{T}, \theta)$ are in the range of $(0, 1)$ then we have:

$$P(x \in \mathcal{X}_c^t | x \in \mathcal{X}^t, \mathcal{T}, \theta) \geq P(x \in \mathcal{X}_c^t | \mathcal{T}, \theta) \tag{A23}$$

Applying log to both sides, then we have

$$log(P(x \in \mathcal{X}_c^t | x \in \mathcal{X}^t, \mathcal{T}, \theta)) \geq log(P(x \in \mathcal{X}_c^t | \mathcal{T}, \theta)) \tag{A24}$$

Multiplying both sides with $-1$, then we have

$$-log(P(x \in \mathcal{X}_c^t | x \in \mathcal{X}^t, \mathcal{T}, \theta)) \leq -log(P(x \in \mathcal{X}_c^t | \mathcal{T}, \theta)) \tag{A25}$$

Let inter-task probability and intra task probability represented as:

$$P(x \in \mathcal{X}_c^t | \mathcal{T}, \theta) = \sum_{t=1}^{t=T} \sum_{c \in C^t} \frac{exp(h_\phi(f_{\psi,\theta}(x))[c])}{\sum_{t=1}^{t=T} \sum_{c' \in C^t} exp(h_\phi(f_{\psi,\theta}(x))[c'])}$$

$$P(x \in \mathcal{X}_c^t | x \in \mathcal{X}^t, \mathcal{T}, \theta) = \sum_{c \in C^t} \frac{exp(h_\phi(f_{\psi,\theta}(x))[c])}{\sum_{c' \in C^t} exp(h_\phi(f_{\psi,\theta}(x))[c'])} \tag{A26}$$

Substituting equations A26 into A25 and combining equation A21, then we have:

$$-log(\sum_{c \in C^t} \frac{exp(h_\phi(f_{\psi,\theta}(x))[c])}{\sum_{c' \in C^t} exp(h_\phi(f_{\psi,\theta}(x))[c'])}) \leq -log(\sum_{t=1}^{t=T} \sum_{c \in C^t} \frac{exp(h_\phi(f_{\psi,\theta}(x))[c])}{\sum_{t=1}^{t=T} \sum_{c' \in C^t} exp(h_\phi(f_{\psi,\theta}(x))[c'])}) \leq \delta \tag{A27}$$

$$-\sum_{c \in C^t} log \frac{exp(h_\phi(f_{\psi,\theta}(x))[c])}{\sum_{c' \in C^t} exp(h_\phi(f_{\psi,\theta}(x))[c'])} \leq \sum_{t=1}^{t=T} \sum_{c \in C^t} log \frac{exp(h_\phi(f_{\psi,\theta}(x))[c])}{\sum_{t=1}^{t=T} \sum_{c' \in C^t} exp(h_\phi(f_{\psi,\theta}(x))[c'])} \leq \delta \tag{A28}$$

Thefore, we have

$$\mathcal{L}_{intra}(x, (\psi, \theta, \phi)) = -\sum_{c \in C^t} log \frac{exp(h_\phi(f_{\psi,\theta}(x))[c])}{\sum_{c' \in C^t} exp(h_\phi(f_{\psi,\theta}(x))[c'])} \leq \delta \tag{A29}$$

or in the simplified form

$$\mathcal{L}_{intra}(x, (\psi, \theta, \phi)) \leq \delta \tag{A30}$$

The explanation above proves Theorem 2 that assuming a pre-trained ViT as the backbone, if the loss error $\mathcal{L} \leq \delta$ then exist $\mathcal{L}_{intra} \leq \delta$ and $\mathcal{L}_{inter} \leq \delta$.

## A.3 PROOF OF THEOREM 3

Let $F$ and $F'$ denote the task-predictor functions parameterized by $(\{K^t\}, \{L_c^t\})$ and $\{K^t\}$ respectively therefore, $F$ is presented in equation 3 and simplified into equation A31 while $F'$ is presented in equation A32.

$$F(x) = t = \underset{t \in [1..T], c \in C^t}{\arg\max} \{cos(x, K^t) cos(x, L_c^t)\} \tag{A31}$$

$$F'(x) = t = \underset{t \in [1..T],}{\arg\max} \{(x.K^t)/(max(||x||_2.||K^t||_2, \epsilon))\} = \underset{t \in [1..T],}{\arg\max} \{cos(x, K^t)\} \tag{A32}$$

Let $cos(.)$ denote cosine similarity function, $\mathcal{L}$ denotes similarity loss i.e. difference between the right and wrong representation. Therefore, for any $x \in \mathcal{X}_c^t$ then we have

$$\mathcal{L}(F(x)) = cos(x, K^t) cos(x, L_c^t) - \frac{1}{(T-1)|C^{t'}|} \sum_{t' \neq t} \sum_{c \in C^{t'}} cos(x, K^{t'}) cos(x, L_c^{t'})$$
$$- \frac{1}{(|C^t|-1)} \sum_{c' \neq c \in C^t} cos(x, K^t) cos(x, L_{c'}^t) \tag{A33}$$

$$\mathcal{L}(F'(x)) = cos(x, K^t) - \frac{1}{T-1} \sum_{t' \neq t} cos(x, K^{t'}) \tag{A34}$$

Thus minimizing $\mathcal{L}(F(x))$ and $\mathcal{L}F'(x)$ imply maximizing $F$ and $F'$ respectively. Suppose that $K^t$ and $L_x^t$ are trained well enough so that for any $x \in \mathcal{X}_c^t$, it satisfy $cos(x, K^t) \approx S_1$, and $cos(x, K^{t'}) \approx S_a = S_1 - \epsilon_1$, $cos(x, L_c^t) \approx S_2$ and $cos(x, L_{c' \neq c}^t) \approx cos(x, L_c^{t'}) \approx S_b = S_2 - \epsilon_2$, where $S_1, S_2, S_a, S_b, \epsilon_1$, and $\epsilon_2$ have positive values. Then equations A33 can be derived into

$$\mathcal{L}(F(x)) = S_1 S_2 - \frac{1}{(T-1)|C^{t'}|} \sum_{t' \neq t} \sum_{c \in C^{t'}} S_a S_b - \frac{1}{(|C^t|-1)} \sum_{c' \neq c \in C^t} S_1 S_b \tag{A35}$$

$$\mathcal{L}(F(x)) = S_1 S_2 - S_a S_b - S_1 S_b \tag{A36}$$

and A34 can be derived into

$$\mathcal{L}(F'(x)) = S_1 - \frac{1}{T-1} \sum_{t' \neq t} S_a \tag{A37}$$

$$\mathcal{L}(F'(x)) = S_1 - S_a \tag{A38}$$

Now, assuming that $S_1 \approx S_2 \approx S$ and $\epsilon_1 \approx \epsilon_2 \approx \epsilon$, then we have

$$\mathcal{L}(F(x)) = S^2 - (S - \epsilon)(S - \epsilon) - S(S - \epsilon) \tag{A39}$$

$$\mathcal{L}(F(x)) = S^2 - (S^2 - 2S\epsilon + \epsilon^2) - (S^2 - S\epsilon) \tag{A40}$$

$$\mathcal{L}(F(x)) = S^2 - S^2 + 2S\epsilon - \epsilon^2 - S^2 + S\epsilon \tag{A41}$$

$$\mathcal{L}(F(x)) = 3S\epsilon - S^2 - \epsilon^2 \tag{A42}$$

and we have,

$$\mathcal{L}(F'(x)) = S_1 - S_a = S - (S - \epsilon) = \epsilon \tag{A43}$$

Suppose that we want to find a condition where $\mathcal{L}(F(x))\mathcal{L} \leq (F'(x))$, then we have

$$3S\epsilon - (S^2 + \epsilon^2) \leq \epsilon \tag{A44}$$

$$3S\epsilon - S^2 - \epsilon^2 \leq \epsilon \tag{A45}$$

Note that $\epsilon < S$. Then we can choose $\epsilon = 1/3S$, so the equation above becomes

$$3S(1/3)S - S^2 - \epsilon^2 \leq \epsilon \tag{A46}$$

$$S^2 - S^2 - \epsilon^2 \leq \epsilon \tag{A47}$$

$$-\epsilon^2 \leq \epsilon \tag{A48}$$

that always true for any $\epsilon > 0$. Note that $\epsilon = (1/3)S$ means $S_a = S - \epsilon = S - (1/3)S = (2/3)S$. Choosing $\epsilon < (1/3)S$ also satisfies the inequation above. The derivation above proves Theorem 3:i.e. let $F$ and $F'$ denote the task-pedictor functions parameterized by $(\{K^t\}, \{L_c^t\})$ and $\{K^t\}$ respectively, and $cos(,)$ denote cosine similarity function. If $K^t$ and $L_x^t$ are trained well enough so that $cos(x, K^t) \approx cos(x, L_c^t) \geq S$, and $cos(x, K^{t'}) \approx cos(x, L_{c'}^t) \approx cos(x, L_c^{t'}) \leq 2/3S$ for any $x \in \mathcal{X}_c^t$ then it guarantees $\mathcal{L}(F(x)) \leq \mathcal{L}(F'(x))$ that implies $P(x \in \mathcal{X}^t | \mathcal{T}, \theta_F) \geq P(x \in \mathcal{X}^t | \mathcal{T}, \theta_{F'})$.

## B  PSEUDO-CODE OF LEAPGEN ALGORITHM

In this section we present the procedures of LEAPGen training and inference that are shown in algorithm 1 and 2 respectively.

---

**Algorithm 1** LEAPGen Training

---

1: **Input:** A sequence of tasks $\mathcal{T}^1, \mathcal{T}^2,...,\mathcal{T}^T$, a frozen $L$-layered (with $H$ MSA heads/layer) pre-trained ViT $f_{\psi(.)}$, training epochs $E$, and batch size $B$.
2: **for** $t = 1 : T$ **do**
3:     Initiate trainable parameters for task $t$, $\theta^t = \{\{G_i^t\}_{i=1}^k, K^t, \{L_c^t\}_{c=1}^{|C^t|}\}$
4:     $\mathcal{B} \leftarrow$ Split $\mathcal{T}^t$ into $B$ sized batches
5:     **for** $e = 1 : E$ **do**
6:         **for** $b = 1 : \mathcal{B}$ **do**
7:             Find top-$k$ $L_c^j$ where $j \in [1..t]$, $c \in [1..|C^j|]$
8:             Generate prompt $P = \{(P_{l,h}^K, P_{l,h}^V)\}_{l\in[1..L],h\in[1..H]}$ as in eq. 2
9:             Compute logits by forwarding the input i.e. $h_\phi(f_{\psi,\theta}(x))$
10:             Compute $\mathcal{L}_{intra}$ as in eq. 6
11:             Compute $\mathcal{L}_{inter}$ as in eq. 7
12:             Compute $\mathcal{L}_t$ as in eq. 8
13:             Compute $\mathcal{L}_c$ as in eq. 9
14:             Compute $\mathcal{L}_{total}$ as in eq. 10
15:             Update parameters $\theta^t = \{\{G_i^t\}_{i=1}^k, K^t, \{L_c^t\}_{c=1}^{|C^t|}\}$ based on $\mathcal{L}_{total}$
16:         **end for**
17:     **end for**
18: **end for**
19: **Output:** Optimum parameters $\theta = \{theta^t\}_{t=1}^T = \{\{G_i^t\}_{i=1}^k, K^t, \{L_c^t\}_{c=1}^{|C^t|}\}_{t=1}^T$

---

## C  COMPLEXITY ANALYSIS

Following the pseudo-code in Algorithm 1, LEAPGen has several atomic operations e.g. selecting top-$k$ $L_c^i$ (line 7), prompt generation (line 8), computing logits (line 9), computing loss (line 10-14) and update parameter (line 15). Let $N^t = |\mathcal{T}|^t$, and accumulating the operations on all batches,

---

**Algorithm 2** LEAPGen Inference

---

1: **Input:** An input $x$, a frozen $L$-layered (with $H$ MSA heads/layer) pre-trained ViT $f_{\psi(.)}$ , and optimized parameters $\theta = \{theta^t\}_{t=1}^T = \{\{G_i^t\}_{i=1}^k, K^t, \{L_c^t\}_{c=1}^{|C^t|}\}_{t=1}^T$
2: Predict task-id $t$ using soft task-id prediction as in eq. 3
3: Find top-$k$ $L_c^j$ where $j \in [1..t]$, $c \in [1..|C^j|]$
4: Generate prompt $P = \{(P_{l,h}^K, P_{l,h}^V)\}_{l \in [1..L], h \in [1..H]}$ as in eq. 2
5: Compute logits by forwarding the input i.e. $logits = h_\phi(f_{\psi,\theta}(x))$
6: Compute Pedicted label $\hat{y} = argmax(logits)$
7: **Output:** Predicted label $\hat{y}$

---

then selecting top-k $L_c^i$ costs $O(N^t.T.|C^t|)$, the prompt generation costs $O(N^t.k)$, computing losses costs $O(N^t)$, and parameters update cost costs $O(N^t)$, then the the complexity of LEAPGen will be:

$$O(LEAPGen) = T.E.(O(N^t.T.|C^t|) + O(N^t.k) + O(N^t) + O(N^t)) \tag{A49}$$

$$O(LEAPGen) = T.E.(O(N^t.T.|C^t|) + O(N^t.k) + O(N^t)) \tag{A50}$$

$$O(LEAPGen) = E.(O(T.N^t.T.|C^t|) + O(T.N^t.k) + O(T.N^t)) \tag{A51}$$

Note that $T.N^t = N$ that represents the number of all samples from the first task until the last task, and $T.|C^t| = |C|$ that represents the number of all learned classes where $|C| << N$, and $k$ is small constant ¡ 10. Then LEAPGen complexity becomes

$$O(LEAPGen) = E.(O(N|C|) + O(Nk) + O(N)) \tag{A52}$$

$$O(LEAPGen) = E.(O(N|C|)) \tag{A53}$$

$$O(LEAPGen) = O(EN|C|) \tag{A54}$$

Assuming that learning epoch $E$ and number of classes $|C|$ are constant, then the equation above is derived into:

$$O(LEAPGen) = O(EN|C|) = O(N) \tag{A55}$$

# D    EXTENDED ANALYSIS

## D.1    HIGH SIMILARITY OF TASK KEY VECTORS

We investigate the further detail of conventional task-id prediction computed by cosine similarity between input $x$ and task-key $K$ as implemented by DualPrompt. After completing the 10 task learning on the ImageNet-R dataset, we evaluate the similarity between task keys. Table A7 shows the cosine similarity between $K^t$ and $K^{t'}$ for all $t, t' \in [1..T]$. The tables show that a pair of different task keys has a high similarity. For example, except for task-3, task-1 has 0.8 to 0.96 cosine similarity to other task keys. During the inference step, this high similarity between different task keys causes a false task prediction that leads to the wrong selected prompt, unexpected feature value, and finally, misclassification.

This empirical analysis is one of our motivations to develop a better task-id prediction i.e. by utilizing both task-wise key $K^t$ and class-wise key $L_c^t$. Combining the similarity score of input $x$ to both $K^t$ and key $L_c^t$ produces a more discriminative task prediction than using $K^t$ only. Another way to solve this issue is by creating an uninstructed representation where a class may have several prototypes as used in HiDe-Prompt. However, this mechanism takes a longer time since the uninstructed representation optimization needs prototype augmentation and many iteration updates.

| | Task-1 | Task-2 | Task-3 | Task-4 | Task-5 | Task-6 | Task-7 | Task-8 | Task-9 | Task-10 |
|---|---|---|---|---|---|---|---|---|---|---|
| Task-1 | 1.000 | 0.961 | 0.728 | 0.857 | 0.932 | 0.901 | 0.866 | 0.827 | 0.809 | 0.877 |
| Task-2 | 0.961 | 1.000 | 0.749 | 0.866 | 0.932 | 0.909 | 0.879 | 0.844 | 0.825 | 0.899 |
| Task-3 | 0.728 | 0.749 | 1.000 | 0.873 | 0.738 | 0.785 | 0.725 | 0.672 | 0.660 | 0.687 |
| Task-4 | 0.857 | 0.866 | 0.873 | 1.000 | 0.872 | 0.893 | 0.828 | 0.771 | 0.755 | 0.806 |
| Task-5 | 0.932 | 0.932 | 0.738 | 0.872 | 1.000 | 0.921 | 0.893 | 0.850 | 0.843 | 0.908 |
| Task-6 | 0.901 | 0.909 | 0.785 | 0.893 | 0.921 | 1.000 | 0.933 | 0.893 | 0.873 | 0.893 |
| Task-7 | 0.866 | 0.879 | 0.725 | 0.828 | 0.893 | 0.933 | 1.000 | 0.937 | 0.920 | 0.907 |
| Task-8 | 0.827 | 0.844 | 0.672 | 0.771 | 0.850 | 0.893 | 0.937 | 1.000 | 0.905 | 0.857 |
| Task-9 | 0.809 | 0.825 | 0.660 | 0.755 | 0.843 | 0.873 | 0.920 | 0.905 | 1.000 | 0.872 |
| Task-10 | 0.877 | 0.899 | 0.687 | 0.806 | 0.908 | 0.893 | 0.907 | 0.857 | 0.872 | 1.000 |

Table A7: Cosine similarity of task-key vectors after completing 10-task training on ImageNet-R by DualPrompt method.

## D.2 HIGH SIMILARITY OF LANGUAGE EMBEDDING

We continue our deeper investigation on the encoded language embedding as implemented in previous SOTAs i.e. LGCL. LGCL generates the language embedding by forming the string "The photo of class name". then patches it into the CLIP encoder. Surprisingly, prototypes from some classes have high similarity to the other classes' prototypes. For example, in class 'Great White Shark' has 0.9 similarities to the prototypes of class 'Tree Frog' and 'Iguana', despite those classes being totally different and not the sub-classes of the same superclass. On the contrary, the class 'Great White Shark' has lower similarity to the class 'Goldfish' which is despite both of them are the species from the same super-category i.e. fish.

In that case, we can't utilize the language embedding as a reference or anchor in the training process as it may pull the adjusted (ViT) prototype closer to its value, while its value is highly similar to other classes' prototypes. Thus, in dealing with this challenge, we choose to use the language embedding as an input for prompt generation. In a such way, the trainable generators can produce a more discriminative prompt than the undiscriminating language embedding.

| | 'Gold-fish' | 'Great White Shark' | 'Ham-mer-head' | 'Sting-ray' | 'Hen' | 'Os-trich' | 'Gold-finch' | 'Junco' | 'Bald Eagle' | 'Vul-ture' | 'Newt' | 'Axo-lotl' | 'Tree Frog' | 'Igu-ana' | 'Afri-can Chame-leon' | 'Co-bra' | 'Scor-pion' | 'Taran-tula' | 'Centi-pede' | 'Pea-cock' |
|---|---|---|---|---|---|---|---|---|---|---|---|---|---|---|---|---|---|---|---|---|
| 'Goldfish' | 1.000 | 0.711 | 0.732 | 0.724 | 0.659 | 0.766 | 0.746 | 0.750 | 0.697 | 0.424 | 0.589 | 0.734 | 0.714 | 0.759 | 0.724 | 0.682 | 0.595 | 0.734 | 0.565 | 0.736 |
| 'Great White Shark' | 0.711 | 1.000 | 0.916 | 0.608 | 0.760 | 0.719 | 0.848 | 0.887 | 0.744 | 0.538 | 0.719 | 0.901 | 0.904 | 0.896 | 0.654 | 0.668 | 0.751 | 0.824 | 0.577 | 0.882 |
| 'Hammer-head' | 0.732 | 0.916 | 1.000 | 0.631 | 0.654 | 0.714 | 0.811 | 0.878 | 0.657 | 0.445 | 0.573 | 0.850 | 0.871 | 0.846 | 0.584 | 0.608 | 0.648 | 0.806 | 0.575 | 0.815 |
| 'Stingray' | 0.724 | 0.608 | 0.631 | 1.000 | 0.616 | 0.676 | 0.652 | 0.693 | 0.680 | 0.425 | 0.579 | 0.669 | 0.655 | 0.651 | 0.644 | 0.635 | 0.606 | 0.674 | 0.617 | 0.680 |
| 'Hen' | 0.659 | 0.760 | 0.654 | 0.616 | 1.000 | 0.693 | 0.798 | 0.807 | 0.858 | 0.780 | 0.927 | 0.794 | 0.805 | 0.875 | 0.613 | 0.682 | 0.858 | 0.713 | 0.589 | 0.886 |
| 'Ostrich' | 0.766 | 0.719 | 0.714 | 0.676 | 0.693 | 1.000 | 0.626 | 0.736 | 0.742 | 0.425 | 0.614 | 0.752 | 0.665 | 0.734 | 0.646 | 0.689 | 0.671 | 0.742 | 0.621 | 0.696 |
| 'Goldfinch' | 0.746 | 0.848 | 0.811 | 0.652 | 0.798 | 0.626 | 1.000 | 0.813 | 0.778 | 0.685 | 0.772 | 0.778 | 0.916 | 0.917 | 0.677 | 0.598 | 0.767 | 0.718 | 0.564 | 0.919 |
| 'Junco' | 0.750 | 0.887 | 0.878 | 0.693 | 0.807 | 0.736 | 0.813 | 1.000 | 0.774 | 0.542 | 0.719 | 0.929 | 0.873 | 0.913 | 0.590 | 0.716 | 0.726 | 0.859 | 0.562 | 0.893 |
| 'Bald-Eagle' | 0.697 | 0.744 | 0.657 | 0.680 | 0.858 | 0.742 | 0.778 | 0.774 | 1.000 | 0.669 | 0.832 | 0.765 | 0.794 | 0.821 | 0.639 | 0.690 | 0.805 | 0.742 | 0.665 | 0.856 |
| 'Vulture' | 0.424 | 0.538 | 0.445 | 0.425 | 0.780 | 0.425 | 0.685 | 0.542 | 0.669 | 1.000 | 0.815 | 0.462 | 0.686 | 0.690 | 0.360 | 0.389 | 0.723 | 0.502 | 0.371 | 0.718 |
| 'Newt' | 0.589 | 0.719 | 0.573 | 0.579 | 0.927 | 0.614 | 0.772 | 0.719 | 0.832 | 0.815 | 1.000 | 0.721 | 0.762 | 0.830 | 0.609 | 0.623 | 0.835 | 0.655 | 0.557 | 0.844 |
| 'Axolotl' | 0.734 | 0.901 | 0.850 | 0.669 | 0.794 | 0.752 | 0.778 | 0.929 | 0.765 | 0.462 | 0.721 | 1.000 | 0.822 | 0.879 | 0.665 | 0.745 | 0.713 | 0.866 | 0.597 | 0.864 |
| 'Tree Frog' | 0.714 | 0.904 | 0.871 | 0.655 | 0.805 | 0.665 | 0.916 | 0.873 | 0.794 | 0.686 | 0.762 | 0.822 | 1.000 | 0.922 | 0.584 | 0.623 | 0.786 | 0.753 | 0.554 | 0.926 |
| 'Iguana' | 0.759 | 0.896 | 0.846 | 0.651 | 0.875 | 0.734 | 0.917 | 0.913 | 0.821 | 0.690 | 0.830 | 0.879 | 0.922 | 1.000 | 0.657 | 0.671 | 0.804 | 0.815 | 0.572 | 0.941 |
| 'African Chameleon' | 0.724 | 0.654 | 0.584 | 0.644 | 0.613 | 0.646 | 0.677 | 0.590 | 0.639 | 0.360 | 0.609 | 0.665 | 0.584 | 0.657 | 1.000 | 0.629 | 0.594 | 0.585 | 0.648 | 0.645 |
| 'Cobra' | 0.682 | 0.668 | 0.608 | 0.635 | 0.682 | 0.689 | 0.598 | 0.716 | 0.690 | 0.389 | 0.623 | 0.745 | 0.623 | 0.671 | 0.629 | 1.001 | 0.651 | 0.709 | 0.674 | 0.692 |
| 'Scorpion' | 0.595 | 0.751 | 0.648 | 0.606 | 0.858 | 0.671 | 0.767 | 0.726 | 0.805 | 0.723 | 0.835 | 0.713 | 0.786 | 0.804 | 0.594 | 0.651 | 1.000 | 0.664 | 0.665 | 0.826 |
| 'Tarantula' | 0.734 | 0.824 | 0.806 | 0.674 | 0.713 | 0.742 | 0.718 | 0.859 | 0.742 | 0.502 | 0.655 | 0.866 | 0.753 | 0.815 | 0.585 | 0.709 | 0.664 | 1.000 | 0.581 | 0.794 |
| 'Centipede' | 0.565 | 0.577 | 0.575 | 0.617 | 0.589 | 0.621 | 0.564 | 0.562 | 0.665 | 0.371 | 0.557 | 0.597 | 0.554 | 0.572 | 0.648 | 0.674 | 0.665 | 0.581 | 1.000 | 0.619 |
| 'Peacock' | 0.736 | 0.882 | 0.815 | 0.680 | 0.886 | 0.696 | 0.919 | 0.893 | 0.856 | 0.718 | 0.844 | 0.864 | 0.926 | 0.941 | 0.645 | 0.692 | 0.826 | 0.794 | 0.619 | 1.000 |

Table A8: Encoded Language Embedding similarity of the first 20 classes of ImageNet-R, encoded by C:IP encoder as implemented in LGCL.

## D.3 TRADE-OFFS BETWEEN COST AND PERFORMANCE

We extend our analysis on the trade-offs between Cost and Performance. Table A9 shows the cost performance on LEAPGen in comparison to ConvPrompt as the SOTA of the growing component approach and HiDe-Prompt as the SOTA of the task-wise prompt approach. We also compare our

proposed method to the bigger version of ConvPrompt and HiDe-Prompt i.e. ConvPrompt-Large and HiDe-Prompt large that utilize a bigger number of executed parameters than our method. Please note that the number of executed parameters in the table means the total executed parameters in forward operation, excluding ViT backbone and classifiers head.

**(a) Performance vs Parameter and Running Time:** . Our method outperforms the default version of ConvPrompt and HiDe-Prompt with up to 9% average accuracy margin and up to 4% average forgetting measure with the drawback of the higher number of executed parameters. However, with the higher number of executed parameters, our method requires a significantly lower running time than both of the method i.e. 1.5 h vs 2.11h and 5.4h for ConvPrompt and Hide-Promt respectively. Even though utilizing far smaller parameters, Hide-Prompt requires extremely high running time since it executes too many operations i.e. augmenting the prototypes and then forming an uninstructed representation that requires high epochs to converge. Thus, it spends far higher running time in total.

The large version of ConvPrompt and HiDe-Prompt executes a higher number of parameters than LEAPGen. In line to the higher parameters, they require far higher running time than their original version. Unfortunately, with the extremely high number of parameters, they didn't achieve a higher performance, rather lower than the performance of their standard version. Compared to the large version of ConvPrompt and HiDe-Prompt, LEAPGen has a better performance and lower cost. This evidence confirm our sensitivity analysis that a model has the best setting that achieves optimal performance. The higher number of parameters or training epochs doesn't imply a higher performance. Thus we evaluate the competitor methods following their advised setting described on the respective papers or official code.

LEAPGen-lite outperforms ConvPrompt and Hide-Prompt with a significant margin i.e. 6-8% even though with utilizing a smallest number of parameters on average and without class descriptors as in ConvPrompt and LEAPGen. However, it achieves smaller final (task-10) average accuracy than LEAPGen with more than 1.5% margin.

**(b). Total Running Time:** Table A10 shows the simulation time for LEAPGen, ConvPrompt as SOTA in 3 benchmark datasets i.e. CIFAR100, ImageNet-R, and CUB. The table shows a similar trend to the analysis on the ImageNet-R dataset. Descriptor generation consumes a fair running time i.e. 0.53-1.09 h since it is conducted via online query through the internet connection. In such a manner, we don't need extra storage to save the LLM model. Please note that descriptor generation is conducted only once before the training phase. Thus the LLm is not utilized during the training and testing phases of our method. Despite requiring additional time to generate classes' descriptors, our method still has significantly lower total grinning time than HiDe-Prompt i.e. 3 times, 3 times, and 2.8 times smaller in CIFAR100, ImageNet-R and, and CUB datasets respectively. Compared to our method, ConvPrompt requires a higher training time in those 3 datasets with a significant gap i.e. up to 7.7 hours. Please note that ConvPrompt utilizes evolving generators and the number of its generators grows with the increasing of tasks. Thus in the later tasks, its forward operation takes more time than in the earlier tasks. Having no additional time for descriptor generation LEAPGen-lite spends the smallest total running time and training+inference time.

**(c). Inference time only:** Table A10 shows that our methods i.e. LEAPGen and LEAPGen-lite hae a moderate inference time i.e. lower than ConvPrompt and higher than HiDe-Prompt. This empirical evidence is in line with our previous analysis where growing components/generators approach executes more generators in its forward function with the increasing number of tasks. Thus, on average, it takes more inference time than our task-wise generator approach. Hide-Prompt requires less inference time since it utilizes task-wise prompting where the prompts are not generated, but rather directly prepended into the ViT MSA heads. Thus it takes less operation to execute which implies its lower inference time. In spite of its lower inference time, HiDe-Prompt requires far higher training time which makes it a higher total running time than our method. In summary, this evidence proves the running efficiency of our proposed method both in terms of inference time or the total running time in a simulation, in spite that our method executes a higher number of parameters.

**(d). Performance vs Storage:** Table A11 shows the performance vs storage of the consolidated methods in the ImageNet-R dataset with 10 task settings. Model saving is conducted by executing the method "torch.save(model.state_dict().filename)". Please note that this storage doesn't include the LLM model, since it is used via online (remote) query, not saved in our storage. The table shows that LEAPGen requires a bigger storage size than ConvPrompt, as LEAPGen has more at-

tributes and parameters. Please note that the model saving is represented in a dictionary object, thus, more attributes require more dictionary keys and items that imply more storage consumption. Different from ConvPrompt and HiDe-Prompt, LEAPGen has a more complex generator structure and task-id predictors that utilize task-wise key and class-wise keys. In addition, LEAPGen also saves the descriptors embedding for the testing phase, different from ConvPrompt which doesn't save the embeddings as it uses the descriptors embedding to calculate the number of generators during the training phase only. Therefore it requires a bigger storage to save its model. The storage consumption can be minimized by using a lightweight generator. In comparison to large versions of ConvPrompt and HiDe-Prompt that have a higher number of parameters, our method achieves a lower storage consumption while achieving a higher performance i.e. FAA and CAA.

| Method | Metrics | Performance After Finishing Task | | | | | | | | | | AVG | Running Time (h) |
|--------|---------|------|------|------|------|------|------|------|------|------|------|-----|------------------|
| | | 1 | 2 | 3 | 4 | 5 | 6 | 7 | 8 | 9 | 10 | | |
| HiDe-Prompt | Avg.Accuracy | 85.22 | 82.93 | 82.35 | 79.74 | 78.85 | 77.81 | 77.37 | 76.38 | 76.31 | 75.75 | 79.27 | - |
| HiDe-Prompt-Large | Avg.Accuracy | 85.76 | 83.12 | 82.03 | 79.49 | 78.55 | 76.55 | 76.00 | 74.82 | 74.98 | 74.30 | 78.56 | - |
| ConvPrompt | Avg.Accuracy | 89.53 | 86.24 | 84.40 | 81.88 | 80.79 | 80.04 | 78.80 | 78.15 | 77.74 | 77.08 | 81.47 | - |
| ConvPrompt-Large | Avg.Accuracy | 90.12 | 86.22 | 83.96 | 81.45 | 80.52 | 79.73 | 76.40 | 74.19 | 74.56 | - | 80.79 | - |
| LEAPGen-lite | Avg.Accuracy | 89.73 | 88.66 | 86.94 | 86.43 | 86.04 | 83.03 | 82.75 | 82.52 | 82.94 | 82.38 | 85.14 | - |
| LEAPGen | Avg.Accuracy | 89.83 | 87.63 | 86.46 | 85.97 | 84.71 | 84.79 | 84.01 | 83.77 | 84.12 | 84.09 | 85.54 | - |
| HiDe-Prompt | Avg.Forgetting | 3.29 | 1.50 | 2.41 | 2.19 | 2.53 | 2.20 | 2.25 | 2.30 | 2.29 | 2.33 | 2.33 | - |
| HiDe-Prompt-Large | Avg.Forgetting | - | 3.05 | 1.89 | 2.14 | 2.15 | 3.25 | 2.91 | 2.93 | 2.66 | 2.70 | 2.63 | - |
| ConvPrompt | Avg.Forgetting | - | 2.57 | 1.57 | 2.60 | 2.93 | 3.15 | 3.53 | 3.44 | 4.07 | 4.17 | 3.11 | - |
| ConvPrompt-Large | Avg.Forgetting | - | 4.65 | 3.39 | 4.62 | 4.69 | 5.56 | 8.91 | 10.55 | 10.50 | - | 6.61 | - |
| LEAPGen-lite | Avg.Forgetting | 1.11 | 0.75 | 0.58 | 0.56 | 4.13 | 3.37 | 2.92 | 2.70 | 3.01 | 2.13 | 2.13 | - |
| LEAPGen | Avg.Forgetting | - | 3.97 | 1.96 | 1.38 | 2.63 | 2.24 | 1.99 | 1.77 | 1.60 | 1.46 | 2.11 | - |
| HiDe-Prompt | #Parameters (M) | 0.15 | 0.15 | 0.15 | 0.15 | 0.15 | 0.15 | 0.15 | 0.15 | 0.15 | 0.15 | 0.15 | 5.40 |
| HiDe-Prompt-Large | #Parameters (M) | 12.29 | 12.29 | 12.29 | 12.29 | 12.29 | 12.29 | 12.29 | 12.29 | 12.29 | 12.29 | 12.29 | 15.15 |
| ConvPrompt | #Parameters (M) | 0.55 | 0.60 | 0.70 | 0.80 | 0.85 | 0.89 | 0.99 | 1.09 | 1.19 | 1.28 | 0.89 | 2.11 |
| ConvPrompt-Large | #Parameters (M) | 5.18 | 5.82 | 6.65 | 7.28 | 7.91 | 8.60 | 9.81 | 10.94 | 12.01 | 13.23 | 8.74 | 15.28 |
| LEAPGen-lite | #Parameters (M) | 0.02 | 0.03 | 0.05 | 0.07 | 0.08 | 0.10 | 0.11 | 0.13 | 0.15 | 0.16 | 0.09 | 0.53 |
| LEAPGen | #Parameters (M) | 6.21 | 6.23 | 6.24 | 6.26 | 6.27 | 6.29 | 6.31 | 6.32 | 6.34 | 6.35 | 6.28 | 1.79 |

Table A9: Trade-offs between cost and performance on ImageNet-R dataset with 10 task setting. ConvPrompt-Large is run for 9 task only since it crashed on the final task.

| Method | Running time (h) | | | | | | | | | | | |
|--------|------------------|--|--|--|--|--|--|--|--|--|--|--|
| | CIFAR100 | | | | ImageNet-R | | | | CUB | | | |
| | Descriptor Generation | Inference Only | Training + Inference | Total | Descriptor Generation | Inference Only | Training + Inference | Total | Descriptor Generation | Inference Only | Training + Inference | Total |
| HiDe-Prompt | - | 0.037 | 4.63 | 4.63 | - | 0.019 | 5.40 | 5.40 | - | 0.017 | 3.94 | 3.94 |
| ConvPrompt | 0.53 | 0.023 | 2.01 | 2.54 | 1.07 | 0.033 | 1.04 | 2.11 | 1.09 | 0.035 | 8.08 | 9.17 |
| LEAPGen-lite | - | 0.016 | 1.2 | 1.2 | - | 0.028 | 0.53 | 0.53 | - | 0.028 | 0.38 | 0.38 |
| LEAPGen | 0.53 | 0.017 | 0.98 | 1.51 | 1.07 | 0.025 | 0.72 | 1.79 | 1.09 | 0.025 | 0.32 | 1.41 |

Table A10: Total running time including descriptor generation, training, and testing.

| Method | Performance | | Storage (MB) |
|--------|------|------|--------------|
| | FAA | CFA | |
| HiDe-Prompt | 75.75 | 79.27 | 334 |
| HiDe-Prompt-Large | 74.30 | 78.56 | 797 |
| ConvPrompt | 77.08 | 81.47 | 346 |
| ConvPrompt-Large | 74.56* | 80.79 | 632 |
| LAEPGen-lite | 82.38 | 85.14 | 332 |
| LEAPGen | 84.09 | 85.54 | 567 |

Table A11: Performance vs Storage Trade-offs in ImageNet-R dataset, * denotes measurement at task-9 since the method crashed after task-9

## D.4 CLASS NAME AS LANGUAGE DESCRIPTOR

We extend our investigation of our proposed method in case there is no descriptor generated by GPT or other LLMs. We evaluate LEAPGen performance in a such condition in those 3 datasets as presented in table A12.LEAPGen-CN indicates the variant of LEAPGen that utilizes class names as descriptors embedding. LEAPGen-CN only utilizes a text encoder similar to the majority of language-guided methods such as LGCL. In addition, it doesn't utilize LLM such as GPT as in

ConvPrompt and GMM. Table A12 shows that substituting descriptors with class names doesn't significantly decrease LEAPGen performance i.e. less than 1.1% performance. Despite utilizing class names as descriptors LEAPGen-CN to outperforms existing methods with a significant margin i.e. 5-14% FAA and 3-9% CAA in CIFAR100, 6-20% FAA and 3-17% CAA in ImageNet-R, and 1-21% FAA and 2.8-16% CAA in CUB dataset. This evidence confirms the promising impact of our ideas i.e. task-wise prompt generators and soft task-id prediction. Second, It confirms that utilizing language embedding as input for prompt generator remains effective, rather than utilizing the embedding directly as a guiding prototype or loss anchor.

| Method | CIFAR100 10-Task @10 classes/task | | | | | | ImageNet-R 10-Task @20 classes/task | | | | | | CUB 10-Task @20 classes/task | | | | | |
|---|---|---|---|---|---|---|---|---|---|---|---|---|---|---|---|---|---|---|
| | FAA | CAA | FFM | CFM | F.Gap | C.Gap | FAA | CAA | FFM | CFM | F.Gap | C.Gap | FAA | CAA | FFM | CFM | F.Gap | C.Gap |
| L2P | 83.84 | 88.67 | 6.55 | 5.16 | 14.46 | 9.50 | 62.50 | 67.05 | 5.01 | 4.41 | 20.73 | 17.46 | 66.95 | 74.03 | 5.18 | 7.39 | 21.41 | 16.46 |
| DualPrompt | 85.36 | 89.77 | 5.41 | 4.33 | 12.94 | 8.40 | 68.59 | 72.18 | 4.61 | 3.70 | 14.64 | 12.33 | 73.95 | 80.29 | 7.87 | 8.60 | 14.41 | 10.20 |
| CODA-P | 86.44 | 91.27 | 6.38 | 5.09 | 11.86 | 6.90 | 73.77 | 79.38 | 7.94 | 6.72 | 9.46 | 5.13 | 72.99 | 82.64 | 11.71 | 9.91 | 15.37 | 7.85 |
| LGCL | 85.68 | 90.16 | 5.46 | 4.25 | 12.62 | 8.01 | 68.65 | 72.57 | 4.75 | 3.38 | 14.58 | 11.94 | 79.93 | 83.07 | 5.45 | 5.58 | 8.43 | 7.42 |
| HiDe-Prompt | 92.89 | 95.01 | 1.98 | 1.56 | 5.41 | 3.16 | 75.75 | 79.27 | 2.29 | 2.33 | 7.48 | 5.24 | 87.21 | 87.66 | 1.90 | 2.66 | 1.15 | 2.83 |
| PGP | 86.36 | 90.83 | 5.49 | 4.28 | 11.94 | 7.34 | 68.62 | 72.19 | 4.53 | 3.63 | 14.61 | 12.32 | 78.35 | 82.21 | 5.76 | 6.10 | 10.01 | 8.28 |
| EvoPrompt | 88.17 | 92.18 | 5.39 | 3.97 | 10.13 | 5.99 | 76.00 | 80.97 | 4.22 | 3.59 | 7.23 | 3.54 | 76.23 | 81.00 | 3.96 | 3.57 | 12.13 | 9.49 |
| CPrompt | 86.92 | 91.73 | 5.43 | 4.01 | 11.38 | 6.44 | 76.32 | 81.50 | 6.10 | 5.60 | 6.91 | 3.01 | 77.14 | 85.67 | 11.65 | 8.69 | 11.22 | 4.82 |
| ConvPrompt | 88.77 | 92.71 | 4.12 | 2.67 | 9.53 | 5.46 | 77.08 | 81.47 | 4.17 | 3.11 | 6.15 | 3.04 | 80.12 | 84.70 | 6.04 | 4.61 | 8.24 | 5.79 |
| LEAPGen | 98.38 | 98.15 | 0.10 | 0.05 | -0.08 | 0.02 | 84.09 | 85.54 | 1.46 | 2.11 | -0.86 | -1.03 | 88.45 | 90.90 | 1.32 | 1.29 | -0.09 | -0.41 |
| **LEAPGen-CN** | **98.30** | **98.17** | **0.14** | **0.14** | **0.00** | **0.00** | **83.23** | **84.51** | **1.55** | **2.79** | **0.00** | **0.00** | **88.36** | **90.49** | **1.31** | **1.23** | **0.00** | **0.00** |

Table A12: Performance of LEAPGen with class name as descriptors compared to existing method and original LEAPGen. F.Gap and C.Gap indicate gap of the respective method to LEAPGen-CN in terms of final average accuracy (FAA) and cumulative average accuracy (CAA).

## D.5 PERFORMANCE ON VARIOUS DESCRIPTORS

We explore 3 types of descriptors i.e. short, long, and narrative generated by popular LLms i.e. GPT, Gemini, and Llama. The examples of descriptors for classes "Strawberry" and "Lemon" for respective types and LLMs are shown in Table A13. The table shows that GPT generates a longer phrase than Gemini and Llama for the same descriptor type. We evaluated the performance of LEAPGen and compared it to ConvPrompt which is the second-best achiever in ImageNet-R dataset and also uses descriptors. Note that ConvPrompt utilizes descriptors to define the number of new generators, while LEAPGen utilizes them as input to generate prompt components.

Table A14 shows the summarized result of LEAPGen and ConvPrompt performance on various descriptors. Overall, In any descriptors type generated by any LLMs, LEAPGen achieves at least 83.6% FAA and 85.0% CAA, where ConvPrompt achieve at maximum 77.6% FAA and 81.5% CAA. This implies than regardless any type of descriptors generated by any LLM, LEAPGen consistently outperforms ConvPrompt with at least 6% and 3.5% margin for FAA and CAA respectively. Looking at the average forgetting metrics, LEAPGEn also consistently outperforms ConvPrompt with least 3% and 1% margin for FFM and CFM respectively. Figure A5 shows the historical performance of LEAPGEn and ConvPrompt in various types of detectors and LLMs. The figure shows that despite LEAPGen achieving slightly lower average accuracy in the first task, It consistently achieves higher accuracy in the second until the last task. In addition, the gap is getting higher with the increasing number of learned tasks. The figure also shows that despite LEAPGen achieving a slightly higher forgetting in the second task, It manages to decrease the forgetting measure in the following task, until the last task. On the other hand, ConvPrompt suffers from the increasing average forgetting so its average forgetting becomes far higher than LEAPGen's average forgetting in the later tasks.

Table A14 shows that the best result is achieved by LEAPGen with short descriptors generated by LLama. However, LEAPGen with short descriptors generated by Gemini achieves the least performance. Surprisingly, short descriptors generated by Llama are the shortest descriptors among all the evaluated types. This result gives us insight that the length of descriptors is not the main factor in achieving maximum performance, rather the discriminative text has more impact. According to table A14, we also get insight that GPT produces more stable descriptors than the other LLSm, as the performance difference between types of descriptors is insignificant ($< 0.15\%$). Thus, regardless type of preferred descriptors, GPT is preferrable than the other LLMs.

| LLM | Type of Descriptors | | |
|---|---|---|---|
| | Short | Long | Narrative |
| GPT-4 | "Strawberry":
[
"red, heart-shaped fruit",
"small, green leaves",
"white seeds on the surface",
"yellow or white center",
"pointed tip",
"juicy texture"
]
"Lemon":
[
"yellow or greenish-yellow
in color",
"oval or round shape",
smooth, glossy skin",
"dimpled surface",
"acidic smell",
"sour taste",
"thick, white pith",
"seeds inside"
] | "Strawberry":
[
"Red, heart-shaped fruit",
"Small, green leafy crown",
"Juicy and sweet",
"Tiny seeds on surface",
"Bright red color",
"Soft, tender texture",
"Used in desserts and salads",
"Often enjoyed fresh or
in preserves",
"Distinctive aroma",
"Popular in summer"
]
"Lemon":
[
"Bright yellow, oval fruit",
"Tangy, acidic flavor",
"Thick, dimpled skin",
"Juicy, citrusy interior",
"Used for flavoring and zest",
"Common in cooking
and beverages",
"Refreshing and tart",
"Variety of sizes",
"Often used in garnishes",
"Distinctive, vibrant color"
] | "Strawberry":
[
"A strawberry rests temptingly,
its red surface dotted with
tiny seeds hinting at juiciness.",
"The green leaves crown the top,
contrasting beautifully with
its vibrant hue.",
"With a sweet spirit, it embodies
the essence of summer."
]
"Lemon":
[
"A lemon sits cheerfully, its bright
yellow skin and dimpled texture
promising a burst of citrus flavor.",
"The zesty aroma and vibrant color
invite culinary exploration.",
"With a tangy spirit, it embodies
the essence of refreshment."
] |
| Gemini-1.5 | "Strawberry":
[
"red fruit with seeds on
the outside",
"red color",
"small size and
Sweet taste"
]
"Lemon":
[
"yellow citrus fruit",
"yellow skin",
"sour taste"
] | "Strawberry":
[
"red fruit with seeds on
The outside",
"red color",
"small size and sweet taste",
"fruit",
"red color",
"small size and sweet taste",
"used in desserts and jams"
]
"Lemon":
[
"yellow citrus fruit",
"yellow skin",
"sour taste",
"citrus fruit",
"yellow skin",
"sour taste",
"used in drinks and cooking"
] | Strawberry":
[
"A small, red fruit with seeds
on the outside.
It is a popular fruit and is often
used in jams, jellies, and desserts."
]
"Lemon":
[
"A yellow citrus fruit with a
sour taste.
It is often used in drinks,
cooking, and baking."
] |
| Llama-3.0 | "Strawberry":
[
"fruit",
"red color",
"sweet"
]
"Lemon":
[
"fruit",
"yellow skin",
"sour"
] | "Strawberry":
[
"fruit",
"sweet",
"juicy",
"red color",
"summer",
"shortcake",
"vitamin-rich",
"antioxidant",
"delicious",
"romantic"
]
"Lemon":
[
"fruit",
"citrus",
"sour taste",
"yellow skin",
"juicing",
"cooking",
"flavorful",
"refreshing",
"zesty",
"invigorating"
] | "Strawberry":
[
"Sweet fruit delights,
luscious landscape indulging,
happy heart enjoying like a
Joyful, little treat."
]
"Lemon":
[
"Zesty slice refreshes,
citrusy landscape revitalizing,
Rejuvenated heart refreshing
Like a bright, little sunrise."
] |

Table A13: Types and examples of descriptors generated by popular LLMs.

| LLM | Desc. Type | LEAPGen | | | | ConvPrompt | | | |
|---|---|---|---|---|---|---|---|---|---|
| | | FAA | CAA | FFM | CFM | FAA | CAA | FFM | CFM |
| GPT-4 | Short | 84.70 | 86.36 | 0.70 | 1.74 | 77.08 | 81.47 | 4.17 | 2.98 |
| GPT-4 | Long | 84.73 | 86.14 | 0.91 | 1.48 | 77.03 | 81.40 | 3.95 | 3.28 |
| GPT-4 | Narrative | 84.60 | 85.72 | 0.84 | 1.66 | 77.33 | 81.53 | 3.95 | 3.28 |
| Gemini-1.5 | Short | 83.67 | 85.07 | 0.96 | 1.45 | 77.27 | 81.33 | 3.89 | 3.43 |
| Gemini-1.5 | Long | 84.65 | 85.69 | 0.81 | 1.74 | 77.18 | 81.37 | 4.24 | 3.52 |
| Gemini-1.5 | Narrative | 84.63 | 85.68 | 0.53 | 1.28 | 77.22 | 81.37 | 3.96 | 3.40 |
| Llama-3.0 | Short | 85.22 | 86.35 | 0.79 | 1.54 | 77.25 | 81.37 | 3.94 | 3.40 |
| Llama-3.0 | Long | 84.72 | 86.24 | 0.77 | 1.49 | 77.15 | 81.45 | 3.92 | 3.29 |
| Llama-3.0 | Narrative | 83.72 | 85.33 | 0.80 | 1.49 | 77.59 | 81.49 | 3.93 | 3.27 |

Table A14: Performance of LEAPGEn vs ConvPrompt in various descriptors

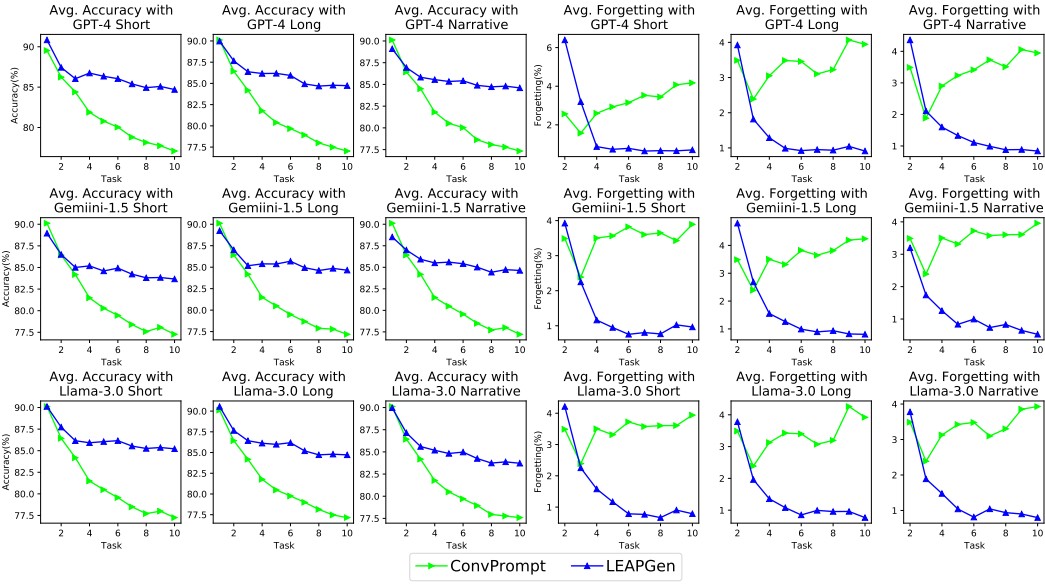

Figure A5: Visualization of the historical performance of consolidated algorithms in ImageNet-R with 10 tasks setting, in various descriptors and LLM

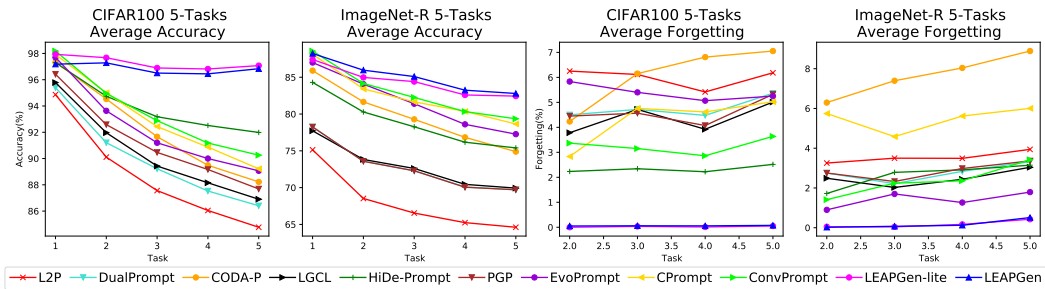

Figure A7: Visualization of the historical performance of consolidated algorithms in CIFAR100 and ImageNet-R with 5 tasks setting

## D.6 PERFORMANCE ON VARIOUS TYPES OF DESCRIPTORS

LEAPGen's default generator $(G^{tK}_{i,l,h}, G^{tV}_{i,l,h})$ is a single-head self-attention (SSA) network parameterized by 3 matrices as explained in section sub-4.d. However, our method accommodates the different types of generators e.g. MLP (3-layered), LSTM, and GRU. Figure A6 shows that LAPGen with any type generator achieves higher performance than without generators (None). The proposed SSA generator achieves fairly higher performance than MLP, slightly higher than GRU and comparable to

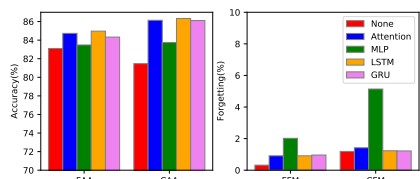

Figure A6: FAA, CAA, FFM, and CFM of the proposed method w.r.t types of generator.

LSTM. However, LSTM and GRU has approximately 2.5 and 2 times larger number of parameters than SSA, while MLP has the same number of parameters as SSA.

## D.7 HISTORICAL PERFORMANCE ON 5 AND 20-TASK SETTINGS

We continue our analysis on the historical performance of consolidated methods in CIFAR100 and ImageNet-R with 5 and 20-task settings. Figures A7 and A8 plot the historical performance of the consolidated methods. The figures show that both in 5-task and 20-task settings, both in CIFAR100 and ImageNet-R datasets, LEAPGen consistently outperforms the existing methods. Figure A7 shows that in the 5-task setting, LEAPGen manages to maintain its performance in CIFAR100 in all tasks, and suffers from less than 4% accuracy degradation in Imagent-R. The figure also shows that our method experiences less than 1% forgetting in all tasks in both datasets.

Figure A8 shows that in the 20-task setting, LEAPGen suffers from less than 2% accuracy drop CIFAR100, while the existing methods suffer from up to 17% accuracy drop. In the Imagenet-R dataset, our methods experience less than 7% accuracy drop CIFAR100, while the existing methods suffer from up to 25% accuracy degradation. This is a remarkable achievement pf our proposed method since the larger task number implies a more challenging setting. In a such difficult setting, our method can maintain its performance so that experiences the least degradation among the evaluated methods. The figure also shows that our method experiences nearly zero forgetting in the first ten tasks of CIFAR100, and experiences less than 2% forgetting afterward. Our method also manages to decrease its forgetting measure in the latest tasks on ImageNet-R datasets. On the other hand, the existing methods suffer from increasing forgetting both in CIFAR100 and Imagenet-R so its magnitude gets higher i.e. up to 9% and 11% in CIFAF100 and ImageNet-R respectively.

This historical analysis emphasizes the robustness of our method in various number of task settings and confirms its superiority to the existing SOTAs.

## E DETAILED EXPERIMENTAL SETTING

**Existing Methods:** The existing SOTAs are run by executing the official implementation (code) of the respective methods. The hyper-parameters setting is chosen based on the official setting. HiDe-Prompt utilizes S-Prompt(Wang et al., 2022a) i.e. similar to DualPrompt but without the global (task-shared) prompt. LGCL and PGP don't propose a specific prompt structure but rather utilize L2P and DualPrompt. The reported results for PGP and LGCL are obtained with DualPrompt

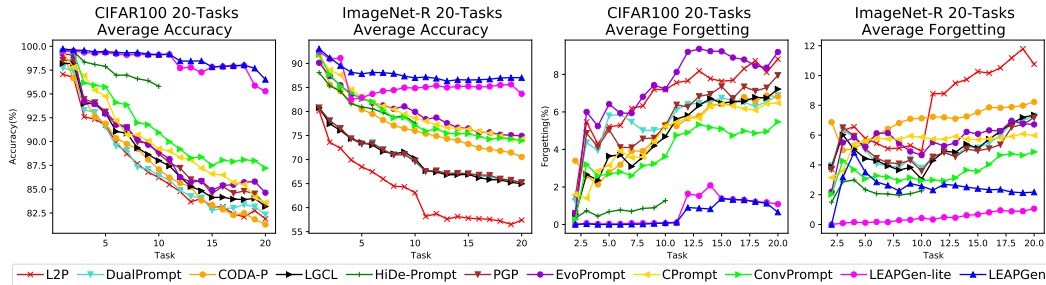

Figure A8: Visualization of the historical performance of consolidated algorithms in CIFAR100 and ImageNet-R with 20 tasks setting

structure which is the best from their result. The other methods i.e. L2P, DualPrompt, CODA-P, EvoPrompt, CPrompt, and ConvPrompt utilize their proposed structure. All the evaluated methods utilize ViT B/16 pre-trained on ImageNet 21K as the backbone model. LGCL utilizes pre-trained CLIP text encoder, while ConvPrompt utilizes SentenceTransformer pre-trained on BERT as its text encoder.

**LEAPGen:** Our proposed method is implemented on top of the ViT backbone pre-trained on ImageNet-21K. The prompt structure is as defined in section 4. The prompt length is set to 30, and the prefix tuning layers is set to 7 i.e. [0,1,2,3,4,5,6] for all main experiments. We utilize Adam optimizer with a cosine learning rate scheduler. For CIFAR100 dataset, We set 0.01 initial learning rate and 3, 10, and 10 epochs for 5-task, 10-task, and 20-task settings respectively. For ImageNet-R dataset, We choose 5, 10, and 20 epochs for 5-task, 10-task, and 20-task settings respectively. The initial learning rate is chosen from the best of 0.04,0.05,0.06. For CUB dataset, We choose 20 epochs and 0.005 initial learning rate. Our method accommodates a flat learning rate (not decayed) for $K^t$ in a different optimizer; the soft task-id predictor is computed in a batch-wise manner for convenience. Similar to ConvPrompt, we utilize SentenceTransformer as the text encoder. The pre-trained models i.e. ViT, SentenceTransformer and CLIP(LGCL) parameters are frozen (not fine-tuned) following ConvPrompt (Roy et al., 2024) implementation.

The $\lambda_2$ and $\lambda_3$ are set to 1.0, while $\lambda_1$ is set to 1.0 for CIFAR100, and 0.1 for CUB and ImageNet-R. The number of generators per task that is the same as top-$k$ descriptors is set to 3. All the consolidated methods are run under the same machine and computing environment i.e. single NVIDIA A100 GPU with 40 GB memory, python 3.8 and Pytorch 2.2.0.

**Performance Metrics:** Adapted from HidePrompt, we measure both accuracy and forgetting of the methods. Suppose that $A_{i,t}$ denotes the accuracy on the $t$-th task after learning the $t$-th task. The average accuracy of all learned task is defined as $AA_t = (1/t)\Sigma_{i=1}^t A_{i,t}$. Suppose that $T$ is the number of all tasks, we measure final average accuracy (FAA), cumulative average accuracy(CAA), final forgetting measure (FFM), and cumulative forgetting measure (CFM) as defined in the equations below.

$$FAA = AA_T \tag{A56}$$

$$CAA = \frac{1}{T}\Sigma_{t=1}^T AA_t \tag{A57}$$

$$FFM = \frac{1}{T-1}\Sigma_{i=1}^{T-1} max_{t\in 1,...,T-1}(AA_{i,t} - AA_{i,T}) \tag{A58}$$

$$CFM = \frac{1}{T-1}\frac{1}{T-1}\Sigma_{j=2}^T\Sigma_{i=1}^{j-1} max_{t\in 1,...,j-1}(AA_{i,t} - AA_{i,j}) \tag{A59}$$

## F EXTENDED LITERATURE REVIEW

Rehearsal-based CL e.g. ICARL (Rebuffi et al., 2017), GD (Prabhu et al., 2020), XDER(Boschini et al., 2022) that saves exemplars from the previous tasks and replays them along with current task samples. The methods tune the whole backbone e.g. ResNet in each to adapt to learn new knowledge. The forgetting to previously learned tasks is minimized via replaying saved exemplars and distillation between the previous (saved) model and the current tuned model. The bias correction approach e.g. BiC (Wu et al., 2019) and LUCIR (Hou et al., 2019) adapt to a new task by adding

a task-wise bias layer. The bis layer is tuned to learn a new task with a minimum change of model weight. Therefore It minimizes the forgetting on the previous tasks and achieves stability-plasticity. The bisa correction approach also utilizes memory replay to minimize forgetting. Regularization approach e.g. EWC (Kirkpatrick et al., 2017), MAS (Aljundi et al., 2018), and DMC (Zhang et al., 2020) that tunes the learner parameters to accommodate the previous and current tasks. The regularization can be implemented as a function or a saved gradient to avoid forgetting the previously learned tasks. The knowledge distillation can be applied in the regularization approach using a non-memory exemplar e.g. openly accessed dataset that is not part of learned task samples.

The prompt-based approach proposes a breakthrough solution for Class Incremental Learning (CIL) by training tiny task-aware parameters called **prompts** on top of frozen pre-trained feature extractor e.g. visual transformer (ViT)(Dosovitskiy et al., 2020) that contains far bigger parameters. ViT is a sequence of $L$-layered model $f_\theta(.)$ where a layer-$l$ ($l \in [1..L]$) contains $H$ Multi Head Self Attention (MSA) combined with Multi-Layer Perceptron (MLP) block, Norm, addition and residual connections. Given an input patched embedding $x_{e_{l,h}}$, the $h^{th}$ head MSA of layer-$l$, parameterized by key-query-value matrices i.e. $W_{l,h}^Q, W_{l,h}^K, W_{l,h}^V$ performs self-attention function as defined in equation below:

$$A(Q_{l,h}, K_{l,h}, V_{l,h}) = Softmax((Q_{l,h}K_{l,h}^T)/\sqrt{D_h})V_{l,h} \tag{A60}$$

where $Q_{l,h} = x_{e_{l,h}}W_{l,h}^Q, K_{l,h} = x_{e_{l,h}}W_{l,h}^K, V_{l,h} = x_{e_{l,h}}W_{l,h}^V$ are query, key, and value and $D_h$ denotes their dimension. Prompt-based CL attaches a small set of trainable parameters called prompts into a frozen pre-trained ViT model (fixed $\psi$) (Wang et al., 2022c;b). The attachment techniques are mainly divided into 2 types i.e. prompt tuning (Lester et al., 2021) and prefix tuning (Li & Liang, 2021). Prompt tuning prepends a prompt vector $P \in \mathbb{R}^{L_P \times D}$ into query, key, and value of MSA. Suppose the prompt is divided into head-wise parts i.e. $P = [P_1, P_2, ...P_H]$, then $P_h \in \mathbb{R}^{L_P \times D/H}$ is prepended into $h^{th}$ head of $l^{th}$ layer, and the attention mechanism is computed as $A([P_h : Q_{l,h}], [P_h : K_{l,h}], [P_h : V_{l,h}])$. The symbols $L_P$, $D$, and ":" denote prompt length, embedding dimension and concatenation operator respectively. Prefix tuning as utilized in DualPrompt (Wang et al., 2022b) divides $P_h$ into pair of $P_h^K$ and $P_h^V$ for key and value respectively, then the attention mechanism becomes $A(Q_{l,h}, [P_h^K : K_{l,h}], [P_h^V : V_{l,h}])$.

Based on the prompt structure point of view, we can categorize the prompt-based approach into several categories i.e. (1)Pool-based approach e.g. L2P(Wang et al., 2022c) that uses a pool as prompt container and select a prompt based on input-similarity, (2) Task-specific approach e.g. DualPrompt(Wang et al., 2022b), SPrompt(Wang et al., 2022a), HiDE-Prompt(Wang et al., 2024a) and CPrompt (Gao et al., 2024) that train specific prompt for a specific task and uses task identifier in the inference process, (3) Growing component approach e.g. CODA-P(Smith et al., 2023b), EvoPrompt(Kurniawan et al., 2024), and ConvPrompt(Roy et al., 2024) that increase prompt components to adapt with a new task, instead of the task identifier. DualPrompt, and CPrompt utilize learnable keys as task predictor, while HiDe-Prompt utilizes unstructured representation (clusters) as the task predictor. CODA-P adds new hand-crafted (fixed size) learnable parameters as new prompt components, while EvoPrompt and ConvPrompt generate the new prompt components using newly added generators. Therefore, EvoPrompt and ConvPrompt can be considered as growing generators approach.

## G   COMPLETE NUMERICAL RESULTS

In this section, we present the complete numerical results on CIFAR100, ImageNet-R and CUB datasets in all settings, as presented in table A15 to A21. Color blue and red denote the highest and runner-up performance respectively. Table A15 to A21 show that our proposed method almost achieves the highest average accuracy and average forgetting in all cases.

| Method | Task | | | | | Avg. | Task | | | | | Avg. |
|---|---|---|---|---|---|---|---|---|---|---|---|---|
| | 1 | 2 | 3 | 4 | 5 | | 1 | 2 | 3 | 4 | 5 | |
| | **Average Accuracy (AA)** | | | | | | **Average Forgetting (FM)** | | | | | |
| L2P | 94.87 ± 0.30 | 90.11 ± 0.43 | 87.56 ± 0.33 | 86.05 ± 0.40 | 84.77 ± 0.48 | 88.67 ± 0.30 | - | 6.25 ± 0.54 | 6.12 ± 0.28 | 5.42 ± 0.28 | 6.18 ± 0.57 | 5.99 ± 0.29 |
| DualPrompt | 95.40 ± 0.15 | 91.21 ± 0.14 | 89.22 ± 0.15 | 87.53 ± 0.15 | 86.41 ± 0.21 | 89.95 ± 0.10 | - | 4.50 ± 0.95 | 4.72 ± 0.55 | 4.48 ± 0.29 | 5.37 ± 0.21 | 4.77 ± 0.46 |
| CODA-P | 97.33 ± 0.45 | 94.53 ± 1.79 | 91.66 ± 2.69 | 89.49 ± 0.99 | 88.22 ± 1.06 | 92.25 ± 1.28 | - | 4.23 ± 3.55 | 6.15 ± 3.63 | 6.81 ± 1.50 | 7.05 ± 2.18 | 6.06 ± 2.66 |
| LGCL | 95.78 ± 0.18 | 91.97 ± 0.27 | 89.43 ± 0.03 | 88.15 ± 0.14 | 86.90 ± 0.40 | 90.45 ± 0.18 | - | 3.78 ± 0.33 | 4.72 ± 0.34 | 3.92 ± 0.20 | 5.01 ± 0.35 | 4.36 ± 0.13 |
| HiDe-Prompt | 97.30 ± 0.22 | 94.74 ± 0.08 | 93.18 ± 0.17 | 92.52 ± 0.11 | 91.99 ± 0.03 | 93.95 ± 0.09 | - | 2.23 ± 0.33 | 2.34 ± 0.15 | 2.22 ± 0.07 | 2.52 ± 0.18 | 2.33 ± 0.15 |
| PGP | 96.42 ± 0.35 | 92.58 ± 0.12 | 90.47 ± 0.09 | 89.17 ± 0.08 | 87.69 ± 0.06 | 91.26 ± 0.13 | - | 4.45 ± 0.41 | 4.57 ± 0.25 | 4.07 ± 0.04 | 5.32 ± 0.18 | 4.60 ± 0.15 |
| EvoPrompt | 97.70 ± 0.40 | 93.63 ± 0.25 | 91.20 ± 0.20 | 90.00 ± 0.20 | 89.07 ± 0.38 | 92.32 ± 0.26 | - | 5.83 ± 0.15 | 5.40 ± 0.13 | 5.07 ± 0.12 | 5.25 ± 0.65 | 5.39 ± 0.24 |
| CPrompt | 97.92 ± 0.32 | 95.01 ± 0.71 | 92.43 ± 0.14 | 90.87 ± 0.18 | 89.22 ± 0.05 | 93.09 ± 0.06 | - | 2.83 ± 0.64 | 4.76 ± 0.38 | 4.62 ± 0.20 | 5.02 ± 0.17 | 4.31 ± 0.35 |
| ConvPrompt | 98.18 ± 0.10 | 94.96 ± 0.23 | 92.88 ± 0.23 | 91.18 ± 0.16 | 90.26 ± 0.44 | 93.49 ± 0.19 | - | 3.37 ± 0.15 | 3.15 ± 0.23 | 2.86 ± 0.28 | 3.64 ± 0.28 | 3.25 ± 0.16 |
| **LEAPGen** | 97.18 ± 0.67 | 97.28 ± 0.32 | 96.51 ± 0.14 | 96.45 ± 0.10 | 96.84 ± 0.12 | 96.85 ± 0.26 | - | 0.05 ± 0.05 | 0.06 ± 0.01 | 0.06 ± 0.05 | 0.08 ± 0.07 | 0.06 ± 0.04 |

Table A15: Complete numerical result of the consolidated algorithms in CIFAR100 dataset with 5 tasks setting. Color "blue" and "red" denotes the highest and runner-up performance respectively.

| Method | 1 | 2 | 3 | 4 | 5 | 6 | 7 | 8 | 9 | 10 | Avg. |
|---|---|---|---|---|---|---|---|---|---|---|---|
| | **Average Accuracy (AA)** | | | | | | | | | | |
| L2P | 97.40 ± 0.20 | 93.75 ± 0.30 | 91.21 ± 0.11 | 89.73 ± 0.30 | 88.35 ± 0.24 | 86.78 ± 0.29 | 85.82 ± 0.38 | 85.14 ± 0.30 | 84.64 ± 0.31 | 83.84 ± 0.32 | 88.67 ± 0.16 |
| DualPrompt | 98.13 ± 0.15 | 94.60 ± 0.13 | 92.22 ± 0.13 | 90.55 ± 0.13 | 89.38 ± 0.20 | 87.76 ± 0.44 | 87.04 ± 0.32 | 86.33 ± 0.30 | 86.36 ± 0.20 | 85.36 ± 0.20 | 89.77 ± 0.20 |
| CODA-P | 98.67 ± 0.12 | 95.83 ± 1.25 | 94.12 ± 1.66 | 92.98 ± 2.56 | 91.33 ± 1.18 | 90.01 ± 1.09 | 88.36 ± 0.21 | 87.68 ± 0.60 | 87.27 ± 0.50 | 86.44 ± 0.16 | 91.27 ± 0.56 |
| LGCL | 98.23 ± 0.12 | 94.82 ± 0.42 | 92.99 ± 0.30 | 91.15 ± 0.56 | 89.88 ± 0.54 | 87.94 ± 0.30 | 87.33 ± 0.13 | 86.83 ± 0.14 | 86.71 ± 0.31 | 85.68 ± 0.43 | 90.16 ± 0.29 |
| HiDe-Prompt | 99.17 ± 0.06 | 97.37 ± 0.03 | 96.03 ± 0.15 | 95.57 ± 0.19 | 94.77 ± 0.09 | 93.99 ± 0.09 | 93.62 ± 0.17 | 93.30 ± 0.10 | 93.44 ± 0.16 | 92.89 ± 0.11 | 95.01 ± 0.08 |
| PGP | 98.83 ± 0.29 | 95.32 ± 0.43 | 93.79 ± 0.28 | 91.67 ± 0.29 | 90.64 ± 0.13 | 88.51 ± 0.23 | 88.09 ± 0.27 | 87.69 ± 0.27 | 87.40 ± 0.17 | 86.36 ± 0.19 | 90.83 ± 0.17 |
| EvoPrompt | 99.47 ± 0.06 | 96.37 ± 0.38 | 94.57 ± 0.35 | 93.13 ± 0.55 | 92.23 ± 0.47 | 89.97 ± 0.95 | 89.43 ± 0.57 | 89.20 ± 0.78 | 89.23 ± 0.55 | 88.17 ± 0.51 | 92.18 ± 0.49 |
| CPrompt | 98.20 ± 1.05 | 96.28 ± 0.83 | 94.18 ± 0.43 | 93.34 ± 0.24 | 91.79 ± 1.06 | 90.69 ± 1.04 | 89.49 ± 0.94 | 88.62 ± 0.96 | 87.76 ± 0.60 | 86.92 ± 1.04 | 91.73 ± 0.66 |
| ConvPrompt | 99.60 ± 0.00 | 96.83 ± 0.18 | 95.06 ± 0.12 | 93.78 ± 0.24 | 92.77 ± 0.13 | 90.66 ± 0.16 | 90.18 ± 0.11 | 89.71 ± 0.02 | 89.74 ± 0.26 | 88.77 ± 0.24 | 92.71 ± 0.04 |
| **LEAPGen** | 97.47 ± 1.33 | 98.30 ± 0.53 | 98.18 ± 0.37 | 98.22 ± 0.29 | 98.33 ± 0.36 | 98.16 ± 0.33 | 98.08 ± 0.22 | 98.14 ± 0.21 | 98.24 ± 0.15 | 98.38 ± 0.15 | 98.15 ± 0.39 |
| | **Average Forgetting Measure (FM)** | | | | | | | | | | |
| L2P | - | 3.27 ± 0.40 | 4.13 ± 0.57 | 4.53 ± 0.33 | 5.66 ± 0.32 | 5.34 ± 0.23 | 5.55 ± 0.39 | 5.44 ± 0.19 | 5.95 ± 0.16 | 6.55 ± 0.34 | 5.16 ± 0.14 |
| DualPrompt | - | 3.47 ± 0.23 | 3.60 ± 0.13 | 3.58 ± 0.19 | 4.22 ± 0.28 | 4.62 ± 0.26 | 4.73 ± 0.24 | 4.66 ± 0.21 | 4.70 ± 0.19 | 5.41 ± 0.33 | 4.33 ± 0.15 |
| CODA-P | - | 2.53 ± 1.65 | 3.73 ± 0.65 | 4.33 ± 2.77 | 5.12 ± 2.21 | 5.50 ± 1.36 | 6.11 ± 1.05 | 5.98 ± 0.93 | 6.10 ± 1.03 | 6.38 ± 1.46 | 5.09 ± 1.19 |
| LGCL | - | 3.67 ± 1.15 | 3.05 ± 0.48 | 3.30 ± 0.67 | 4.19 ± 0.36 | 4.66 ± 0.21 | 4.69 ± 0.28 | 4.57 ± 0.31 | 4.64 ± 0.39 | 5.46 ± 0.22 | 4.25 ± 0.32 |
| HiDe-Prompt | - | 0.83 ± 0.38 | 1.13 ± 0.21 | 1.26 ± 0.22 | 1.63 ± 0.25 | 1.91 ± 0.09 | 1.74 ± 0.14 | 1.80 ± 0.14 | 1.75 ± 0.05 | 1.98 ± 0.05 | 1.56 ± 0.15 |
| PGP | - | 3.50 ± 0.92 | 2.77 ± 0.71 | 3.94 ± 0.51 | 4.36 ± 0.23 | 4.77 ± 0.33 | 4.51 ± 0.32 | 4.47 ± 0.23 | 4.72 ± 0.31 | 5.49 ± 0.35 | 4.28 ± 0.27 |
| EvoPrompt | - | 1.17 ± 0.70 | 2.75 ± 0.69 | 3.61 ± 1.08 | 4.06 ± 0.88 | 4.47 ± 1.16 | 4.84 ± 0.70 | 4.73 ± 0.65 | 4.68 ± 0.43 | 5.39 ± 0.45 | 3.97 ± 0.73 |
| CPrompt | - | 1.60 ± 1.25 | 2.60 ± 1.13 | 2.94 ± 1.15 | 3.53 ± 0.49 | 4.59 ± 0.89 | 4.88 ± 0.69 | 5.06 ± 0.55 | 5.48 ± 1.08 | 5.43 ± 0.74 | 4.01 ± 0.81 |
| ConvPrompt | - | 1.17 ± 0.35 | 1.85 ± 0.18 | 1.96 ± 0.31 | 2.66 ± 0.22 | 2.95 ± 0.19 | 3.03 ± 0.03 | 3.04 ± 0.22 | 3.22 ± 0.54 | 4.12 ± 0.44 | 2.67 ± 0.11 |
| **LEAPGen** | - | 0.00 ± 0.00 | 0.05 ± 0.00 | 0.02 ± 0.04 | 0.03 ± 0.04 | 0.04 ± 0.03 | 0.06 ± 0.03 | 0.08 ± 0.02 | 0.09 ± 0.04 | 0.10 ± 0.03 | 0.05 ± 0.00 |

Table A16: Complete numerical result of the consolidated algorithms in CIFAR100 dataset with 10 tasks setting. Color "blue" and "red" denotes the highest and runner-up performance respectively.

| | | | | | Average Accuracy (AA) | | | | | | |
|---|---|---|---|---|---|---|---|---|---|---|---|
| Method | 1 | 2 | 3 | 4 | 5 | 6 | 7 | 8 | 9 | 10 | Avg. |
| L2P | 97.07 ± 0.12 | 96.73 ± 0.15 | 92.60 ± 0.29 | 92.35 ± 0.09 | 91.73 ± 0.46 | 89.89 ± 0.42 | 88.71 ± 0.20 | 87.67 ± 0.32 | 86.79 ± 0.42 | 86.22 ± 0.47 | - |
| DualPrompt | 97.73 ± 0.31 | 97.40 ± 0.53 | 93.31 ± 0.67 | 93.08 ± 0.58 | 91.55 ± 0.40 | 89.52 ± 0.35 | 89.10 ± 0.20 | 87.32 ± 0.17 | 87.10 ± 0.17 | 86.45 ± 0.16 | - |
| CODA-P | 99.13 ± 0.12 | 96.67 ± 1.48 | 94.47 ± 2.40 | 92.60 ± 1.75 | 91.88 ± 1.39 | 90.34 ± 1.54 | 89.75 ± 3.19 | 89.23 ± 2.62 | 88.11 ± 0.92 | 87.08 ± 0.51 | - |
| LGCL | 98.20 ± 0.00 | 98.20 ± 0.10 | 93.98 ± 0.10 | 94.00 ± 0.36 | 93.01 ± 0.22 | 91.06 ± 0.18 | 90.88 ± 0.29 | 89.33 ± 0.34 | 88.64 ± 0.40 | 87.99 ± 0.26 | - |
| HiDe-Prompt | 99.60 ± 0.20 | 99.50 ± 0.10 | 98.36 ± 0.20 | 98.10 ± 0.05 | 97.89 ± 0.16 | 96.97 ± 0.20 | 96.97 ± 0.20 | 96.59 ± 0.16 | 96.39 ± 0.22 | 95.80 ± 0.14 | - |
| PGP | 98.53 ± 0.23 | 98.57 ± 0.40 | 94.42 ± 0.14 | 94.10 ± 0.18 | 93.13 ± 0.12 | 91.94 ± 0.13 | 91.49 ± 0.10 | 89.94 ± 0.63 | 89.87 ± 0.39 | 88.71 ± 0.04 | - |
| EvoPrompt | 99.20 ± 0.00 | 99.10 ± 0.26 | 94.20 ± 1.35 | 93.93 ± 0.81 | 92.93 ± 0.47 | 91.77 ± 0.40 | 91.47 ± 0.60 | 90.23 ± 0.47 | 89.67 ± 0.55 | 88.90 ± 0.35 | - |
| CPrompt | 99.40 ± 0.00 | 97.80 ± 0.00 | 96.93 ± 0.00 | 95.40 ± 0.00 | 94.76 ± 0.00 | 92.20 ± 0.00 | 90.74 ± 0.00 | 90.72 ± 0.00 | 90.07 ± 0.00 | 89.24 ± 0.00 | - |
| ConvPrompt | 99.47 ± 0.12 | 99.43 ± 0.15 | 96.16 ± 0.17 | 95.92 ± 0.25 | 95.73 ± 0.16 | 94.10 ± 0.26 | 93.86 ± 0.37 | 91.98 ± 0.53 | 91.77 ± 0.61 | 90.95 ± 0.66 | - |
| LEAPGen | 99.73 ± 0.31 | 99.60 ± 0.17 | 99.58 ± 0.08 | 99.42 ± 0.12 | 99.45 ± 0.09 | 99.36 ± 0.07 | 99.31 ± 0.08 | 99.35 ± 0.09 | 99.19 ± 0.07 | 99.14 ± 0.09 | - |
| **Method** | **11** | **12** | **13** | **14** | **15** | **16** | **17** | **18** | **19** | **20** | **Avg.** |
| L2P | 85.50 ± 0.56 | 84.74 ± 0.85 | 83.68 ± 0.74 | 84.01 ± 0.49 | 83.12 ± 0.46 | 83.17 ± 0.50 | 82.40 ± 0.62 | 82.10 ± 0.54 | 82.75 ± 0.33 | 81.89 ± 0.38 | 87.16 ± 0.33 |
| DualPrompt | 85.96 ± 0.22 | 84.89 ± 0.23 | 84.24 ± 0.18 | 84.11 ± 0.18 | 82.78 ± 0.26 | 82.98 ± 0.37 | 83.01 ± 0.23 | 83.37 ± 0.22 | 83.16 ± 0.19 | 82.32 ± 0.22 | 87.47 ± 0.24 |
| CODA-P | 86.21 ± 1.81 | 85.64 ± 1.58 | 85.24 ± 1.53 | 83.82 ± 0.62 | 83.35 ± 0.98 | 82.88 ± 0.87 | 82.29 ± 0.14 | 82.50 ± 0.05 | 81.83 ± 0.34 | 81.29 ± 0.16 | 87.72 ± 0.44 |
| LGCL | 87.44 ± 0.25 | 86.29 ± 0.07 | 85.29 ± 0.54 | 84.83 ± 0.66 | 84.12 ± 0.54 | 84.15 ± 0.50 | 83.96 ± 0.51 | 83.91 ± 0.38 | 84.03 ± 0.51 | 83.18 ± 0.40 | 88.63 ± 0.18 |
| HiDe-Prompt | - | - | - | - | - | - | - | - | - | - | 97.62 ± 0.14 |
| PGP | 87.78 ± 0.45 | 87.22 ± 0.55 | 86.01 ± 0.20 | 85.81 ± 0.13 | 84.55 ± 0.42 | 85.29 ± 0.43 | 84.57 ± 0.41 | 84.80 ± 0.51 | 84.52 ± 0.21 | 83.41 ± 0.35 | 89.23 ± 0.13 |
| EvoPrompt | 88.17 ± 0.15 | 86.27 ± 0.68 | 85.73 ± 0.70 | 85.87 ± 0.80 | 84.90 ± 0.35 | 85.40 ± 0.26 | 85.47 ± 0.23 | 85.73 ± 0.29 | 85.80 ± 0.17 | 84.63 ± 0.21 | 89.47 ± 0.21 |
| CPrompt | 89.02 ± 0.00 | 88.33 ± 0.00 | 88.08 ± 0.00 | 87.20 ± 0.00 | 86.61 ± 0.00 | 86.49 ± 0.00 | 85.72 ± 0.00 | 85.49 ± 0.00 | 84.14 ± 0.00 | 83.60 ± 0.00 | 90.10 ± 0.00 |
| ConvPrompt | 89.99 ± 0.69 | 89.22 ± 0.54 | 88.29 ± 0.66 | 88.38 ± 0.57 | 87.49 ± 0.44 | 88.05 ± 0.49 | 87.90 ± 0.37 | 88.10 ± 0.65 | 87.99 ± 0.47 | 87.21 ± 0.20 | 91.60 ± 0.36 |
| LEAPGen | 99.18 ± 0.10 | 98.46 ± 1.22 | 98.45 ± 1.08 | 98.47 ± 1.02 | 97.83 ± 0.95 | 97.91 ± 0.89 | 97.96 ± 0.81 | 98.03 ± 0.73 | 97.69 ± 1.35 | 96.51 ± 2.16 | 98.73 ± 0.26 |
| | | | | | Average Forgetting Measure (FM) | | | | | | |
| **Method** | **1** | **2** | **3** | **4** | **5** | **6** | **7** | **8** | **9** | **10** | **Avg.** |
| L2P | - | 1.47 ± 0.50 | 5.47 ± 0.93 | 4.29 ± 0.21 | 5.22 ± 0.48 | 5.28 ± 0.56 | 6.19 ± 0.20 | 6.32 ± 0.40 | 7.20 ± 0.41 | 7.19 ± 0.42 | - |
| DualPrompt | - | 1.27 ± 0.23 | 4.37 ± 0.25 | 3.96 ± 0.43 | 5.80 ± 0.18 | 5.87 ± 0.33 | 5.47 ± 0.24 | 5.01 ± 0.21 | 5.03 ± 0.18 | 5.16 ± 0.14 | - |
| CODA-P | - | 3.40 ± 1.40 | 2.60 ± 0.75 | 2.13 ± 0.70 | 2.78 ± 1.14 | 3.19 ± 1.12 | 3.91 ± 1.64 | 3.91 ± 1.59 | 4.46 ± 1.24 | 5.33 ± 1.78 | - |
| LGCL | - | 0.53 ± 0.31 | 2.67 ± 0.61 | 2.36 ± 0.81 | 3.65 ± 0.61 | 3.71 ± 0.66 | 3.09 ± 0.42 | 3.59 ± 0.87 | 4.19 ± 0.78 | 4.69 ± 0.48 | - |
| HiDe-Prompt | - | 0.33 ± 0.12 | 0.73 ± 0.21 | 0.44 ± 0.14 | 0.68 ± 0.13 | 0.77 ± 0.02 | 0.70 ± 0.03 | 0.86 ± 0.00 | 0.89 ± 0.07 | 1.27 ± 0.07 | - |
| PGP | - | 0.60 ± 0.60 | 4.87 ± 0.67 | 4.13 ± 0.70 | 5.05 ± 0.33 | 4.11 ± 0.42 | 4.10 ± 0.38 | 4.57 ± 0.23 | 4.62 ± 0.04 | 5.26 ± 0.19 | - |
| EvoPrompt | - | 0.60 ± 0.20 | 6.00 ± 2.35 | 5.24 ± 1.44 | 6.42 ± 0.85 | 5.93 ± 0.77 | 5.93 ± 1.19 | 6.81 ± 1.09 | 7.43 ± 1.06 | 7.21 ± 0.59 | - |
| CPrompt | - | 1.60 ± 0.00 | 1.40 ± 0.00 | 2.87 ± 0.00 | 3.15 ± 0.00 | 3.92 ± 0.00 | 3.57 ± 0.00 | 3.60 ± 0.00 | 4.43 ± 0.00 | 5.24 ± 0.00 | - |
| ConvPrompt | - | 0.27 ± 0.12 | 3.20 ± 0.79 | 2.60 ± 0.50 | 2.68 ± 0.38 | 2.80 ± 0.29 | 2.60 ± 0.23 | 3.13 ± 0.45 | 3.23 ± 0.43 | 3.62 ± 0.29 | - |
| LEAPGen | - | 0.00 ± 0.00 | 0.03 ± 0.06 | 0.00 ± 0.00 | 0.00 ± 0.00 | 0.00 ± 0.00 | 0.01 ± 0.02 | 0.02 ± 0.03 | 0.05 ± 0.03 | 0.09 ± 0.00 | - |
| **Method** | **11** | **12** | **13** | **14** | **15** | **16** | **17** | **18** | **19** | **20** | **Avg.** |
| L2P | 7.56 ± 0.50 | 7.66 ± 0.55 | 8.18 ± 0.60 | 7.77 ± 0.41 | 7.63 ± 0.25 | 7.69 ± 0.75 | 8.31 ± 0.55 | 8.73 ± 0.31 | 8.11 ± 0.13 | 8.81 ± 0.10 | 6.79 ± 0.33 |
| DualPrompt | 6.12 ± 0.17 | 6.41 ± 0.34 | 6.47 ± 0.22 | 6.52 ± 0.36 | 6.75 ± 0.29 | 6.61 ± 0.51 | 6.51 ± 0.27 | 6.29 ± 0.21 | 6.54 ± 0.23 | 6.88 ± 0.35 | 5.63 ± 0.23 |
| CODA-P | 5.25 ± 1.87 | 5.66 ± 1.49 | 5.79 ± 1.53 | 6.35 ± 1.33 | 6.42 ± 1.14 | 6.40 ± 1.25 | 6.79 ± 1.84 | 6.57 ± 1.59 | 6.87 ± 1.74 | 6.82 ± 1.60 | 4.98 ± 0.95 |
| LGCL | 5.62 ± 0.90 | 5.82 ± 0.51 | 6.34 ± 0.94 | 6.72 ± 1.02 | 6.47 ± 0.87 | 6.55 ± 0.76 | 6.56 ± 0.64 | 6.78 ± 0.43 | 6.74 ± 0.56 | 7.22 ± 0.47 | 4.91 ± 0.56 |
| HiDe-Prompt | - | - | - | - | - | - | - | - | - | - | 0.74 ± 0.03 |
| PGP | 6.38 ± 0.34 | 6.27 ± 0.32 | 6.83 ± 0.12 | 6.95 ± 0.18 | 7.36 ± 0.46 | 6.72 ± 0.48 | 7.35 ± 0.50 | 7.11 ± 0.66 | 7.29 ± 0.19 | 7.95 ± 0.23 | 5.66 ± 0.22 |
| EvoPrompt | 8.13 ± 0.50 | 9.20 ± 0.63 | 9.36 ± 0.69 | 9.24 ± 0.83 | 9.23 ± 0.38 | 8.90 ± 0.45 | 8.79 ± 0.45 | 8.46 ± 0.39 | 8.36 ± 0.33 | 9.19 ± 0.41 | 7.39 ± 0.65 |
| CPrompt | 5.46 ± 0.00 | 5.69 ± 0.00 | 5.65 ± 0.00 | 6.29 ± 0.00 | 6.41 ± 0.00 | 6.32 ± 0.00 | 6.18 ± 0.00 | 6.09 ± 0.00 | 6.44 ± 0.00 | 6.47 ± 0.00 | 4.78 ± 0.00 |
| ConvPrompt | 4.76 ± 0.50 | 4.90 ± 0.58 | 5.33 ± 0.35 | 5.17 ± 0.28 | 5.10 ± 0.22 | 4.74 ± 0.23 | 4.98 ± 0.19 | 4.87 ± 0.48 | 4.96 ± 0.43 | 5.47 ± 0.33 | 3.92 ± 0.33 |
| LEAPGen | 0.11 ± 0.02 | 0.90 ± 1.31 | 0.87 ± 1.20 | 0.83 ± 1.11 | 1.39 ± 0.97 | 1.32 ± 0.90 | 1.30 ± 0.82 | 1.22 ± 0.75 | 1.13 ± 0.75 | 0.66 ± 0.75 | 0.52 ± 0.39 |

Table A17: Complete numerical result of the consolidated algorithms in CIFAR100 dataset with 20 tasks setting. Color "blue" and "red" denotes the highest and runner-up performance respectively.

| Method | Task | | | | | Avg. | Task | | | | | Avg. |
|---|---|---|---|---|---|---|---|---|---|---|---|---|
| | 1 | 2 | 3 | 4 | 5 | | 1 | 2 | 3 | 4 | 5 | |
| | | | Average Accuracy (AA) | | | | | | Average Forgetting (FM) | | | |
| L2P | 75.14 ± 0.53 | 68.52 ± 0.66 | 66.55 ± 0.64 | 65.24 ± 0.49 | 64.62 ± 0.32 | 68.01 ± 0.42 | - | 3.25 ± 0.48 | 3.50 ± 0.19 | 3.49 ± 0.24 | 3.94 ± 0.16 | 3.55 ± 0.20 |
| DualPrompt | 78.25 ± 0.12 | 73.59 ± 0.22 | 72.29 ± 0.27 | 70.09 ± 0.26 | 69.71 ± 0.11 | 72.78 ± 0.14 | - | 2.75 ± 0.62 | 2.22 ± 0.13 | 2.83 ± 0.16 | 3.32 ± 0.16 | 2.78 ± 0.25 |
| CODA-P | 85.90 ± 1.68 | 81.65 ± 2.50 | 79.28 ± 1.96 | 76.83 ± 0.36 | 74.89 ± 0.36 | 79.71 ± 1.27 | - | 6.30 ± 1.52 | 7.39 ± 1.29 | 8.04 ± 0.76 | 8.89 ± 0.65 | 7.65 ± 0.98 |
| LGCL | 77.72 ± 0.47 | 73.83 ± 0.16 | 72.61 ± 0.33 | 70.44 ± 0.19 | 69.93 ± 0.21 | 72.91 ± 0.19 | - | 2.49 ± 0.54 | 2.03 ± 0.49 | 2.44 ± 0.30 | 3.04 ± 0.36 | 2.50 ± 0.38 |
| HiDe-Prompt | 84.30 ± 0.15 | 80.28 ± 0.32 | 78.27 ± 0.16 | 76.17 ± 0.08 | 75.40 ± 0.27 | 78.88 ± 0.04 | - | 1.72 ± 0.31 | 2.79 ± 0.25 | 2.91 ± 0.22 | 3.15 ± 0.46 | 2.64 ± 0.16 |
| PGP | 78.25 ± 0.12 | 73.57 ± 0.09 | 72.29 ± 0.15 | 70.05 ± 0.17 | 69.71 ± 0.15 | 72.77 ± 0.07 | - | 2.75 ± 0.46 | 2.33 ± 0.10 | 2.98 ± 0.22 | 3.36 ± 0.23 | 2.85 ± 0.25 |
| EvoPrompt | 87.00 ± 0.35 | 84.07 ± 0.40 | 81.40 ± 0.20 | 78.60 ± 0.00 | 77.27 ± 0.40 | 81.67 ± 0.18 | - | 0.90 ± 0.20 | 1.70 ± 0.61 | 1.27 ± 0.47 | 1.79 ± 0.31 | 1.41 ± 0.32 |
| CPrompt | 88.12 ± 0.00 | 83.42 ± 0.00 | 81.66 ± 0.00 | 80.36 ± 0.00 | 78.65 ± 0.00 | 82.44 ± 0.00 | - | 5.75 ± 0.00 | 4.58 ± 0.00 | 5.62 ± 0.00 | 6.00 ± 0.00 | 5.49 ± 0.00 |
| ConvPrompt | 88.56 ± 0.40 | 84.15 ± 0.40 | 82.26 ± 0.37 | 80.33 ± 0.25 | 79.36 ± 0.08 | 82.93 ± 0.24 | - | 1.41 ± 0.18 | 2.24 ± 0.28 | 2.36 ± 0.21 | 3.42 ± 0.05 | 2.36 ± 0.16 |
| LEAPGen | 88.23 ± 0.45 | 85.96 ± 0.07 | 85.10 ± 0.03 | 83.24 ± 0.58 | 82.79 ± 0.32 | 85.06 ± 0.29 | - | 0.02 ± 0.04 | 0.07 ± 0.12 | 0.12 ± 0.19 | 0.51 ± 0.04 | 0.18 ± 0.07 |

Table A18: Complete numerical result of consolidated algorithm in ImageNet-R dataset with 5 tasks setting. Color "blue" and "red" denotes the highest and runner-up performance respectively.

| Method | \multicolumn{11}{c}{Task} | Avg. |
|---|---|---|---|---|---|---|---|---|---|---|---|
| | 1 | 2 | 3 | 4 | 5 | 6 | 7 | 8 | 9 | 10 | |
| \multicolumn{12}{c}{Average Accuracy (AA)} | |
| L2P | 76.89 ± 0.00 | 72.96 ± 0.33 | 69.74 ± 0.78 | 66.35 ± 0.43 | 65.43 ± 0.43 | 65.10 ± 0.59 | 64.27 ± 0.63 | 63.65 ± 0.63 | 63.57 ± 0.83 | 62.50 ± 0.51 | 67.05 ± 0.47 |
| DualPrompt | 78.97 ± 0.83 | 76.34 ± 0.30 | 74.96 ± 0.46 | 72.82 ± 0.21 | 71.34 ± 0.30 | 70.92 ± 0.24 | 69.82 ± 0.24 | 69.16 ± 0.26 | 68.87 ± 0.10 | 68.59 ± 0.24 | 72.18 ± 0.20 |
| CODA-P | 89.24 ± 0.93 | 84.47 ± 3.00 | 82.10 ± 2.60 | 80.19 ± 2.81 | 78.97 ± 2.44 | 77.84 ± 1.69 | 76.79 ± 1.97 | 75.64 ± 1.43 | 74.80 ± 1.07 | 73.77 ± 0.50 | 79.38 ± 1.48 |
| LGCL | 78.59 ± 0.93 | 76.61 ± 0.26 | 75.51 ± 0.40 | 73.50 ± 0.30 | 72.24 ± 0.22 | 71.45 ± 0.17 | 70.10 ± 0.34 | 69.67 ± 0.41 | 69.41 ± 0.14 | 68.65 ± 0.25 | 72.57 ± 0.19 |
| HiDe-Prompt | 85.22 ± 0.22 | 82.93 ± 0.59 | 82.35 ± 0.76 | 79.74 ± 0.10 | 78.85 ± 0.56 | 77.81 ± 0.46 | 77.37 ± 0.58 | 76.38 ± 0.35 | 76.31 ± 0.22 | 75.75 ± 0.40 | 79.27 ± 0.17 |
| PGP | 78.97 ± 0.83 | 76.44 ± 0.40 | 74.96 ± 0.45 | 72.87 ± 0.09 | 71.32 ± 0.39 | 70.90 ± 0.26 | 69.78 ± 0.19 | 69.20 ± 0.18 | 68.87 ± 0.12 | 68.62 ± 0.14 | 72.19 ± 0.20 |
| EvoPrompt | 89.27 ± 0.67 | 84.90 ± 0.35 | 83.43 ± 0.23 | 82.03 ± 0.55 | 80.97 ± 0.91 | 80.00 ± 0.72 | 78.83 ± 0.74 | 77.43 ± 0.47 | 76.83 ± 0.25 | 76.00 ± 0.26 | 80.97 ± 0.30 |
| CPrompt | 90.91 ± 0.70 | 86.17 ± 2.15 | 84.25 ± 0.38 | 82.57 ± 0.17 | 81.11 ± 0.45 | 79.76 ± 0.38 | 78.67 ± 0.55 | 78.14 ± 0.36 | 77.13 ± 0.15 | 76.32 ± 0.53 | 81.50 ± 0.30 |
| ConvPrompt | 89.53 ± 0.52 | 86.24 ± 0.48 | 84.40 ± 0.20 | 81.88 ± 0.14 | 80.79 ± 0.10 | 80.04 ± 0.07 | 78.80 ± 0.02 | 78.15 ± 0.26 | 77.74 ± 0.28 | 77.08 ± 0.26 | 81.47 ± 0.10 |
| LEAPGen | 89.83 ± 0.38 | 87.63 ± 0.68 | 86.46 ± 0.63 | 85.97 ± 0.68 | 84.71 ± 1.68 | 84.79 ± 1.55 | 84.01 ± 1.41 | 83.77 ± 1.32 | 84.12 ± 1.05 | 84.09 ± 0.93 | 85.54 ± 0.65 |
| \multicolumn{12}{c}{Average Forgetting Measure (FM)} | |
| L2P | - | 5.52 ± 0.52 | 3.98 ± 0.29 | 4.52 ± 0.65 | 4.35 ± 0.48 | 3.81 ± 0.42 | 4.00 ± 0.48 | 4.06 ± 0.40 | 4.43 ± 0.40 | 5.01 ± 0.40 | 4.41 ± 0.43 |
| DualPrompt | - | 3.83 ± 0.80 | 2.51 ± 0.07 | 2.97 ± 0.39 | 3.76 ± 0.48 | 3.38 ± 0.29 | 3.77 ± 0.19 | 3.87 ± 0.12 | 4.58 ± 0.08 | 4.61 ± 0.07 | 3.70 ± 0.18 |
| CODA-P | - | 5.04 ± 2.63 | 6.06 ± 0.85 | 5.83 ± 1.47 | 6.61 ± 1.12 | 6.94 ± 0.73 | 7.13 ± 0.89 | 7.37 ± 0.94 | 7.56 ± 0.37 | 7.94 ± 0.08 | 6.72 ± 0.79 |
| LGCL | - | 2.91 ± 1.40 | 2.08 ± 0.74 | 2.72 ± 0.84 | 3.15 ± 0.51 | 3.23 ± 0.31 | 3.65 ± 0.62 | 3.68 ± 0.54 | 4.29 ± 0.18 | 4.75 ± 0.33 | 3.38 ± 0.58 |
| HiDe-Prompt | - | 3.29 ± 0.73 | 1.50 ± 0.30 | 2.41 ± 0.56 | 2.19 ± 0.15 | 2.53 ± 0.50 | 2.20 ± 0.55 | 2.25 ± 0.37 | 2.30 ± 0.16 | 2.29 ± 0.27 | 2.33 ± 0.17 |
| PGP | - | 3.83 ± 0.80 | 2.39 ± 0.12 | 2.94 ± 0.26 | 3.67 ± 0.54 | 3.32 ± 0.45 | 3.76 ± 0.40 | 3.75 ± 0.28 | 4.52 ± 0.55 | 4.53 ± 0.40 | 3.63 ± 0.35 |
| EvoPrompt | - | 5.17 ± 0.21 | 3.57 ± 0.38 | 3.57 ± 0.82 | 2.98 ± 1.07 | 2.85 ± 0.72 | 2.70 ± 0.66 | 3.24 ± 0.40 | 4.00 ± 0.40 | 4.22 ± 0.42 | 3.59 ± 0.52 |
| CPrompt | - | 4.55 ± 2.51 | 5.28 ± 2.36 | 5.57 ± 1.42 | 5.49 ± 1.47 | 5.71 ± 1.48 | 5.99 ± 1.15 | 5.75 ± 0.80 | 5.97 ± 0.68 | 6.10 ± 0.75 | 5.60 ± 1.35 |
| ConvPrompt | - | 2.57 ± 0.51 | 1.57 ± 0.08 | 2.60 ± 0.31 | 2.93 ± 0.24 | 3.15 ± 0.31 | 3.53 ± 0.36 | 3.44 ± 0.09 | 4.07 ± 0.17 | 4.17 ± 0.04 | 3.11 ± 0.17 |
| LEAPGen | - | 3.97 ± 0.08 | 1.96 ± 0.13 | 1.38 ± 0.08 | 2.63 ± 2.66 | 2.24 ± 2.19 | 1.99 ± 1.83 | 1.77 ± 1.59 | 1.60 ± 1.35 | 1.46 ± 1.25 | 2.11 ± 1.21 |

Table A19: Complete numerical result of the consolidated algorithms in ImageNet-R dataset with 10 tasks setting. Color "blue" and "red" denotes the highest and runner-up performance respectively.

| \multicolumn{12}{c}{Average Accuracy (AA)} | |
|---|---|---|---|---|---|---|---|---|---|---|---|
| Method | 1 | 2 | 3 | 4 | 5 | 6 | 7 | 8 | 9 | 10 | Avg. |
| L2P | 80.16 ± 0.46 | 73.60 ± 0.53 | 72.34 ± 0.85 | 69.92 ± 0.99 | 68.49 ± 1.10 | 67.47 ± 1.11 | 65.97 ± 0.53 | 64.47 ± 0.60 | 64.33 ± 0.33 | 63.10 ± 0.24 | - |
| DualPrompt | 80.78 ± 0.31 | 78.08 ± 0.82 | 76.46 ± 0.40 | 74.33 ± 0.34 | 73.22 ± 0.45 | 73.58 ± 0.52 | 72.12 ± 0.08 | 71.10 ± 0.36 | 70.99 ± 0.55 | 69.47 ± 0.17 | - |
| CODA-P | 91.77 ± 1.70 | 85.57 ± 1.17 | 84.23 ± 0.66 | 82.04 ± 2.20 | 80.42 ± 2.33 | 79.51 ± 1.98 | 78.23 ± 1.94 | 77.09 ± 2.09 | 76.45 ± 1.92 | 75.99 ± 2.12 | - |
| LGCL | 80.78 ± 0.81 | 77.43 ± 0.41 | 76.07 ± 0.38 | 74.37 ± 0.41 | 73.56 ± 0.77 | 73.02 ± 0.53 | 71.67 ± 0.43 | 71.21 ± 0.14 | 71.57 ± 0.31 | 70.22 ± 0.35 | - |
| HiDe-Prompt | 88.10 ± 0.70 | 85.34 ± 0.66 | 84.22 ± 0.20 | 81.61 ± 0.52 | 80.95 ± 0.35 | 80.56 ± 0.67 | 79.77 ± 0.28 | 78.76 ± 0.47 | 79.13 ± 0.80 | 77.56 ± 0.83 | - |
| PGP | 80.78 ± 0.31 | 78.11 ± 0.77 | 76.49 ± 0.58 | 74.36 ± 0.54 | 73.28 ± 0.27 | 73.33 ± 0.28 | 72.04 ± 0.16 | 70.93 ± 0.46 | 71.06 ± 0.66 | 69.73 ± 0.30 | - |
| EvoPrompt | 90.13 ± 0.75 | 87.37 ± 0.45 | 85.47 ± 0.57 | 83.53 ± 1.55 | 82.90 ± 1.56 | 81.80 ± 0.79 | 81.27 ± 0.15 | 80.57 ± 0.15 | 81.07 ± 0.15 | 79.93 ± 0.21 | - |
| CPrompt | 91.45 ± 1.66 | 88.61 ± 0.93 | 87.58 ± 1.38 | 84.57 ± 2.40 | 83.04 ± 1.00 | 81.94 ± 1.21 | 81.13 ± 1.01 | 80.45 ± 0.68 | 79.46 ± 0.49 | 78.64 ± 0.71 | - |
| ConvPrompt | 91.89 ± 0.55 | 87.69 ± 0.61 | 85.29 ± 0.53 | 83.90 ± 0.11 | 82.58 ± 0.31 | 82.03 ± 0.71 | 80.27 ± 0.59 | 78.67 ± 0.58 | 78.81 ± 0.69 | 77.10 ± 0.53 | - |
| LEAPGen | 93.03 ± 0.85 | 91.18 ± 0.59 | 89.54 ± 1.66 | 88.18 ± 2.27 | 87.83 ± 2.28 | 88.14 ± 1.87 | 88.06 ± 1.53 | 87.89 ± 1.38 | 87.34 ± 1.07 | 87.01 ± 1.12 | - |
| | 11 | 12 | 13 | 14 | 15 | 16 | 17 | 18 | 19 | 20 | Avg. |
| L2P | 58.15 ± 0.23 | 58.75 ± 0.13 | 57.61 ± 0.47 | 58.14 ± 0.08 | 57.80 ± 0.30 | 57.69 ± 0.05 | 57.56 ± 0.58 | 57.21 ± 0.63 | 56.49 ± 0.14 | 57.40 ± 0.31 | 63.33 ± 0.21 |
| DualPrompt | 67.55 ± 0.11 | 67.65 ± 0.25 | 67.12 ± 0.41 | 67.14 ± 0.53 | 67.02 ± 0.47 | 66.65 ± 0.34 | 66.35 ± 0.35 | 65.87 ± 0.16 | 65.60 ± 0.38 | 65.19 ± 0.17 | 70.31 ± 0.29 |
| CODA-P | 75.39 ± 1.36 | 74.92 ± 1.46 | 74.34 ± 1.07 | 73.98 ± 1.35 | 73.24 ± 0.76 | 72.44 ± 0.99 | 72.14 ± 0.77 | 71.82 ± 0.80 | 71.46 ± 0.49 | 70.55 ± 0.71 | 77.08 ± 1.02 |
| LGCL | 67.67 ± 0.40 | 67.40 ± 0.29 | 66.83 ± 1.01 | 66.85 ± 0.60 | 66.91 ± 0.54 | 66.21 ± 0.47 | 65.85 ± 0.37 | 65.74 ± 0.40 | 65.27 ± 0.64 | 64.96 ± 0.67 | 70.18 ± 0.37 |
| HiDe-Prompt | - | - | - | - | - | - | - | - | - | - | 81.6 ± 0.48 |
| PGP | 67.54 ± 0.22 | 67.68 ± 0.50 | 67.09 ± 0.36 | 67.18 ± 0.53 | 67.08 ± 0.23 | 66.80 ± 0.22 | 66.72 ± 0.13 | 65.99 ± 0.32 | 65.74 ± 0.08 | 65.24 ± 0.25 | 70.36 ± 0.26 |
| EvoPrompt | 78.40 ± 0.20 | 78.73 ± 0.29 | 77.57 ± 0.21 | 76.63 ± 0.55 | 76.47 ± 0.71 | 75.63 ± 0.31 | 75.67 ± 0.29 | 75.17 ± 0.25 | 75.07 ± 0.40 | 74.93 ± 0.64 | 79.92 ± 0.13 |
| CPrompt | 77.99 ± 0.51 | 77.26 ± 0.42 | 76.66 ± 0.71 | 76.44 ± 0.62 | 76.35 ± 0.29 | 75.87 ± 0.17 | 75.50 ± 0.39 | 74.95 ± 0.44 | 74.25 ± 0.10 | 74.23 ± 0.17 | 79.82 ± 0.51 |
| ConvPrompt | 75.95 ± 0.81 | 76.36 ± 0.70 | 75.45 ± 0.63 | 75.40 ± 0.40 | 75.38 ± 0.62 | 74.79 ± 0.15 | 74.25 ± 0.23 | 74.51 ± 0.41 | 74.28 ± 0.32 | 73.93 ± 0.36 | 78.92 ± 0.37 |
| LEAPGen | 87.16 ± 1.06 | 86.89 ± 0.25 | 86.33 ± 0.19 | 86.61 ± 0.11 | 86.56 ± 0.14 | 86.60 ± 0.15 | 86.73 ± 0.17 | 86.98 ± 0.15 | 87.04 ± 0.22 | 87.03 ± 0.12 | 87.81 ± 0.48 |
| \multicolumn{12}{c}{Average Forgetting Measure (FM)} | |
| Method | 1 | 2 | 3 | 4 | 5 | 6 | 7 | 8 | 9 | 10 | Avg. |
| L2P | - | 3.70 ± 0.26 | 6.31 ± 1.08 | 6.56 ± 1.11 | 5.78 ± 0.77 | 5.45 ± 0.43 | 5.10 ± 0.31 | 5.13 ± 0.15 | 5.30 ± 0.14 | 4.91 ± 0.33 | - |
| DualPrompt | - | 3.97 ± 1.47 | 6.34 ± 0.57 | 5.86 ± 0.55 | 5.04 ± 0.49 | 4.07 ± 0.74 | 3.95 ± 0.29 | 3.83 ± 0.32 | 4.41 ± 0.45 | 3.80 ± 0.23 | - |
| CODA-P | - | 6.87 ± 3.87 | 4.98 ± 1.71 | 5.08 ± 1.24 | 5.55 ± 1.18 | 5.97 ± 1.21 | 6.41 ± 0.30 | 6.81 ± 1.32 | 7.10 ± 1.30 | 7.13 ± 1.50 | - |
| LGCL | - | 3.88 ± 1.80 | 6.02 ± 0.60 | 5.05 ± 0.63 | 4.42 ± 0.61 | 4.29 ± 0.50 | 3.94 ± 0.23 | 3.65 ± 0.17 | 3.83 ± 0.11 | 3.04 ± 0.09 | - |
| HiDe-Prompt | - | 1.50 ± 1.19 | 2.87 ± 0.11 | 2.98 ± 0.26 | 2.33 ± 0.32 | 2.07 ± 0.35 | 2.05 ± 0.28 | 1.97 ± 0.29 | 2.07 ± 0.43 | 2.26 ± 0.41 | - |
| PGP | - | 3.70 ± 1.32 | 6.51 ± 0.86 | 5.94 ± 0.65 | 5.08 ± 0.60 | 4.46 ± 0.33 | 4.14 ± 0.30 | 4.08 ± 0.13 | 4.31 ± 0.37 | 3.58 ± 0.33 | - |
| EvoPrompt | - | 2.20 ± 0.53 | 5.50 ± 0.09 | 5.84 ± 1.49 | 5.21 ± 1.29 | 6.12 ± 0.49 | 6.14 ± 0.45 | 5.47 ± 0.09 | 4.90 ± 0.10 | 4.66 ± 0.38 | - |
| CPrompt | - | 3.16 ± 0.67 | 3.60 ± 0.95 | 5.42 ± 1.70 | 5.60 ± 1.03 | 5.79 ± 0.92 | 5.74 ± 1.19 | 5.72 ± 0.82 | 5.95 ± 0.64 | 5.86 ± 0.46 | - |
| ConvPrompt | - | 2.03 ± 0.81 | 4.28 ± 0.71 | 3.69 ± 0.64 | 3.07 ± 0.34 | 3.27 ± 0.11 | 3.18 ± 0.18 | 2.93 ± 0.20 | 3.14 ± 0.17 | 2.95 ± 0.19 | - |
| LEAPGen | - | 0.00 ± 0.00 | 3.17 ± 2.83 | 4.83 ± 3.40 | 3.51 ± 2.70 | 2.85 ± 2.08 | 2.62 ± 1.62 | 2.26 ± 1.39 | 2.75 ± 1.16 | 2.57 ± 0.86 | - |
| | 11 | 12 | 13 | 14 | 15 | 16 | 17 | 18 | 19 | 20 | Avg. |
| L2P | 8.78 ± 0.25 | 8.78 ± 0.45 | 9.48 ± 0.72 | 9.73 ± 0.19 | 10.27 ± 0.28 | 10.16 ± 0.23 | 10.52 ± 0.88 | 11.18 ± 0.98 | 11.79 ± 0.31 | 10.76 ± 0.45 | 7.88 ± 0.17 |
| DualPrompt | 4.51 ± 0.31 | 4.82 ± 0.32 | 4.61 ± 0.03 | 5.19 ± 0.31 | 5.16 ± 0.35 | 5.46 ± 0.35 | 5.97 ± 0.12 | 6.81 ± 0.23 | 6.97 ± 0.22 | 7.30 ± 0.18 | 5.16 ± 0.34 |
| CODA-P | 7.23 ± 1.21 | 7.19 ± 1.20 | 7.10 ± 1.03 | 7.25 ± 0.79 | 7.53 ± 0.56 | 7.86 ± 0.34 | 7.83 ± 0.79 | 7.89 ± 0.54 | 7.98 ± 0.51 | 8.23 ± 0.86 | 6.95 ± 0.70 |
| LGCL | 4.29 ± 0.43 | 4.92 ± 0.18 | 4.84 ± 0.99 | 5.39 ± 0.60 | 5.11 ± 0.52 | 5.73 ± 0.53 | 6.24 ± 0.50 | 6.79 ± 0.40 | 7.19 ± 0.62 | 7.35 ± 0.65 | 5.05 ± 0.32 |
| HiDe-Prompt | - | - | - | - | - | - | - | - | - | - | 2.23 ± 0.38 |
| PGP | 4.54 ± 0.07 | 4.84 ± 0.31 | 4.55 ± 0.25 | 5.05 ± 0.35 | 4.95 ± 0.11 | 5.10 ± 0.17 | 5.36 ± 0.16 | 6.53 ± 0.29 | 6.72 ± 0.08 | 7.17 ± 0.21 | 5.09 ± 0.25 |
| EvoPrompt | 5.54 ± 0.40 | 5.27 ± 0.54 | 5.48 ± 0.74 | 6.22 ± 0.77 | 6.09 ± 0.69 | 6.32 ± 0.54 | 6.28 ± 0.78 | 6.96 ± 0.59 | 6.76 ± 0.73 | 6.72 ± 0.90 | 5.67 ± 0.26 |
| CPrompt | 5.73 ± 0.36 | 5.73 ± 0.24 | 5.92 ± 0.08 | 5.73 ± 0.06 | 5.70 ± 0.13 | 5.75 ± 0.19 | 5.80 ± 0.17 | 5.92 ± 0.06 | 6.07 ± 0.30 | 5.98 ± 0.24 | 5.54 ± 0.48 |
| ConvPrompt | 2.99 ± 0.23 | 2.93 ± 0.28 | 3.12 ± 0.38 | 3.65 ± 0.16 | 3.57 ± 0.25 | 4.04 ± 0.51 | 4.65 ± 0.38 | 4.74 ± 0.29 | 4.65 ± 0.40 | 4.87 ± 0.57 | 3.57 ± 0.25 |
| LEAPGen | 2.31 ± 0.78 | 2.69 ± 0.26 | 2.63 ± 0.19 | 2.51 ± 0.22 | 2.41 ± 0.21 | 2.33 ± 0.14 | 2.35 ± 0.13 | 2.18 ± 0.09 | 2.13 ± 0.16 | 2.17 ± 0.17 | 2.54 ± 0.77 |

Table A20: Complete numerical result of the consolidated algorithms in ImageNet-R dataset with 20 tasks setting. Color "blue" and "red" denotes the highest and runner-up performance respectively.

| Method | Task 1 | 2 | 3 | 4 | 5 | 6 | 7 | 8 | 9 | 10 | Avg. |
|---|---|---|---|---|---|---|---|---|---|---|---|
| **Average Accuracy (AA)** | | | | | | | | | | | |
| L2P | 86.34 ± 0.62 | 81.21 ± 0.31 | 75.09 ± 0.88 | 72.83 ± 0.79 | 75.51 ± 0.48 | 74.58 ± 0.32 | 71.85 ± 0.28 | 70.72 ± 0.17 | 65.19 ± 0.41 | 66.95 ± 0.13 | 74.03 ± 0.32 |
| DualPrompt | 93.33 ± 0.90 | 93.33 ± 0.90 | 86.79 ± 0.45 | 81.47 ± 0.47 | 79.40 ± 0.19 | 80.41 ± 0.47 | 79.38 ± 0.27 | 77.76 ± 0.39 | 76.37 ± 0.96 | 73.95 ± 0.73 | 80.29 ± 0.15 |
| CODA-P | 95.13 ± 4.31 | 91.97 ± 2.95 | 88.84 ± 1.34 | 84.55 ± 0.74 | 83.27 ± 1.46 | 81.00 ± 2.63 | 78.64 ± 1.56 | 75.94 ± 1.01 | 74.11 ± 0.68 | 72.99 ± 0.30 | 82.64 ± 0.64 |
| LGCL | 92.75 ± 0.45 | 87.06 ± 0.34 | 82.88 ± 0.39 | 80.97 ± 0.30 | 82.54 ± 0.21 | 82.24 ± 0.26 | 81.33 ± 0.22 | 81.15 ± 0.19 | 79.82 ± 0.35 | 79.93 ± 0.30 | 83.07 ± 0.23 |
| HiDe-Prompt | 94.08 ± 0.41 | 87.97 ± 0.18 | 87.70 ± 0.28 | 85.87 ± 0.30 | 87.06 ± 0.27 | 86.98 ± 0.32 | 86.77 ± 0.04 | 86.45 ± 0.11 | 86.55 ± 0.01 | 87.21 ± 0.18 | 87.66 ± 0.01 |
| PGP | 92.62 ± 0.89 | 86.97 ± 0.62 | 82.34 ± 0.26 | 80.17 ± 0.53 | 81.70 ± 0.72 | 81.11 ± 0.50 | 80.31 ± 0.36 | 80.19 ± 0.56 | 78.32 ± 0.66 | 78.35 ± 0.68 | 82.21 ± 0.56 |
| EvoPrompt | 88.87 ± 1.27 | 85.10 ± 0.00 | 83.83 ± 0.47 | 82.40 ± 0.95 | 80.93 ± 0.90 | 79.93 ± 0.64 | 78.53 ± 0.84 | 77.37 ± 0.72 | 76.83 ± 0.49 | 76.23 ± 0.51 | 81.00 ± 0.40 |
| CPrompt | 97.39 ± 1.09 | 92.82 ± 1.78 | 90.32 ± 0.54 | 87.83 ± 1.56 | 85.66 ± 1.12 | 83.99 ± 0.84 | 81.76 ± 1.57 | 80.55 ± 1.54 | 79.29 ± 1.54 | 77.14 ± 1.16 | 85.67 ± 0.56 |
| ConvPrompt | 96.57 ± 0.40 | 88.81 ± 0.09 | 85.11 ± 0.51 | 82.95 ± 1.09 | 84.56 ± 0.84 | 83.69 ± 0.84 | 82.59 ± 0.67 | 81.48 ± 0.35 | 81.12 ± 0.68 | 80.12 ± 1.37 | 84.70 ± 0.64 |
| **LEAPGen** | 95.86 ± 0.96 | 92.17 ± 0.62 | 93.38 ± 0.55 | 90.87 ± 0.66 | 90.15 ± 1.68 | 90.24 ± 1.46 | 89.55 ± 1.37 | 89.00 ± 1.17 | 89.33 ± 1.11 | 88.45 ± 0.58 | 90.90 ± 0.81 |
| **Average Forgetting Measure (FM)** | | | | | | | | | | | |
| L2P | - | 2.07 ± 0.30 | 14.72 ± 1.15 | 10.04 ± 0.63 | 8.04 ± 0.29 | 8.28 ± 0.08 | 6.95 ± 0.07 | 5.96 ± 0.08 | 5.25 ± 0.27 | 5.18 ± 0.11 | 7.39 ± 0.23 |
| DualPrompt | - | 3.75 ± 0.49 | 13.52 ± 0.86 | 10.09 ± 0.60 | 9.89 ± 0.78 | 9.49 ± 0.54 | 8.25 ± 0.25 | 7.62 ± 0.41 | 6.92 ± 0.19 | 7.87 ± 0.76 | 8.60 ± 0.37 |
| CODA-P | - | 3.92 ± 1.48 | 7.65 ± 0.76 | 10.66 ± 1.91 | 10.26 ± 1.25 | 10.33 ± 0.99 | 10.84 ± 1.61 | 11.83 ± 1.18 | 11.97 ± 2.02 | 11.71 ± 1.49 | 9.91 ± 1.04 |
| LGCL | - | 1.29 ± 0.49 | 10.08 ± 0.74 | 6.92 ± 0.59 | 5.82 ± 0.40 | 5.94 ± 0.31 | 5.29 ± 0.35 | 4.88 ± 0.29 | 4.57 ± 0.39 | 5.45 ± 0.33 | 5.58 ± 0.39 |
| HiDe-Prompt | - | 2.23 ± 0.41 | 2.70 ± 0.86 | 3.74 ± 0.53 | 3.20 ± 0.43 | 2.98 ± 0.16 | 2.80 ± 0.02 | 2.37 ± 0.18 | 2.04 ± 0.26 | 1.90 ± 0.45 | 2.66 ± 0.13 |
| PGP | - | 1.29 ± 0.49 | 10.92 ± 0.71 | 7.99 ± 0.44 | 6.63 ± 0.43 | 6.71 ± 0.22 | 5.78 ± 0.11 | 5.13 ± 0.08 | 4.70 ± 0.11 | 5.76 ± 0.10 | 6.10 ± 0.26 |
| EvoPrompt | - | 4.67 ± 1.02 | 3.28 ± 1.07 | 3.28 ± 1.50 | 3.08 ± 1.08 | 3.22 ± 0.52 | 3.11 ± 1.01 | 3.44 ± 0.68 | 4.08 ± 0.54 | 3.96 ± 0.43 | 3.57 ± 0.83 |
| CPrompt | - | 4.25 ± 1.97 | 6.34 ± 1.04 | 6.95 ± 1.15 | 7.71 ± 0.84 | 9.18 ± 0.84 | 10.91 ± 0.85 | 10.38 ± 1.10 | 10.87 ± 1.27 | 11.65 ± 0.47 | 8.69 ± 0.31 |
| ConvPrompt | - | 0.13 ± 0.22 | 7.44 ± 0.87 | 5.01 ± 0.65 | 4.83 ± 0.30 | 5.24 ± 0.24 | 4.74 ± 0.53 | 4.17 ± 0.44 | 3.89 ± 0.60 | 6.04 ± 0.97 | 4.61 ± 0.45 |
| **LEAPGen** | - | 0.19 ± 0.34 | 0.13 ± 0.22 | 1.63 ± 1.31 | 1.85 ± 1.83 | 1.50 ± 1.44 | 1.87 ± 0.82 | 1.62 ± 0.67 | 1.46 ± 0.52 | 1.32 ± 0.52 | 1.29 ± 0.82 |

Table A21: Complete numerical result of the consolidated algorithms in CUB dataset with 10 tasks setting. Color "blue" and "red" denotes the highest and runner-up performance respectively.

| Dataset | Setting | 1 | 2 | 3 | 4 | 5 | 6 | 7 | 8 | 9 | 10 | 11 | 12 | 13 | 14 | 15 | 16 | 17 | 18 | 19 | 20 | Avg. |
|---|---|---|---|---|---|---|---|---|---|---|---|---|---|---|---|---|---|---|---|---|---|---|
| **LEAPGen-lite Average Accuracy on each Task** | | | | | | | | | | | | | | | | | | | | | | |
| CIFAR100 | 5-task | 97.93 | 97.68 | 96.89 | 96.82 | 97.07 | - | - | - | - | - | - | - | - | - | - | - | - | - | - | - | 97.28 |
| ImageNet-R | 5-task | 87.41 | 84.98 | 84.41 | 82.59 | 82.44 | - | - | - | - | - | - | - | - | - | - | - | - | - | - | - | 84.37 |
| CIFAR100 | 10-task | 99.50 | 99.08 | 98.70 | 98.65 | 98.67 | 98.48 | 98.36 | 98.37 | 98.47 | 98.58 | - | - | - | - | - | - | - | - | - | - | 98.69 |
| ImageNet-R | 10-task | 89.73 | 88.66 | 86.94 | 86.43 | 86.04 | 83.03 | 82.75 | 82.52 | 82.94 | 82.38 | - | - | - | - | - | - | - | - | - | - | 85.14 |
| CUB200 | 10-task | 96.05 | 91.56 | 93.18 | 89.28 | 86.45 | 84.65 | 83.97 | 84.07 | 85.06 | 86.00 | - | - | - | - | - | - | - | - | - | - | 88.03 |
| CIFAR100 | 20-task | 99.47 | 99.50 | 99.36 | 99.28 | 99.35 | 99.21 | 99.10 | 99.17 | 99.11 | 99.15 | 99.72 | 97.78 | 97.27 | 97.83 | 97.88 | 97.91 | 98.00 | 95.88 | 95.28 | | 98.37 |
| ImageNet-R | 20-task | 92.50 | 90.89 | 91.11 | 82.51 | 82.88 | 83.65 | 84.26 | 84.43 | 84.97 | 84.87 | 85.07 | 85.39 | 85.00 | 85.21 | 85.24 | 85.09 | 85.22 | 85.49 | 85.63 | 83.67 | 85.65 |
| **LEAPGen-lite Average Forgetting on each Task** | | | | | | | | | | | | | | | | | | | | | | |
| CIFAR100 | 5-task | - | 0.00 | 0.03 | 0.01 | 0.05 | - | - | - | - | - | - | - | - | - | - | - | - | - | - | - | 0.02 |
| ImageNet-R | 5-task | - | 0.05 | 0.04 | 0.17 | 0.43 | - | - | - | - | - | - | - | - | - | - | - | - | - | - | - | 0.17 |
| CIFAR100 | 10-task | - | 0.03 | 0.00 | 0.03 | 0.05 | 0.07 | 0.08 | 0.09 | 0.10 | 0.11 | - | - | - | - | - | - | - | - | - | - | 0.06 |
| ImageNet-R | 10-task | - | 1.11 | 0.75 | 0.58 | 0.56 | 4.13 | 3.37 | 2.92 | 2.70 | 3.01 | - | - | - | - | - | - | - | - | - | - | 2.13 |
| CUB200 | 10-task | - | 0.03 | 3.68 | 2.79 | 2.24 | 3.59 | 3.11 | 2.72 | 2.46 | 2.29 | - | - | - | - | - | - | - | - | - | - | 2.54 |
| CIFAR100 | 20-task | - | 0.00 | 0.07 | 0.02 | 0.03 | 0.03 | 0.07 | 0.07 | 0.07 | 0.06 | 0.09 | 1.64 | 1.53 | 2.08 | 1.38 | 1.34 | 1.31 | 1.21 | 1.18 | 1.08 | 0.70 |
| ImageNet-R | 20-task | - | 0.00 | 0.09 | 0.16 | 0.12 | 0.17 | 0.16 | 0.28 | 0.31 | 0.43 | 0.34 | 0.49 | 0.46 | 0.61 | 0.67 | 0.81 | 0.95 | 0.88 | 0.90 | 1.06 | 0.47 |

Table A22: Complete Numerical Results of LEAPGen-lite on CIFAR100, ImageNet-R and CUB dataset.

