# OpenReview forum: "Vision and Language Synergy for Rehearsal Free Continual Learning"
_ICLR.cc/2025/Conference — ICLR 2025 Poster_

### Official Review · Reviewer_bACr · 2024-10-22

**Soundness:** 3
**Presentation:** 3
**Contribution:** 2
**Rating:** 3
**Confidence:** 3

**Summary:**

This paper introduces a new prompt-based approach for continual learning that take advantage of language guidance. It uses task-wise generators, soft task ID prediction, and generated prompts as auxiliary data. The method outperforms state-of-the-art models on CIFAR100, ImageNetR, and CUB datasets with significant improvements in accuracy and forgetting measures.

**Strengths:**

1. The paper provides a detailed explanation of current prompt-based continual learning methods and analyzes the weakness associated with these methods, effectively leading to the introduction of the authors' proposed method.
2. The paper presents a novel language-guided continual learning method and thoroughly demonstrates its effectiveness through extensive experiments.

**Weaknesses:**

1. The comparison mentioned in lines 183-184 between "the photo of car" and "the photo of cat," stating they have 15/16 similarity but 1/16 dissimilarity, is confusing. The authors compared the similarity between these two prompts on a **letter-by-letter** basis and concluded that they are similar. However, the similarity between two prompts should not be evaluated in such a manner. After being processed by the tokenizer, the embeddings of the two prompts are not similar, as their meanings differ significantly.
2. The proposed method relies on using ChatGPT to generate descriptive terms for each class and requires an additional Sentence Transformer to obtain embeddings. The introduction of these extra resources creates an unfair comparison with other methods and limits the practical applicability of the approach in real-world scenarios.
3. Although the proposed method achieves state-of-the-art performance on multiple datasets, the improvement is minor on some, such as the comparison with HiDe-Prompt on the CUB200 dataset. This is particularly relevant given the substantial additional resources, like ChatGPT, required by the method, which other approaches do not utilize. Moreover, this suggests that the performance gains on fine-grained datasets may not be significant, as the generated descriptive terms for each class are quite similar.
4. The authors have used excessive line spacing adjustments, which negatively impact the visual presentation of the paper, such as in lines 437-438 and 446-447. They should revise the layout of the entire paper to provide a better reading experience for the reader.

**Questions:**

Please the weakness.

---

> ### Author Response · Authors · 2024-11-22
> **Autrhor Response to Reviewer bACr for Q1/W1-Q2/W2**
>
> Q1/W1. The comparison mentioned in lines 183-184 between "the photo of car" and "the photo of cat," stating they have 15/16 similarity but 1/16 dissimilarity, is confusing. The authors compared the similarity between these two prompts on a letter-by-letter basis and concluded that they are similar. However, the similarity between two prompts should not be evaluated in such a manner. After being processed by the tokenizer, the embeddings of the two prompts are not similar, as their meanings differ significantly.
>
> Our Response: Thank you for your concern.
>
> -  About the writing error in lines 183-184. We apologize for our writing error. We have revised the sentences in our revised manuscript. The sentences are revised into:  "LGCL produces class prototype $L_c^n$ by encoding string ”the photo of class name”. However, the prototypes could be misleading due to high similarity between different classes, e.g. the prototype of class ”Great White
> Shark” has 0.9 cosine similarity to the prototypes of class ”Tree Frog” and ”Iguana”, please see Appendix D.2. "
>
> - Our statement is supported by our empirical study on the high similarity of language embedding encoded by CLIP as presented in Appendix D.2.
>
> Q2/W2. The proposed method relies on using ChatGPT to generate descriptive terms for each class and requires an additional Sentence Transformer to obtain embeddings. The introduction of these extra resources creates an unfair comparison with other methods and limits the practical applicability of the approach in real-world scenarios.

---

> ### Author Response · Authors · 2024-11-22
> **Autrhor Response to Reviewer bACr for Q3/W3-Q4/W4**
>
> Q3/W3. Although the proposed method achieves state-of-the-art performance on multiple datasets, the improvement is minor on some, such as the comparison with HiDe-Prompt on the CUB200 dataset. This is particularly relevant given the substantial additional resources, like ChatGPT, required by the method, which other approaches do not utilize. Moreover, this suggests that the performance gains on fine-grained datasets may not be significant, as the generated descriptive terms for each class are quite similar.
>
>
> Our Response: Thank you for your concern.
>
> -  We added a new experiment and analysis on the "class name" as descriptors where we have no class descriptors generated by GPT or other LLMs. Thus we utilize equal resources as language-guided prompting such as LGCL and even less resource than ConvPrompt and GMM.  Our experiment results as presented in Appendix D4 shows that our method outperforms the existing SOTAs by a significant margin in 3 datasets despite not using the descriptors by LLMs. It proves that our method can substitute class descriptors with the class name to generate prompts.
>
> - Regarding comparison on CUB dataset, It may look small difference between our method and HiDe-prompt in CUB dataset if we see from final task measurements i.e. 1.24\% FAA and 0.6\% FFM. However, in terms of cumulative measurements, our method outperforms HiDe-Prompt with a significant margin i.e. 3.3\% CAA and 1.36\% CFM. The table A19 in Appendix G shows the complete numerical result of the CUB dataset. The table shows that except in the last 2 tasks, LEAPGen outperforms HiDe-Prompt by a significant margin i.e. 2.8-5.5\%. In addition, HiDE-Prompt suffers from a significant drop after the first task i.e. 6\% accuracy drop, while LEAPGen only experiences a 3\% drop. Thus, in earlier tasks, LEAPGen outperforms HiDePrompt with more than a 5\% margin. In terms of forgetting measures, LEAPGen consistently experiences lower average forgetting than HiDe-Prompt in every task with a 1-2.5\% margin. In the earlier tasks, It is even close to zero. Those numerical result shows that LEAPGen outperforms HiDE-Prompt in CUB dataset.
>
> - Please see the running time analysis in Table A9 in Appendix D9 that shows HiDe-Prompt requires 12x higher running time than our proposed method (3.94h vs 0.32h) for the CUB dataset. This drawback is important to be considered along with the comparison of the performance.
>
> - As for the fine-grained dataset, we confirm that ImageNet-R also has fine-grained classes i.e.:
>
> a. Subtypes of Dog: ['west highland white terrier', 'golden retriever', 'labrador retriever', 'cocker spaniels', 'collie', 'border collie', 'rottweiler', 'german shepherd dog', 'boxer', 'french bulldog'
> 'saint bernard', 'husky', 'dalmatian', 'pug', 'pomeranian', 'Chow Chow', 'pembroke welsh corgi', 'toy poodle', 'standard poodle']
>
> b. Subtypes of fish: ['goldfish', 'great white shark', 'hammerhead','stingray', 'clown fish', 'puffer fish', 'grey whale', 'killer whale']
>
> c. Subtypes of bird: ['hen', 'ostrich', 'goldfinch', 'junco', 'bald eagle', 'vulture,'peacock', 'lorikeet', 'hummingbird', 'toucan', 'duck', 'goose', 'black swan','flamingo', 'american egret', 'pelican']
>
> The result in ImageNet-R as shown in Table 2 shows that our method works well in spite of the presence of fine-grained classes. Our method outperforms existing SOTA with a significant margin i.e. 12-30\% FAA and 8-24\% CAA in 20-task settings, 7-22\% FAA and 4-18\% CAA in 10-task settings, and 3-18\% FAA and 2-17\% CAA in 5-task settings.
> These numerical results given the fact that Imagenet-R contains fine-grained classes prove that fine-grained classes don't decrease the significance of our method.
>
>
> Q4/W4. The authors have used excessive line spacing adjustments, which negatively impact the visual presentation of the paper, such as in lines 437-438 and 446-447. They should revise the layout of the entire paper to provide a better reading experience for the reader.
>
> Our Response: Thank you for your suggestion. We have ensured the the clear spacing for the mentioned parts and other components to improve the visualization of our paper. Please kindly see our latest manuscript.

---

> > ### Comment · Reviewer_bACr · 2024-11-25
> >
> > Thanks for your reply. Would you please showcase latency metrics such as inference speed, fps, storage and latency? I am curious about it given that additional ChatGPT are utilized and it plays a crucial role in the pipeline. I do not see them explicitely in the reply and updated manuscripts

---

> > > ### Author Response · Authors · 2024-12-01
> > > **Third Follow Up**
> > >
> > > Dear Reviewer bACr,
> > >
> > > We would like to follow up on our responses. Could you please kindly review our updated manuscript or previous comments regarding your concerns?
> > >
> > > We believe we have addressed all your concerns in our updated paper and previous responses (comments). Thus, we kindly request Reviewer bACr to reevaluate the score for our paper.
> > >
> > > Thank you.
> > >
> > > Best Regards,
> > >
> > > Author of Submission 6684

---

> > > > ### Comment · Reviewer_bACr · 2024-12-03
> > > >
> > > > Thanks for your response. Answer to the first question seems to be the primary concern and I still find it not convincing.
> > > >
> > > > As for the utilization of GPT, I understand what the author means, but the actual storage and consumption should not be dismissed even if it is pre-computed. Since GPT is more like an external knowledge base, I'm not sure if it is fair to compare it with existing methods. More substantial improvements are supposed to observe since the introduction of these large-scale models.

---

> ### Author Response · Authors · 2024-11-22
> **Follow Up on Author Response and Revised Manuscript**
>
> We would like to follow up on our response and revised manuscript. We would appreciate it if Reviewer bACr could look at our revised manuscript, and offer additional comments.

---

> ### Author Response · Authors · 2024-11-26
> **Response to Reviewer bACr for Additional Comments**
>
> Thank you for your suggestion.
>
> 1. We have added the advised metrics i.e. inference time only, along with total time, including additional time for descriptors generation as utilized by LEAPGen and ConvPrompt. Our method has the lowest total running time, and moderate inference time i.e. lower than ConvPrompt and higher than HiDe-Prompt. We have added the required storage, please see Appendix D.3.
>
> 2. We would like to emphasize the following points:
>
> - Following ConvPrompt, the LLM such as GPT is one of the alternatives to generate descriptors before the training phase. Thus the GPT is not part of the methods. In the case of no classes' descriptors,  Our method still works excellently i.e. utilizing class names as descriptors (Appendix D.4).
>
> - GPT is utilized via online query, thus we don't need extra storage to save it.
>
> - Descriptors generation by GPT indeed consumes a fair amount of time. Even though spending additional time for descriptors generations, our method still has a lower total running time than HiDe-Prompt and ConvPrompt.

---

> ### Author Response · Authors · 2024-11-27
> **Follow Up**
>
> Dear Reviewer bACr,
>
> Thank you for your feedback on our paper. Could you please kindly review our latest manuscript regarding your concern?
> Here are the pointers to your points on the use of ChatGPT:
>
> - Section 5.2.d Tables 2,3,4, and Figure 3:  Analysis of LEAPGen-lite (lightweight generators and without descriptors) performance that proves the significant performance of our ideas is not bounded by LLM-generated descriptors or the complexity (parameters and size) of the generators.
>
> - Section 5.2.g: Analysis of parameters, running time, and storage
>
> - Appendix D.3: Detailed Analysis of Performance and Cost TradeOffs.
>
> Thank you.
>
> Best Regards,
>
> Author of Submission 6684

---

> ### Author Response · Authors · 2024-11-29
> **Second Follow Up**
>
> Dear Reviewer bACr,
>
> Thank you for your feedback on our paper. Could you please kindly review our latest manuscript regarding your concern?
>
> Here are the pointers to your key points:
>
> We also copied the updates (or snapshots) here, thus you can review them directly.
>
>
> - Section 5.2.g: Analysis of parameters, running time, and storage
>
> | Method       | Desc | #Params(M) | Running Time (h) |       |        |       | Storage (MB) |
> |--------------|:----:|:----------:|:----------------:|:-----:|:------:|:-----:|:------------:|
> |              |      |            |      T.Desc      |  Inf  | Tr+Inf | Total |              |
> | HiDe-Prompt  |   -  |    0.15    |         -        | 0.019 |  5.40  |  5.40 |      334     |
> | ConvPrompt   |   v  |    1.28    |       1.07       | 0.033 |  1.04  |  2.11 |      346     |
> | LEAPGen-lite |   -  |    0.16    |         -        | 0.028 |  0.53  |  0.53 |      332     |
> | LEAPGen      |   v  |    6.35    |       1.07       | 0.025 |  0.72  |  1.79 |      567     |
>
> T.Desc, Tr, and Inf denote time for generating descriptors, training, and inference respectively, detailed in Appendix D3.
>
> 5.2.g) Parameters, Running Time, and Storage: The table above compares the number of parameters, running time (ImageNet-R), and storage of our methods and existing SOTAs. Despite having a higher number of parameters and storage, LEAPGen consumes less running time than existing SOTAs both training+inference and total running time. LEAPGen-lite consumes the least costs in total running time and storage and requires relatively low parameters and inference time. LEAPGen and ConvPrompt require additional time to generate descriptors that increase their total simulation time. Despite having the least parameters, Hide-Prompt requires the longest training and total times since it needs extra operations to generate uninstructed class representations.
>
>
> - Appendix D.3: Detailed Analysis of Performance and Cost TradeOffs.
>
> (a) Running Time for All Datasets
>
> | Method       |   Time (h)   |       |        |       |                |       |        |       |         |       |        |       |
> |--------------|:------------:|:-----:|:------:|:-----:|:--------------:|:-----:|:------:|:-----:|:-------:|:-----:|:------:|:-----:|
> |              | CIFAR100 10T |       |        |       | ImageNet-R 10T |       |        |       | CUB 10T |       |        |       |
> |              |    T.Desc.   |  Inf  | Tr+Inf | Total |     T.Desc.    |  Inf  | Tr+Inf | Total | T.Desc. |  Inf  | Tr+Inf | Total |
> | HiDe-Prompt  |       -      | 0.037 |  4.63  |  4.63 |        -       | 0.019 |  5.40  |  5.40 |    -    | 0.017 |  3.94  |  3.94 |
> | ConvPrompt   |     0.53     | 0.023 |  2.01  |  2.54 |      1.07      | 0.033 |  1.04  |  2.11 |   1.09  | 0.035 |  8.08  |  9.17 |
> | LEAPGen-lite |       -      | 0.016 |   1.2  |  1.2  |        -       | 0.028 |  0.53  |  0.53 |    -    | 0.028 |  0.38  |  0.38 |
> | LEAPGen      |     0.53     | 0.017 |  0.98  |  1.51 |      1.07      | 0.025 |  0.72  |  1.79 |   1.09  | 0.025 |  0.32  |  1.41 |
>
> (a) Performance vs Storage.
>
> | Method            | Performance |       | Storage(MB) |
> |-------------------|:-----------:|:-----:|:-----------:|
> |                   |     FAA     |  CFA  |             |
> | HiDe-Prompt       |    75.75    | 79.27 |     334     |
> | HiDe-Prompt-Large |    74.30    | 78.56 |     797     |
> | ConvPrompt        |    77.08    | 81.47 |     346     |
> | ConvPrompt-Large  |    74.56*   | 80.79 |     632     |
> | LEPGen-lite       |    82.38    | 85.14 |     332     |
> | LEAPGen           |    84.09    | 85.54 |     567     |
>
> HiDe-Prompt-Large and ConvPrompt-Large is the large version of HiDe-Prompt and ConvPrompt that have 12.2M and 13.3M parameters respectively.
>
> **Again, we want to emphasize these points:**
>
> - Following ConvPrompt, the LLM such as GPT is one of the alternatives to generate descriptors before the training phase (Please see figure 2 and section 4.2-4.3). Thus the GPT is not part of the methods. In the case of no classes' descriptors, Our method works excellently i.e. utilizing class names as descriptors as demonstrated by LEAPGen-lite and LEAPGen-CN(Appendix D.4).
>
> - GPT is utilized via online query, thus we don't need extra storage to save it.

---

> ### Author Response · Authors · 2024-11-29
> **Second Follow Up (Cont'd)**
>
> W2. Increase of Parameters:
> - Section 5.2.d Tables 2,3,4, and Figure 3: Analysis of LEAPGen-lite (lightweight generators and without descriptors) performance.
>
> 5.2.d) LEAPGen-lite’s Performance: As shown in Table 2-4 and Figure 3 Despite utilizing far smaller
> (2.67% params) generators and without class descriptors generated by LLM, LEAPGen-lite still
> outperforms the existing methods significantly i.e. 4.7-25% FAA and 3-11% CAA in CIFAR100,
> and 3-30% FAA and 2-26% CAA in ImageNet-R dataset. LEAPGen-lite also achieves a low for-
> getting rate in these 2 datasets for all task-settings. In the CUB dataset, LEAPGen-lite archives a
> comparable performance to HiDe-Prompt and outperforms other SOTA with a significant margin i.e.
> 6-20% FAA and 3.3-14% CAA. This evidence proves our ideas i.e. language embedding as input
> for prompt generation, task-wise generators, soft task-id predictor, and learning with auxiliary data
> are not bounded by the generated descriptors and the size of generators.
>
> Here is the snapshot of the tables, please see our revised paper for the full tables.
>
> | **Method**       | **ImageNet-R**   |                  |                 |                 |
> |------------------|------------------|------------------|-----------------|-----------------|
> |                  | **FAA**          | **CAA**          | **FFM**         | **CFM**         |
> |                  | 5 Tasks @40c     |                  |                 |                 |
> | L2P              | 64.62 ± 0.32     | 68.01 ± 0.42     | 3.94 ± 0.16     | 3.55 ± 0.20     |
> | DualPrompt       | 69.71 ± 0.11     | 72.78 ± 0.14     | 3.32 ± 0.16     | 2.78 ± 0.25     |
> | CODA-P           | 74.89 ± 0.36     | 79.71 ± 1.27     | 8.89 ± 0.65     | 7.65 ± 0.98     |
> | LGCL             | 69.93 ± 0.21     | 72.91 ± 0.19     | 3.04 ± 0.36     | 2.50 ± 0.38     |
> | HiDe-Prompt      | 75.40 ± 0.27     | 78.88 ± 0.04     | 3.15 ± 0.46     | 2.64 ± 0.16     |
> | PGP              | 69.71 ± 0.15     | 72.77 ± 0.07     | 3.36 ± 0.23     | 2.85 ± 0.25     |
> | EvoPrompt        | 77.27 ± 0.40     | 81.67 ± 0.18     | 1.79 ± 0.31     | 1.41 ± 0.32     |
> | CPrompt          | 78.65 ± 0.00     | 82.44 ± 0.00     | 6.00 ± 0.00     | 5.49 ± 0.00     |
> | ConvPrompt       | 79.36 ± 0.08     | 82.93 ± 0.24     | 3.42 ± 0.05     | 2.36 ± 0.16     |
> | **LEAPGen-lite** | **82.44 ± 0.63** | **84.37 ± 0.90** | **0.43 ± 0.08** | **0.17 ± 0.06** |
> | **LEAPGen**      | **82.79 ± 0.32** | **85.06 ± 0.29** | **0.51 ± 0.04** | **0.18 ± 0.07** |
> |                  | ImageNet-R       |                  |                 |                 |
> | L2P              | 62.50 ± 0.51     | 67.05 ± 0.47     | 5.01 ± 0.40     | 4.41 ± 0.43     |
> | DualPrompt       | 68.59 ± 0.24     | 72.18 ± 0.20     | 4.61 ± 0.07     | 3.70 ± 0.18     |
> | CODA-P           | 73.77 ± 0.50     | 79.38 ± 1.48     | 7.94 ± 0.08     | 6.72 ± 0.79     |
> | LGCL             | 68.65 ± 0.25     | 72.57 ± 0.19     | 4.75 ± 0.33     | 3.38 ± 0.58     |
> | HiDe-Prompt      | 75.75 ± 0.40     | 79.27 ± 0.17     | 2.29 ± 0.27     | 2.33 ± 0.17     |
> | PGP              | 68.62 ± 0.14     | 72.19 ± 0.20     | 4.53 ± 0.40     | 3.63 ± 0.35     |
> | EvoPrompt        | 76.00 ± 0.26     | 80.97 ± 0.30     | 4.22 ± 0.42     | 3.59 ± 0.52     |
> | CPrompt          | 76.32 ± 0.53     | 81.50 ± 0.30     | 6.10 ± 0.75     | 5.60 ± 1.35     |
> | ConvPrompt       | 77.08 ± 0.26     | 81.47 ± 0.10     | 4.17 ± 0.04     | 3.11 ± 0.17     |
> | **LEAPGen-lite** | **82.38 ± 1.04** | **85.14 ± 0.52** | **3.01 ± 1.19** | **2.13 ± 0.60** |
> | **LEAPGen**      | **84.09 ± 0.93** | **85.54 ± 0.65** | **1.46 ± 1.25** | **2.11 ± 1.21** |
> |                  | ImageNet-R       |                  |                 |                 |
> | L2P              | 57.40 ± 0.31     | 63.33 ± 0.21     | 10.76 ± 0.45    | 7.88 ± 0.17     |
> | DualPrompt       | 65.19 ± 0.17     | 70.31 ± 0.29     | 7.30 ± 0.18     | 5.16 ± 0.34     |
> | CODA-P           | 70.55 ± 0.71     | 77.08 ± 1.02     | 8.23 ± 0.86     | 6.95 ± 0.70     |
> | LGCL             | 64.96 ± 0.67     | 70.18 ± 0.37     | 7.35 ± 0.65     | 5.05 ± 0.32     |
> | HiDe-Prompt      | -                | 81.60 ± 0.48     | -               | 2.23 ± 0.38     |
> | PGP              | 65.24 ± 0.25     | 70.36 ± 0.26     | 7.17 ± 0.21     | 5.09 ± 0.25     |
> | EvoPrompt        | 74.93 ± 0.64     | 79.92 ± 0.13     | 6.72 ± 0.90     | 5.67 ± 0.26     |
> | CPrompt          | 74.23 ± 0.17     | 79.82 ± 0.51     | 5.98 ± 0.24     | 5.54 ± 0.48     |
> | ConvPrompt       | 73.93 ± 0.36     | 78.92 ± 0.37     | 4.87 ± 0.57     | 3.57 ± 0.25     |
> | **LEAPGen-lite** | **83.67 ± 0.39** | **85.65 ± 0.33** | **1.06 ± 0.24** | **0.47± 0.14**  |
> | **LEAPGen**      | **87.03 ± 0.12** | **87.81 ± 0.48** | **2.17 ± 0.17** | **2.54 ± 0.77** |
>
> **Please Note: The analysis of LEAGen-lite empirically proves that our method doesn't rely on GPT/LLM or complex generator networks (high number of parameters)**
>
> Thank you.
>
> Best Regards,
>
> Author of Submis.6684

---

> ### Author Response · Authors · 2024-12-03
> **Response to Reviewer bACr**
>
> Thank you for your response.
>
> Please kindly note that:
>
> -  **LEAPGen utilizes the same setting and resources (text encoder, GPT-generated descriptors, etc) as ConvPrompt.**
>
> - **LEAPGen-lite utilizes even less resources (without GPT-generated descriptors, fewer parameters, less running time, and less storage) that are comparable to the other SOTAs.**
>
> Considering such **comparable resources (LEAPGen vs ConvPrompt, and LEAPGen-lite vs other SOTAs)**, we believe **we have satisfied the fairness aspect.**
>
> Also please kindly note that:
>
> - **Descriptors are optional in our method, our method works excellently and significantly outperforms existing SOTAs by utilizing class names as language modality. Thus, our method doesn't rely on GPT or LLMs.**
> - **GPT is utilized via online query so we don't save the GPT model. It adds fair additional running time, but not storage. Again, LEAPGen and ConvPrompt are evaluated with the same setting.** Online querying GPT utilizing API and Python script is a common practice. We don't need to download and save the GPT model. This is applicable in real-application as in our simulation.
>
> Please go thoughtfully on our revised **manuscripts (mainly sec. 5.2.g. and Appendix D.3)** and our previous responses (comments).
>
> In summary, once again we emphasize that **the impact of our method is not bounded by GPT, parameters, running time, and storage.**
>
> Please let us know which part you are not sure about, Thank you.
>
> Best Regards,
>
> Author of Submission 6684

---

> ### Author Response · Authors · 2024-12-03
> **Follow Up**
>
> Dear Reviewer bACr,
>
> We would like to follow up on our responses. Could you please kindly review our latest response that detailed in our updated manuscript? Could you please point out which part you are not sure regarding the fairness aspect and GPT-generated descriptors?
>
> We believe we have addressed all your concerns in our updated paper and previous responses (comments). Thus, we kindly request Reviewer bACr to reevaluate the score for our paper.
>
> Thank you.
>
> Best Regards,
>
> Author of Submission 6684

---

> ### Author Response · Authors · 2024-12-04
> **Summary of Discussion**
>
> Dear Reviewer bACr,
>
> First, we thank reviewer bACr for your effort and time to review our work, rebuttal, and revised paper.
> We believe we have addressed all your concerns in our revised paper and previous responses (comments)
> Second, since the discussion phase has ended, We would like to summarize our discussion and emphasize the following points:
>
> 1. **Fairness Setting** :   LEAPGen utilizes the same resources as ConvPrompt and LEAPGen-lite(no descriptors with lightweight generators) utilizes less resources than ConvPrompt and comparable resources to existing SOTAs (sec. 5.1, sec. 5.2.g, Appendix E). All methods are evaluated in the same setting and run with respective best (recommended by official paper/code) hyperparameters. Thus we believe we have satisfied the fairness aspect in our study.
>
> 2. **The presence of GPT**  :
>
> - **GPT is not part of our method** (sec 4.2.a, and fig. 2), It is an optional part to generate descriptors following ConvPrompt. Even without GPT-generated descriptors. our method works excellently and significantly outperforms the existing SOTAs, as demonstrated by LEAPGen-lite (sec. 5.2.d) and LEAPGen-CN (Appendix D.4).
>
> - **GPT is utilized in an online way** (through the Internet), by querying it using API and Python script. It is a common practice nowadays as in our simulation (please see https://platform.openai.com/docs/guides/). Thus we never download and save the GPT model in our storage. LEAPGen indeed saves the descriptors embedding (numerical vector), but not the GPT model.  The embedding adds to the whole model's storage size but is still in a reasonable amount, and we have included it in our storage analysis.
>
> - **Descriptor generation by GPT** (before training) indeed takes a fair amount of time. However, despite spending extra time on it, our method still has a **lower total running time** than ConvPrompt and HidePrompt.
>
> - **LEAPGen-lite** proves the significant impacts of our ideas i.e. (1) language as input for prompt generation, (2) task-wise generators, (3) soft task-id predictor, and (4)learning with auxiliary despite without GPT-generated descriptors, and utilize least parameters, running time and storage.
>
> We believe, these highlights clarify your uncertainties about the fairness setting and (again) prove the significant impacts of our ideas, without being bound to the presence of GPT.
>
> Best Regards,
>
> Author of Submission 6684

---

### Official Review · Reviewer_UGi2 · 2024-11-01

**Soundness:** 2
**Presentation:** 3
**Contribution:** 2
**Rating:** 3
**Confidence:** 4

**Summary:**

This paper presents a novel approach to address the catastrophic forgetting in continual learning through a prompt-based structure. This method incorporates language inputs for prompt generation and utilizes task-wise generators and soft task-ID prediction. The authors highlight the advantages over existing methods across various datasets, showcasing substantial improvements accuracy while minimizing forgetting metrics.

**Strengths:**

- The authors systematically review the limitations of existing prompt-based methods, and Figure 1 commendably illustrates the comparison of these methods.

- The proposed method demonstrates performance that far exceeds current methods, which is pleasantly surprising.

- The paper introduces the novel and meaningful use of large language models to generate more refined descriptions.

**Weaknesses:**

- The writing of this paper needs significant improvement. Additionally, the structure is overly compact, affecting readability. For instance, the theorem presented in the paper has weak relevance to the proposed method, and I believe it is unnecessary. I suggest moving it to the supplementary materials. Given the complexity and sophistication of the method, I encourage the authors to provide an overview initially.

- The provided ablation study is not sufficiently convincing. I recommend that the authors validate the effectiveness of the proposed modules step-by-step starting from a standard baseline.

- The inclusion of the ChatGPT model raises the question of how to ensure the fairness of experimental comparisons under such settings, and to what extent the performance gains are due to more accurate descriptions.

- Considering the authors (are about to) open-source their code, I suggest making the descriptions obtained via ChatGPT publicly available for deeper analysis. I believe this will enhance the impact of the paper.

**Questions:**

Please see the comments of weaknesses.

---

> ### Author Response · Authors · 2024-11-22
> **Author Response to Reviewer UGi2**
>
> W1/Q1. The writing of this paper needs significant improvement. Additionally, the structure is overly compact, affecting readability. For instance, the theorem presented in the paper has weak relevance to the proposed method, and I believe it is unnecessary. I suggest moving it to the supplementary materials. Given the complexity and sophistication of the method, I encourage the authors to provide an overview initially.
>
> Our Response: Thank you for your suggestion. We have improved the readability of our paper by adding an overview of our method as presented in section 4.1, moving theorems 2 and 3 into the Appendix section, and improving the general layout of our paper
>
> W2/Q2. The provided ablation study is not sufficiently convincing. I recommend that the authors validate the effectiveness of the proposed modules step-by-step starting from a standard baseline.
>
> Our Response: Thank you for your suggestion. We have revised our ablation study with a step-by-step scenario from fine-tuning (FT) baseline, emphasizing the clarity of each component contribution, Please see section 5.2.d.
>
> W3/Q3. The inclusion of the ChatGPT model raises the question of how to ensure the fairness of experimental comparisons under such settings, and to what extent the performance gains are due to more accurate descriptions.
>
> Our Responses:
>
> - We added a new experiment and analysis on the "class name" as descriptors where we have no class descriptors generated by GPT or other LLMs. Our experiment results as presented in Appendix D4. shows that our method outperforms the existing SOTAs by a significant margin in 3 datasets despite substituting class descriptors with the class name.
>
> - In our main experimental setting, We follow ConvPrompt which utilizes GPT to generate class descriptors before the training process. Thus we emphasize that our method and ConvPrompt utilize the same external resources that are considered as fair.  Please note that in our method, the GPT (LLM) is solely utilized to generate descriptors, It is different from GMM which utilizes an LLM decoder in its learning process.
>
> - For the other methods, we can't enforce ChatGPT in them as they were designed to learn without language descriptors. The most feasible setting is evaluating the methods following their official (optimal) setting and hyperparameters that we have done in our study. Thus we believe that we maintain fairness in our evaluation.
>
> - As for the relation between descriptors and performance, we have extended our analysis about the performance of our method w.r.t types of descriptors and LLM (GPT, Gemini, and Llama) in comparison to ConvPrompt. The table below shows the performance of our method compared to ConvPrompt in those settings. Please kindly see Appendix D.5 for a detailed sample of descriptors, results, and analysis. In summary, we conclude that long descriptors are preferable over short and narrative descriptors as generally they carry richer visual descriptions of an object without being contaminated by unrelated words. However, our methods can utilize three of the descriptors types, and gain better performance than ConvPrompt.
>
> W4/Q4. Considering the authors (are about to) open-source their code, I suggest making the descriptions obtained via ChatGPT publicly available for deeper analysis. I believe this will enhance the impact of the paper.
>
> Our Response: Thank you for your suggestion, We have published all the descriptors including short, long, and narrative descriptors generated by ChatGPT, LLama, and Gemini. As for now, the descriptors can be accessed via https://anonymous.4open.science/r/xt124j05/descriptors/.

---

> ### Author Response · Authors · 2024-11-22
> **Follow Up on Author Response and Revised Manuscript**
>
> We would like to follow up on our response and revised manuscript. We would appreciate it if Reviewer UGi2 could look at our revised manuscript, and we offer additional comments.

---

> ### Comment · Area_Chair_6jbB · 2024-11-25
>
> Dear Reviewer UGi2,
>
> Could you kindly review the rebuttal thoroughly and let us know whether the authors have adequately addressed the issues raised or if you have any further questions.
>
> Best,
>
> AC of Submission6684

---

> ### Author Response · Authors · 2024-11-27
> **Second Follow Up**
>
> Dear Reviewer UGi2,
>
> We would like to follow up on our response. Could you please kindly review our updated manuscript regarding your concerns?
>
> Here are the pointers to your key points:
>
> W1. Proposed Method Overview: Section 4.1
>
> W1. Presentation Improvement: Please check our latest manuscript.
>
>
> W2. Step-by-step Ablation: Section 5.2.e.
>
>
> W3. The use of GPT:
>
> - LEAPGen-lite (lightweight generators and without descriptors) performance analysis in Section 5.2.d, Tables 2,3,4, and Figure 3 that proves the significant performance of our ideas is not bounded by LLM-generated descriptors or the complexity (parameters and size) of the generators.
>
> - Running time analysis in 5.2.g.
>
> - Detailed setting in Appendix E, LEAPGen and ConvPrompt use the same setting
>
> W4. Open the descriptors: Please check https://anonymous.4open.science/r/xt124j05/descriptors/.
>
>
> Thank you.
>
> Best Regards,
>
> Author of Submission 6684

---

> ### Author Response · Authors · 2024-11-29
> **Third Follow Up**
>
> Dear Reviewer UGi2,
>
> We would like to follow up on our response. Could you please kindly review our updated manuscript regarding your concerns? Thank you.
>
> Here are the pointers to your key points:
> We also copied the updates here, thus you can review them directly.
>
> W1. Proposed Method Overview: Section 4.1'
>
> In this study, we propose a novel LanguagE As Prompt Generator (LEAPGen) accommodating our
> main principles that are emphasized in the introduction section. The structure and flow of LEAPGen
> are visualized in figure 2. LEAPGen generators produce prompts from top-k selected embedding
> as input. LEAPGen also produces auxiliary (aux) data from the top-k embedding. The prompts are
> prepended into ViT MSAs while the aux is appended into input patches, thus producing feature and
> final prediction by ViT layers and MLP head respectively. The top-k embedding is selected based on
> the cosine similarities between an input and the class-wise keys. LEAPGen limits the search space
> into task 1 to predicted task t, by performing soft task-id prediction. In each task of the training
> phase, LEAPGen updates task-associated learnable parameters i.e. generator, task-wise key, and
> class-wise keys. In the inference phase, LEAPGen selects the generators based on the predicted task
> t. LEAPGen utilizes cross-entropy loss and cosine similarity loss to optimize its parameters. Task-
> wise generators, language embedding for prompt generation, and soft task-id prediction are unique
> to recent SOTAs of evolving generator methods e.g. (Roy et al., 2024) and (Kurniawan et al., 2024),
> task-wise fixed prompt methods such as (Wang et al., 2024a) and (Gao et al., 2024), and pool-based
> method (Wang et al., 2022c) in terms of prompt generation/selection, task-prediction mechanism,
> and modality for prompt generation. The detailed architecture, flow, and learning mechanism are
> presented in sub-section 4.2 and 4.3.
>
> W3. The use of GPT:
>
> - LEAPGen-lite (lightweight generators and without descriptors) performance analysis in Section 5.2.d, Tables 2,3,4, and Figure 3.
>
> | Method           | Split-CIFAR100   |                  |                 |                 | Split-ImageNet-R |                  |                 |                 |
> |------------------|------------------|------------------|-----------------|-----------------|------------------|------------------|-----------------|-----------------|
> |                  | FAA              | CAA              | FFM             | CFM             | FAA              | CAA              | FFM             | CFM             |
> |                  | 5 Tasks @20c     |                  |                 |                 | 5 Tasks @40c     |                  |                 |                 |
> | L2P              | 84.77 ± 0.48     | 88.67 ± 0.30     | 6.18 ± 0.57     | 5.99 ± 0.29     | 64.62 ± 0.32     | 68.01 ± 0.42     | 3.94 ± 0.16     | 3.55 ± 0.20     |
> | DualPrompt       | 86.41 ± 0.21     | 89.95 ± 0.10     | 5.37 ± 0.21     | 4.77 ± 0.46     | 69.71 ± 0.11     | 72.78 ± 0.14     | 3.32 ± 0.16     | 2.78 ± 0.25     |
> | CODA-P           | 88.22 ± 1.06     | 92.25 ± 1.28     | 7.05 ± 2.18     | 6.06 ± 2.66     | 74.89 ± 0.36     | 79.71 ± 1.27     | 8.89 ± 0.65     | 7.65 ± 0.98     |
> | LGCL             | 86.90 ± 0.40     | 90.45 ± 0.18     | 5.01 ± 0.35     | 4.36 ± 0.13     | 69.93 ± 0.21     | 72.91 ± 0.19     | 3.04 ± 0.36     | 2.50 ± 0.38     |
> | HiDe-Prompt      | 91.99 ± 0.03     | 93.95 ± 0.09     | 2.52 ± 0.18     | 2.33 ± 0.15     | 75.40 ± 0.27     | 78.88 ± 0.04     | 3.15 ± 0.46     | 2.64 ± 0.16     |
> | PGP              | 87.69 ± 0.06     | 91.26 ± 0.13     | 5.32 ± 0.18     | 4.60 ± 0.15     | 69.71 ± 0.15     | 72.77 ± 0.07     | 3.36 ± 0.23     | 2.85 ± 0.25     |
> | EvoPrompt        | 89.07 ± 0.38     | 92.32 ± 0.26     | 5.25 ± 0.65     | 5.39 ± 0.24     | 77.27 ± 0.40     | 81.67 ± 0.18     | 1.79 ± 0.31     | 1.41 ± 0.32     |
> | CPrompt          | 89.22 ± 0.05     | 93.09 ± 0.06     | 5.02 ± 0.17     | 4.31 ± 0.35     | 78.65 ± 0.00     | 82.44 ± 0.00     | 6.00 ± 0.00     | 5.49 ± 0.00     |
> | ConvPrompt       | 90.26 ± 0.44     | 93.49 ± 0.19     | 3.64 ± 0.28     | 3.25 ± 0.16     | 79.36 ± 0.08     | 82.93 ± 0.24     | 3.42 ± 0.05     | 2.36 ± 0.16     |
> | **LEAPGen-lite** | **97.07 ± 0.08** | **97.28 ± 0.10** | **0.05 ± 0.01** | **0.02 ± 0.01** | **82.44 ± 0.63** | **84.37 ± 0.90** | **0.43 ± 0.08** | **0.17 ± 0.06** |
> | **LEAPGen**      | **96.84 ± 0.12** | **96.85 ± 0.26** | **0.08 ± 0.07** | **0.06 ± 0.04** | **82.79 ± 0.32** | **85.06 ± 0.29** | **0.51 ± 0.04** | **0.18 ± 0.07** |

---

> ### Author Response · Authors · 2024-11-29
> **Third Follow Up (Cont'd)**
>
> - LEAPGen-lite (lightweight generators and without descriptors) performance analysis in Section 5.2.d, Tables 2,3,4, and Figure 3.
>
> | Method           |     CIFAR100     |                  |                 |                 |    ImageNet-R    |                  |                 |                 |
> |------------------|:----------------:|:----------------:|:---------------:|:---------------:|:----------------:|:----------------:|:---------------:|:---------------:|
> |                  |        FAA       |        CAA       |       FFM       |       CFM       |        FAA       |        CAA       |       FFM       |       CFM       |
> |                  |   10 Tasks @10c  |                  |                 |                 |   10 Tasks @20c  |                  |                 |                 |
> | L2P              |   83.84 ± 0.32   |   88.67 ± 0.16   |   6.55 ± 0.34   |   5.16 ± 0.14   |   62.50 ± 0.51   |   67.05 ± 0.47   |   5.01 ± 0.40   |   4.41 ± 0.43   |
> | DualPrompt       |   85.36 ± 0.20   |   89.77 ± 0.20   |   5.41 ± 0.33   |   4.33 ± 0.15   |   68.59 ± 0.24   |   72.18 ± 0.20   |   4.61 ± 0.07   |   3.70 ± 0.18   |
> | CODA-P           |   86.44 ± 0.16   |   91.27 ± 0.56   |   6.38 ± 1.46   |   5.09 ± 1.19   |   73.77 ± 0.50   |   79.38 ± 1.48   |   7.94 ± 0.08   |   6.72 ± 0.79   |
> | LGCL             |   85.68 ± 0.43   |   90.16 ± 0.29   |   5.46 ± 0.22   |   4.25 ± 0.32   |   68.65 ± 0.25   |   72.57 ± 0.19   |   4.75 ± 0.33   |   3.38 ± 0.58   |
> | HiDe-Prompt      |   92.89 ± 0.11   |   95.01 ± 0.08   |   1.98 ± 0.05   |   1.56 ± 0.15   |   75.75 ± 0.40   |   79.27 ± 0.17   |   2.29 ± 0.27   |   2.33 ± 0.17   |
> | PGP              |   86.36 ± 0.19   |   90.83 ± 0.17   |   5.49 ± 0.35   |   4.28 ± 0.27   |   68.62 ± 0.14   |   72.19 ± 0.20   |   4.53 ± 0.40   |   3.63 ± 0.35   |
> | EvoPrompt        |   88.17 ± 0.51   |   92.18 ± 0.49   |   5.39 ± 0.45   |   3.97 ± 0.73   |   76.00 ± 0.26   |   80.97 ± 0.30   |   4.22 ± 0.42   |   3.59 ± 0.52   |
> | CPrompt          |   86.92 ± 1.04   |   91.73 ± 0.66   |   5.43 ± 0.74   |   4.01 ± 0.81   |   76.32 ± 0.53   |   81.50 ± 0.30   |   6.10 ± 0.75   |   5.60 ± 1.35   |
> | ConvPrompt       |   88.77 ± 0.24   |   92.71 ± 0.04   |   4.12 ± 0.44   |   2.67 ± 0.11   |   77.08 ± 0.26   |   81.47 ± 0.10   |   4.17 ± 0.04   |   3.11 ± 0.17   |
> | **LEAPGen-lite** | **98.58 ± 0.03** | **98.69 ± 0.10** | **0.11 ± 0.03** | **0.06 ± 0.03** | **82.38 ± 1.04** | **85.14 ± 0.52** | **3.01 ± 1.19** | **2.13 ± 0.60** |
> | **LEAPGen**      | **98.38 ± 0.15** | **98.15 ± 0.39** | **0.10 ± 0.03** | **0.05 ± 0.00** | **84.09 ± 0.93** | **85.54 ± 0.65** | **1.46 ± 1.25** | **2.11 ± 1.21** |
> |                  |   20 Tasks @5c   |                  |                 |                 |   20 Tasks @10c  |                  |                 |                 |
> | L2P              |   81.89 ± 0.38   |   87.16 ± 0.33   |   8.81 ± 0.10   |   6.79 ± 0.33   |   57.40 ± 0.31   |   63.33 ± 0.21   |   10.76 ± 0.45  |   7.88 ± 0.17   |
> | DualPrompt       |   82.32 ± 0.22   |   87.47 ± 0.24   |   6.88 ± 0.35   |   5.63 ± 0.23   |   65.19 ± 0.17   |   70.31 ± 0.29   |   7.30 ± 0.18   |   5.16 ± 0.34   |
> | CODA-P           |   81.29 ± 0.16   |   87.72 ± 0.44   |   6.82 ± 1.60   |   4.98 ± 0.95   |   70.55 ± 0.71   |   77.08 ± 1.02   |   8.23 ± 0.86   |   6.95 ± 0.70   |
> | LGCL             |   83.18 ± 0.40   |   88.63 ± 0.18   |   7.22 ± 0.47   |   4.91 ± 0.56   |   64.96 ± 0.67   |   70.18 ± 0.37   |   7.35 ± 0.65   |   5.05 ± 0.32   |
> | HiDe-Prompt      |         -        |   97.62 ± 0.14   |        -        |   0.74 ± 0.03   |         -        |   81.60 ± 0.48   |        -        |   2.23 ± 0.38   |
> | PGP              |   83.41 ± 0.35   |   89.23 ± 0.13   |   7.95 ± 0.23   |   5.66 ± 0.22   |   65.24 ± 0.25   |   70.36 ± 0.26   |   7.17 ± 0.21   |   5.09 ± 0.25   |
> | EvoPrompt        |   84.63 ± 0.21   |   89.47 ± 0.21   |   9.19 ± 0.41   |   7.39 ± 0.65   |   74.93 ± 0.64   |   79.92 ± 0.13   |   6.72 ± 0.90   |   5.67 ± 0.26   |
> | CPrompt          |   83.60 ± 0.00   |   90.10 ± 0.00   |   6.47 ± 0.00   |   4.78 ± 0.00   |   74.23 ± 0.17   |   79.82 ± 0.51   |   5.98 ± 0.24   |   5.54 ± 0.48   |
> | ConvPrompt       |   87.21 ± 0.20   |   91.60 ± 0.36   |   5.47 ± 0.33   |   3.92 ± 0.33   |   73.93 ± 0.36   |   78.92 ± 0.37   |   4.87 ± 0.57   |   3.57 ± 0.25   |
> | **LEAPGen-lite** | **95.28 ± 3.37** | **98.38 ± 1.13** | **1.08 ± 1.54** | **0.70 ± 0.99** | **83.67 ± 0.39** | **85.65 ± 0.33** | **1.06 ± 0.24** |  **0.47± 0.14** |
> | **LEAPGen**      | **96.51 ± 2.16** | **98.73 ± 0.26** | **0.66 ± 0.75** | **0.52 ± 0.39** | **87.03 ± 0.12** | **87.81 ± 0.48** | **2.17 ± 0.17** | **2.54 ± 0.77** |

---

> ### Author Response · Authors · 2024-11-29
> **Third Follow Up (Cont'd)**
>
> - LEAPGen-lite (lightweight generators and without descriptors) performance analysis in Section 5.2.d, Tables 2,3,4, and Figure 3.
>
>
> | Method           | CUB 10-Tasks @20c |                  |                 |                 |
> |------------------|-------------------|------------------|-----------------|-----------------|
> |                  | FAA               | CAA              | FFM             | CFM             |
> | L2P              | 66.95 ± 0.13      | 74.03 ± 0.32     | 5.18 ± 0.11     | 7.39 ± 0.23     |
> | DualPrompt       | 73.95 ± 0.73      | 80.29 ± 0.15     | 7.87 ± 0.76     | 8.60 ± 0.37     |
> | CODA-P           | 72.99 ± 0.30      | 82.64 ± 0.64     | 11.71 ± 1.49    | 9.91 ± 1.04     |
> | LGCL             | 79.93 ± 0.30      | 83.07 ± 0.23     | 5.45 ± 0.33     | 5.58 ± 0.39     |
> | HiDe-Prompt      | 87.21 ± 0.18      | 87.66 ± 0.01     | 1.90 ± 0.45     | 2.66 ± 0.13     |
> | PGP              | 78.35 ± 0.68      | 82.21 ± 0.56     | 5.76 ± 0.10     | 6.10 ± 0.26     |
> | EvoPrompt        | 76.23 ± 0.51      | 81.00 ± 0.40     | 3.96 ± 0.43     | 3.57 ± 0.83     |
> | CPrompt          | 77.14 ± 1.16      | 85.67 ± 0.56     | 11.65 ± 0.47    | 8.69 ± 0.31     |
> | ConvPrompt       | 80.12 ± 1.37      | 84.70 ± 0.64     | 6.04 ± 0.97     | 4.61 ± 0.45     |
> | **LEAPGen-lite** | **86.00 ± 0.32**  | **88.03 ± 0.47** | **2.46 ± 1.61** | **2.29 ± 1.15** |
> | **LEAPGen**      | **88.45 ± 0.58**  | **90.90 ± 0.81** | **1.32 ± 0.52** | **1.29 ± 0.82** |
>
>
>
> Compared to other approaches
>
> | Method           | ImageNet-R 5T    | ImageNet-R 5T    | ImageNet-R 10T   | ImageNet-R 10T   | ImageNet-R 20T   | ImageNet-R 20T   | CIFAR100 10T     | CIFAR100 10T     |
> |------------------|------------------|------------------|------------------|------------------|------------------|------------------|------------------|------------------|
> |                  | FAA              | CAA              | FAA              | CAA              | FAA              | CAA              | FAA              | CAA              |
> | C-LoRA           | 75.85 ± 0.31     | 78.85 ± 0.34     | 71.89 ± 0.45     | 75.33 ± 0.28     | 65.71 ± 0.60     | 70.63 ± 0.85     | 82.97 ± 0.47     | 87.06 ± 0.25     |
> | LAE              | 73.84 ± 0.14     | 77.29 ± 0.45     | 71.70 ± 0.39     | 76.71 ± 0.10     | 66.98 ± 0.35     | 73.72 ± 0.05     | 88.81 ± 0.34     | 91.59 ± 0.13     |
> | InfLoRA          | 77.52 ± 0.37     | 82.01 ± 0.12     | 75.65 ± 0.14     | 80.82 ± 0.24     | 71.01 ± 0.45     | 77.28 ± 0.45     | 89.84 ± 0.03     | 91.70 ± 0.32     |
> | SLCA             | -                | -                | 77.00 ± 0.33     | 81.17 ± 0.64     | -                | -                | 91.53 ± 0.28     | 94.09 ± 0.87     |
> | GMM              | -                | -                | 80.72            | -                | -                | -                | 87.59            | -                |
> | **LEAPGen-lite** | **82.44 ± 0.63** | **84.37 ± 0.90** | **82.38 ± 1.04** | **85.14 ± 0.52** | **83.67 ± 0.39** | **85.65 ± 0.33** | **98.58 ± 0.03** | **98.69 ± 0.10** |
> | **LEAPGen**      | **82.79 ± 0.32** | **85.06 ± 0.29** | **84.09 ± 0.93** | **85.54 ± 0.65** | **87.03 ± 0.12** | **87.81 ± 0.48** | **98.38 ± 0.15** | **98.15 ± 0.39** |
>
>
> 5.2.d) LEAPGen-lite’s Performance: As shown in Table 2-4 and Figure 3 Despite utilizing far smaller
> (2.67% params) generators and without class descriptors generated by LLM, LEAPGen-lite still
> outperforms the existing methods significantly i.e. 4.7-25% FAA and 3-11% CAA in CIFAR100,
> and 3-30% FAA and 2-26% CAA in ImageNet-R dataset. LEAPGen-lite also achieves a low for-
> getting rate in these 2 datasets for all task-settings. In the CUB dataset, LEAPGen-lite archives a
> comparable performance to HiDe-Prompt and outperforms other SOTA with a significant margin i.e.
> 6-20% FAA and 3.3-14% CAA. This evidence proves our ideas i.e. language embedding as input
> for prompt generation, task-wise generators, soft task-id predictor, and learning with auxiliary data
> are not bounded by the generated descriptors and the size of generators.
>
> **Please Note:  The analysis of LEAGen-lite empirically proves that our method doesn't rely on GPT/LLM or complex generator networks (high number of parameters)**

---

> ### Author Response · Authors · 2024-11-29
> **Third Follow Up (Cont'd)**
>
> -  Running time analysis in 5.2.g.
>
> | Method       | Desc | #Params(M) | Running Time (h) |       |        |       | Storage (MB) |
> |--------------|:----:|:----------:|:----------------:|:-----:|:------:|:-----:|:------------:|
> |              |      |            |      T.Desc      |  Inf  | Tr+Inf | Total |              |
> | HiDe-Prompt  |   -  |    0.15    |         -        | 0.019 |  5.40  |  5.40 |      334     |
> | ConvPrompt   |   v  |    1.28    |       1.07       | 0.033 |  1.04  |  2.11 |      346     |
> | LEAPGen-lite |   -  |    0.16    |         -        | 0.028 |  0.53  |  0.53 |      332     |
> | LEAPGen      |   v  |    6.35    |       1.07       | 0.025 |  0.72  |  1.79 |      567     |
>
> T.Desc, Tr, and Inf denote time for generating descriptors, training, and inference respectively, detailed in Appendix D3.
>
> 5.2.g) Parameters, Running Time, and Storage: The table above compares the number of parameters, running time (ImageNet-R), and storage of our methods and existing SOTAs. Despite having a higher number of parameters and storage, LEAPGen consumes less running time than existing SOTAs both training+inference and total running time. LEAPGen-lite consumes the least costs in total running time and storage and requires relatively low parameters and inference time. LEAPGen and ConvPrompt require additional time to generate descriptors that increase their total simulation time. Despite having the least parameters, Hide-Prompt requires the longest training and total times since it needs extra operations to generate uninstructed class representations.
>
> - Detailed setting in Appendix E, LEAPGen and ConvPrompt use the same setting
>
> E DETAILED E XPERIMENTAL S ETTING
>
> Existing Methods: The existing SOTAs are run by executing the official implementation (code)
> of the respective methods. The hyper-parameters setting is chosen based on the official setting.
> HiDe-Prompt utilizes S-Prompt(Wang et al., 2022a) i.e. similar to DualPrompt but without the
> global (task-shared) prompt. LGCL and PGP don’t propose a specific prompt structure but rather
> utilize L2P and DualPrompt. The reported results for PGP and LGCL are obtained with DualPrompt structure which is the best from their result. The other methods i.e. L2P, DualPrompt, CODA-P,
> EvoPrompt, CPrompt, and ConvPrompt utilize their proposed structure. All the evaluated methods
> utilize ViT B/16 pre-trained on ImageNet 21K as the backbone model. LGCL utilizes pre-trained
> CLIP text encoder, while ConvPrompt utilizes SentenceTransformer pre-trained on BERT as its text
> encoder.
>
> LEAPGen: Our proposed method is implemented on top of the ViT backbone pre-trained on
> ImageNet-21K. The prompt structure is as defined in section 4. The prompt length is set to 30,
> and the prefix tuning layers is set to 7 i.e. [0,1,2,3,4,5,6] for all main experiments. We utilize Adam
> optimizer with a cosine learning rate scheduler. For CIFAR100 dataset, We set 0.01 initial learning
> rate and 3, 10, and 10 epochs for 5-task, 10-task, and 20-task settings respectively. For ImageNet-
> R dataset, We choose 5, 10, and 20 epochs for 5-task, 10-task, and 20-task settings respectively.
> The initial learning rate is chosen from the best of 0.04,0.05,0.06. For CUB dataset, We choose 20
> epochs and 0.005 initial learning rate. Similar to ConvPrompt, we utilize SentenceTransformer as
> a text encoder. All the pre-trained models i.e. ViT, SentenceTransformer, and CLIP(LGCL) are kept
> frozen (not fine-tuned).
>
> W4. Open the descriptors: Please check https://anonymous.4open.science/r/xt124j05/descriptors/.

---

> ### Author Response · Authors · 2024-11-29
> **Third Follow Up (Cont'd)**
>
> W2. Step-by-step Ablation: Section 5.2.e.
>
> | Component                              | Loss                    |  FAA  |  CAA  |  FFM |  CFM |
> |-----------------------------------------|-------------------------|:-----:|:-----:|:----:|:----:|
> | FT                                      | $\mathcal{L}$_intra                 | 61.53 | 65.31 | 5.87 | 5.62 |
> | FT+E                                    | $\mathcal{L}$_intra                  | 72.93 | 78.54 | 5.78 | 5.30 |
> | FT+E+K                                  | $\mathcal{L}$_intra +$\mathcal{L}_t$              | 74.88 | 79.52 | 2.84 | 3.17 |
> | FT+G(E)+K                               | $\mathcal{L}$_intra +$\mathcal{L}_t$             | 76.64 | 80.48 | 1.85 | 2.44 |
> | FT+G(E)+K+L                             | $\mathcal{L}$_intra +$\mathcal{L}_t$+$\mathcal{L}_c$         | 77.23 | 81.05 | 0.50 | 1.27 |
> | FT+G(E)+K+L+Aux                         | $\mathcal{L}$_intra +$\mathcal{L}_t$+$\mathcal{L}_c$         | 78.38 | 81.98 | 0.49 | 1.15 |
> | FT+G(E)+K+L+Aux                         | $\mathcal{L}$_intra +$\mathcal{L}$_inter +$\mathcal{L}_t$+$\mathcal{L}_c$ | 83.70 | 85.98 | 2.46 | 1.93 |
> | FT+G(E)+K+L+Aux+ Soft Task-ID Predictor | $\mathcal{L}$_intra +$\mathcal{L}$_inter+$\mathcal{L}_t$+$\mathcal{L}_c$ | 84.73 | 86.14 | 0.91 | 1.42 |
>
> e) Ablation Study: yable above shows the impacts of the proposed method's components on its performance. \textbf{ (1) Descriptor Embedding $E$} as one of LEAPGen's main components, elevates fine tuning (FT) baseline significantly i.e. $>11\%$ FAA and $>13\%$ CAA. It shows the promising impact of our idea . (2) Task Key $K^t$ and Loss $\mathcal{L}_t$ transform the method into task-wise prompting based on input to $K^t$ and $\mathcal{L}_t$ similarity. They produce a better model with 1-2\% FAA and CAA improvement and reduce FFM and CFM by $>2\%$. It shows the effectiveness of task-wise decomposition as applied in the task-wise prompt approach.
> (3) Task-wise generator $G^t()$ enhances the model recognition capability by $2\%$ and $1\%$ for FAA and CAA respectively. It also decreases its forgetting rate by 1\%. It means the trainable generator produces a more discriminative prompt than the descriptor embedding only.
> (4) $L^t_c$ Class-wise Key $L^t_c$ and Loss $\mathcal{L}_c$ for top-k embedding selection i.e. $E_i$ for $i \in [1...k]$ improves the model performance with 1\% margin. It also reduces the model forgetting rate with up to 1\% margin. Thus, measuring the input similarity to $L^t_c$ offers a better embedding selection than to$E^t_c$ directly.
> (5) Auxiliary} embedding improves the LEAPGen's performance with a more than $1\%$ margin. It shows the secondary contribution of descriptors embedding as an auxiliary modality along with its primary role in prompt generation.
> 6) Inter-tasks loss $\mathcal{L}$_inter improves LEAPGen accuracy significantly i.e. $5.4\%$ and $4.0\%$ for FAA and CAA respectively even though it increases FFM and CFM by $0.5-2\%$. It contributes to balancing the knowledge of previously learned classes (stability) and currently learned classes (plasticity), and emphasizes the risk of forgetting previously learned classes.
> Soft task-id predictor} substitution to conventional task-id (as in DualPrompt) improves the accuracy with up to $1\%$ and reduces FFM and CFM by $1.5\%$ and $0.5\%$ respectively. It implies that our designed task-id predictor outperforms the existing task-id predictor both in accuracy and forgetting.
>
> Thank you.
>
> Best Regards,
>
> Author of Submission 6684

---

> ### Author Response · Authors · 2024-12-01
> **Fourth Follow Up**
>
> Dear Reviewer UGi2,
>
> We would like to follow up on our responses. Could you please kindly review our updated manuscript or previous comments regarding your concerns?
>
> We believe we have addressed all your concerns in our updated paper and previous responses (comments). Thus, we kindly request Reviewer UGi2 to reevaluate the score for our paper.
>
> Thank you.
>
> Best Regards,
>
> Author of Submission 6684

---

> ### Author Response · Authors · 2024-12-04
> **Summary of Revision and Response**
>
> Dear Reviewer UGi2,
>
> First, we thank reviewer UGi2 for your effort and time to review our work.
> We believe we have addressed all your concerns in our revised paper and previous responses (comments)
> Second, since the discussion phase has ended, We would like to summarize our revision and responses, and emphasize the following points:
>
> 1. **Revision of Ablation**: We have revised our ablation study in a step-by-step as you advised. It explains the contribution of each component in a more detailed way (sec. 5.2.e).
>
> 2. **Method Overview and Writing Improvement**: We have added an overview explaining the general ideas of our proposed method in sec 4.1. We have improved the writing of our paper in terms of layout, presentation and clarity.
>
> 3. **Fairness Setting** :   LEAPGen utilizes the same resources as ConvPrompt and LEAPGen-lite(no descriptors with lightweight generators) utilizes less resources than ConvPrompt and comparable resources to existing SOTAs (sec. 5.1, sec. 5.2.g, Appendix E). All methods are evaluated in the same setting and run with respective best (recommended by official paper/code) hyperparameters. Thus we believe we have satisfied the fairness aspect in our study.
>
> 3. **The presence of GPT**  :
>
> - **GPT is not part of our method** (sec 4.2.a, and fig. 2), It is an optional part to generate descriptors following ConvPrompt. Even without GPT-generated descriptors. our method works excellently and significantly outperforms the existing SOTAs, as demonstrated by LEAPGen-lite (sec. 5.2.d) and LEAPGen-CN (Appendix D.4).
>
> - **Descriptor impact** to the model performance has been demonstrated by the performance difference between LEAPGen and LEAPGen-CN (Appendix D.4). We also extend our analysis of the performance of LEAPGen vs ConvPrompt in various types of descriptors (Appendix D.5).
>
> - **LEAPGen-lite** proves the significant impacts of our ideas i.e. (1) language as input for prompt generation, (2) task-wise generators, (3) soft task-id predictor, and (4)learning with auxiliary despite without GPT-generated descriptors, and utilize least parameters, running time and storage.
>
> We believe, these highlights resolve your concerns and prove the significant impacts of our ideas, without being bound to the presence of GPT.
>
> Best Regards,
>
> Author of Submission 6684

---

### Official Review · Reviewer_J5sH · 2024-11-03

**Soundness:** 3
**Presentation:** 4
**Contribution:** 3
**Rating:** 6
**Confidence:** 4

**Summary:**

This paper proposes a novel method LEAPGen (Language as Prompt Generator) for continual learning that leverages language knowledge for prompt generation.
LEAPGen uses language descriptors for each class as input for prompt generation, rather than shared embeddings. For each task, LEAPGen employs a fixed number of generator networks, a learnable task-level key, and learnable class-level parameters associated with the descriptor embeddings. To generate a prompt, it first predicts the task ID using a soft prediction mechanism, then selects top-k matching descriptor embeddings based on cosine similarity to the input. Experimental results on datasets like CIFAR100, ImageNet-R, and CUB demonstrate that LEAPGen significantly outperforms existing state-of-the-art methods in terms of accuracy and forgetting measure, addressing key limitations of previous prompt-based continual learning approaches.

**Strengths:**

- LEAPGen uses language descriptors as input for prompt generation, leveraging richer semantic information compared to previous methods.

- As a prompt-based method, it doesn't require storing and replaying old data.

- By using fixed language descriptors as input, LEAPGen potentially reduces catastrophic forgetting compared to methods that continuously update shared embeddings.

- Experimental results show significant improvements over state-of-the-art methods.

**Weaknesses:**

- The paper does not explicitly specify some key details about the pre-trained models used. Which Sentence Transformer does the paper employ? Whether this is a pre-trained language model and if any fine-tuning of these models occurs during the continual learning process.

- The paper does not provide a clear comparison of the increase in parameters due to the addition of generators for new tasks. Based on the paper, the generator is not a lightweight module. As the proposed method requires adding new generators for some new tasks, even if not all tasks, this could lead to a significant growth in the number of parameters as the number of tasks increases. And also an unfair comparison with related works.

**Questions:**

see above

---

> ### Author Response · Authors · 2024-11-22
> **Autrhor Response to Reviewer J5sH**
>
> Q1/W1. The paper does not explicitly specify some key details about the pre-trained models used. Which Sentence Transformer does the paper employ? Whether this is a pre-trained language model and if any fine-tuning of these models occurs during the continual learning process.
>
> Our Response: Thank you for your concern.
>
> - Following common practice in prompt-based methods, all the evaluated methods in tables 2 and 3 utilize the same model i.e. ViT-B/16 that is pre-trained on imagenet21K. Following ConvPrompt, we utilized SentenceTransformer pre-trained on Phase-BERT as our text encoder. Other language-guided methods i.e. LGCL and GMM utilize CLIP as a text encoder. There is no fine-tuning process for SentenceTransformer in our method. The model is solely employed to encode the descriptors into embedding before starting a training process in each task.
>
> - We explain the detailed setting for all method in Appendix E.
>
>
> Q2/W2. The paper does not provide a clear comparison of the increase in parameters due to the addition of generators for new tasks. Based on the paper, the generator is not a lightweight module. As the proposed method requires adding new generators for some new tasks, even if not all tasks, this could lead to a significant growth in the number of parameters as the number of tasks increases. And also an unfair comparison with related works.
>
> Our Response: Thank you for your concern.
>
> - We added a new experiment and analysis on the trade-offs between cost and performance that comprehensively evaluate the proposed methods and existing SOTAs in terms of performance (accuracy and forgetting), number of parameters, and running time. We also compare our proposed method with the large version of SOTAs that executes a higher number of parameters. Please see Appendix D3. Our analysis shows in comparison to the standard versions and large versions of SOTAs, our method achieves better performance (average accuracy and average forgetting), and requires a shorter running time.
>
> - Even in comparison with the SOTAs in a smaller number of parameters, our proposed method still achieves a smaller running time. The Smallest number of parameters i.e. HiDe-Prompt doesn't imply shorter running time as It requires mode additional operations i.e. drawing (augmenting) prototypes and constructing uninformed representations that run on many epochs.
>
> - To ensure fairness, we evaluated all the methods by applying the best setting i.e. as mentioned in the references and official code manual, including the preferred hyper-parameters. Please note that increasing the number of parameters i.e. by increasing the number of generators, or prompt length doesn't always improve the model performance. As shown in the table above, our sensitivity analysis in sec 5.2. point e and figure 4, after some points, the increase in prompt length or number of generators decreases the model performance.
> Thus, we believe, we have evaluated the consolidated methods in a fair way i.e. by applying their respective official setting and parameters.

---

> ### Author Response · Authors · 2024-11-22
> **Follow Up on Author Response and Revised Manuscript**
>
> We would like to follow up on our response and revised manuscript. We would appreciate it if Reviewer j5sH could look at our revised manuscript, and offer additional comments.

---

> ### Comment · Area_Chair_6jbB · 2024-11-25
>
> Dear Reviewer J5sH,
>
> Could you kindly review the rebuttal thoroughly and let us know whether the authors have adequately addressed the issues raised or if you have any further questions.
>
> Best,
>
> AC of Submission6684

---

> ### Author Response · Authors · 2024-11-27
> **Second Follow Up**
>
> Dear Reviewer J5sH,
>
> We would like to follow up on our response to your comments Could you please kindly check our revised manuscript?
>
> Here are the pointers to your key points:
>
> W1. Details of ViT and Text Encoder: Appendix E. Detailed Experimental Setting
>
> W2. Increase of Parameters:
>
> - Section 5.2.g. Analysis of Parameters, Running Time and Storage
>
> - Appendix D.3. TradeOffs Between Cost and Performance including detailed growing parameters on each task
>
> - Section 5.2.d., Tables 2,3,4, and Figure 3 i.e. Analysis of LEAPGen-lite (lightweight generators and without descriptors) performance that proves the significant performance of our ideas is not bounded by LLM-generated descriptors or the complexity (parameters and size) of the generators.
>
> Thank you.
>
> Best Regards,
>
> Authors of Submission 6684

---

> ### Author Response · Authors · 2024-11-29
> **Third Follow Up**
>
> Dear Reviewer J5sH,
>
> We would like to follow up on our response. Could you please kindly review our latest manuscript regarding your concern?
> Thank you.
>
> Here are the pointers to your key points:
>
> We also copied the updates (or snapshots) here, thus you can review them directly.
>
> W1. Details of ViT and Text Encoder: Appendix E. Detailed Experimental Setting
>
> All the evaluated methods utilize ViT B/16 pre-trained on ImageNet 21K as the backbone model. LGCL utilizes pre-trained
> CLIP text encoder, while ConvPrompt utilizes SentenceTransformer pre-trained on BERT as its text
> encoder.
> LEAPGen: Our proposed method is implemented on top of the ViT backbone pre-trained on
> ImageNet-21K. The prompt structure is as defined in section 4. The prompt length is set to 30,
> and the prefix tuning layers is set to 7 i.e. [0,1,2,3,4,5,6] for all main experiments. We utilize Adam
> optimizer with a cosine learning rate scheduler. For CIFAR100 dataset, We set 0.01 initial learning
> rate and 3, 10, and 10 epochs for 5-task, 10-task, and 20-task settings respectively. For ImageNet-
> R dataset, We choose 5, 10, and 20 epochs for 5-task, 10-task, and 20-task settings respectively.
> The initial learning rate is chosen from the best of 0.04,0.05,0.06. For CUB dataset, We choose 20
> epochs and 0.005 initial learning rate. Similar to ConvPrompt, we utilize SentenceTransformer as
> text encoder. All the pre-trained models i.e. ViT, SentenceTransformer and CLIP(LGCL) are kept
> frozen (not fine-tuned).
>
>
> W2. Increase of Parameters:
>
> - Section 5.2.g: Analysis of parameters, running time, and storage
>
> | Method       | Desc | #Params(M) | Running Time (h) |       |        |       | Storage (MB) |
> |--------------|:----:|:----------:|:----------------:|:-----:|:------:|:-----:|:------------:|
> |              |      |            |      T.Desc      |  Inf  | Tr+Inf | Total |              |
> | HiDe-Prompt  |   -  |    0.15    |         -        | 0.019 |  5.40  |  5.40 |      334     |
> | ConvPrompt   |   v  |    1.28    |       1.07       | 0.033 |  1.04  |  2.11 |      346     |
> | LEAPGen-lite |   -  |    0.16    |         -        | 0.028 |  0.53  |  0.53 |      332     |
> | LEAPGen      |   v  |    6.35    |       1.07       | 0.025 |  0.72  |  1.79 |      567     |
>
> T.Desc, Tr, and Inf denote time for generating descriptors, training, and inference respectively, detailed in Appendix D3.
>
> 5.2.g) Parameters, Running Time, and Storage: The table above compares the number of parameters, running time (ImageNet-R), and storage of our methods and existing SOTAs. Despite having a higher number of parameters and storage, LEAPGen consumes less running time than existing SOTAs both training+inference and total running time. LEAPGen-lite consumes the least costs in total running time and storage and requires relatively low parameters and inference time. LEAPGen and ConvPrompt require additional time to generate descriptors that increase their total simulation time. Despite having the least parameters, Hide-Prompt requires the longest training and total times since it needs extra operations to generate uninstructed class representations.

---

> ### Author Response · Authors · 2024-11-29
> **Third Follow Up**
>
> W2. Increase of Parameters:
>
> - Appendix D.3. TradeOffs Between Cost and Performance including detailed growing parameters on each task
>
> (1). Performance vs #Parameters and Running Time:
>
> | Method            |     Metrics     | Value |       |       |       |       |       |       |       |       |       |  AVG  | RunTime(h) |
> |-------------------|:---------------:|:-----:|:-----:|:-----:|:-----:|:-----:|:-----:|:-----:|:-----:|:-----:|:-----:|:-----:|:----------:|
> |                   |                 |   1   |   2   |   3   |   4   |   5   |   6   |   7   |   8   |   9   |   10  |       |            |
> | HiDe-Prompt       |   Avg.Accuracy  | 85.22 | 82.93 | 82.35 | 79.74 | 78.85 | 77.81 | 77.37 | 76.38 | 76.31 | 75.75 | 79.27 |      -     |
> | HiDe-Prompt-Large |   Avg.Accuracy  | 85.76 | 83.12 | 82.03 | 79.49 | 78.55 | 76.55 | 76.00 | 74.82 | 74.98 | 74.30 | 78.56 |      -     |
> | ConvPrompt        |   Avg.Accuracy  | 89.53 | 86.24 | 84.40 | 81.88 | 80.79 | 80.04 | 78.80 | 78.15 | 77.74 | 77.08 | 81.47 |      -     |
> | ConvPrompt-Large  |   Avg.Accuracy  | 90.12 | 86.22 | 83.96 | 81.45 | 80.52 | 79.73 | 76.40 | 74.19 | 74.56 |   -   | 80.79 |      -     |
> | LEAPGen-lite      |   Avg.Accuracy  | 89.73 | 88.66 | 86.94 | 86.43 | 86.04 | 83.03 | 82.75 | 82.52 | 82.94 | 82.38 | 85.14 |      -     |
> | LEAPGen           |   Avg.Accuracy  | 89.83 | 87.63 | 86.46 | 85.97 | 84.71 | 84.79 | 84.01 | 83.77 | 84.12 | 84.09 | 85.54 |      -     |
> | HiDe-Prompt       |  Avg.Forgetting |  3.29 |  1.50 |  2.41 |  2.19 |  2.53 |  2.20 |  2.25 |  2.30 |  2.29 |  2.33 |  2.33 |      -     |
> | HiDe-Prompt-Large |  Avg.Forgetting |   -   |  3.05 |  1.89 |  2.14 |  2.15 |  3.25 |  2.91 |  2.93 |  2.66 |  2.70 |  2.63 |      -     |
> | ConvPrompt        |  Avg.Forgetting |   -   |  2.57 |  1.57 |  2.60 |  2.93 |  3.15 |  3.53 |  3.44 |  4.07 |  4.17 |  3.11 |      -     |
> | ConvPrompt-Large  |  Avg.Forgetting |   -   |  4.65 |  3.39 |  4.62 |  4.69 |  5.56 |  8.91 | 10.55 | 10.50 |   -   |  6.61 |      -     |
> | LEAPGen-lite      |  Avg.Forgetting |  1.11 |  0.75 |  0.58 |  0.56 |  4.13 |  3.37 |  2.92 |  2.70 |  3.01 |  2.13 |  2.13 |      -     |
> | LEAPGen           |  Avg.Forgetting |   -   |  3.97 |  1.96 |  1.38 |  2.63 |  2.24 |  1.99 |  1.77 |  1.60 |  1.46 |  2.11 |      -     |
> | HiDe-Prompt       | #Parameters (M) |  0.15 |  0.15 |  0.15 |  0.15 |  0.15 |  0.15 |  0.15 |  0.15 |  0.15 |  0.15 |  0.15 |    5.40    |
> | HiDe-Prompt-Large | #Parameters (M) | 12.29 | 12.29 | 12.29 | 12.29 | 12.29 | 12.29 | 12.29 | 12.29 | 12.29 | 12.29 | 12.29 |    15.15   |
> | ConvPrompt        | #Parameters (M) |  0.55 |  0.60 |  0.70 |  0.80 |  0.85 |  0.89 |  0.99 |  1.09 |  1.19 |  1.28 |  0.89 |    2.11    |
> | ConvPrompt-Large  | #Parameters (M) |  5.18 |  5.82 |  6.65 |  7.28 |  7.91 |  8.60 |  9.81 | 10.94 | 12.01 | 13.23 |  8.74 |    15.28   |
> | LEAPGen-lite      | #Parameters (M) |  0.02 |  0.03 |  0.05 |  0.07 |  0.08 |  0.10 |  0.11 |  0.13 |  0.15 |  0.16 |  0.09 |    0.53    |
> | LEAPGen           | #Parameters (M) |  6.21 |  6.23 |  6.24 |  6.26 |  6.27 |  6.29 |  6.31 |  6.32 |  6.34 |  6.35 |  6.28 |    1.79    |
>
> (2). Detailed Running Time: T.Desc, Tr, and Inf denote time for generating descriptors, training, and inference respectively.
>
> | Method       |   Time (h)   |       |        |       |                |       |        |       |         |       |        |       |
> |--------------|:------------:|:-----:|:------:|:-----:|:--------------:|:-----:|:------:|:-----:|:-------:|:-----:|:------:|:-----:|
> |              | CIFAR100 10T |       |        |       | ImageNet-R 10T |       |        |       | CUB 10T |       |        |       |
> |              |    T.Desc.   |  Inf  | Tr+Inf | Total |     T.Desc.    |  Inf  | Tr+Inf | Total | T.Desc. |  Inf  | Tr+Inf | Total |
> | HiDe-Prompt  |       -      | 0.037 |  4.63  |  4.63 |        -       | 0.019 |  5.40  |  5.40 |    -    | 0.017 |  3.94  |  3.94 |
> | ConvPrompt   |     0.53     | 0.023 |  2.01  |  2.54 |      1.07      | 0.033 |  1.04  |  2.11 |   1.09  | 0.035 |  8.08  |  9.17 |
> | LEAPGen-lite |       -      | 0.016 |   1.2  |  1.2  |        -       | 0.028 |  0.53  |  0.53 |    -    | 0.028 |  0.38  |  0.38 |
> | LEAPGen      |     0.53     | 0.017 |  0.98  |  1.51 |      1.07      | 0.025 |  0.72  |  1.79 |   1.09  | 0.025 |  0.32  |  1.41 |
>
> (3). Performance vs Storage:
>
> | Method            | Performance |       | Storage(MB) |
> |-------------------|:-----------:|:-----:|:-----------:|
> |                   |     FAA     |  CFA  |             |
> | HiDe-Prompt       |    75.75    | 79.27 |     334     |
> | HiDe-Prompt-Large |    74.30    | 78.56 |     797     |
> | ConvPrompt        |    77.08    | 81.47 |     346     |
> | ConvPrompt-Large  |    74.56*   | 80.79 |     632     |
> | LEPGen-lite       |    82.38    | 85.14 |     332     |
> | LEAPGen           |    84.09    | 85.54 |     567     |

---

> ### Author Response · Authors · 2024-11-29
> **Third Follow Up (Cont'd)**
>
> W2. Increase of Parameters:
> - Section 5.2.d., Tables 2,3,4, and Figure 3 i.e. Analysis of LEAPGen-lite (lightweight generators and without descriptors) performance
>
> 5.2.d) LEAPGen-lite’s Performance: As shown in Table 2-4 and Figure 3 Despite utilizing far smaller
> (2.67% params) generators and without class descriptors generated by LLM, LEAPGen-lite still
> outperforms the existing methods significantly i.e. 4.7-25% FAA and 3-11% CAA in CIFAR100,
> and 3-30% FAA and 2-26% CAA in ImageNet-R dataset. LEAPGen-lite also achieves a low for-
> getting rate in these 2 datasets for all task-settings. In the CUB dataset, LEAPGen-lite archives a
> comparable performance to HiDe-Prompt and outperforms other SOTA with a significant margin i.e.
> 6-20% FAA and 3.3-14% CAA. This evidence proves our ideas i.e. language embedding as input
> for prompt generation, task-wise generators, soft task-id predictor, and learning with auxiliary data
> are not bounded by the generated descriptors and the size of generators.
>
> Here is the snapshot of the tables, please see our revised paper for the full tables.
>
> | **Method**       | **ImageNet-R**   |                  |                 |                 |
> |------------------|------------------|------------------|-----------------|-----------------|
> |                  | **FAA**          | **CAA**          | **FFM**         | **CFM**         |
> |                  | 5 Tasks @40c     |                  |                 |                 |
> | L2P              | 64.62 ± 0.32     | 68.01 ± 0.42     | 3.94 ± 0.16     | 3.55 ± 0.20     |
> | DualPrompt       | 69.71 ± 0.11     | 72.78 ± 0.14     | 3.32 ± 0.16     | 2.78 ± 0.25     |
> | CODA-P           | 74.89 ± 0.36     | 79.71 ± 1.27     | 8.89 ± 0.65     | 7.65 ± 0.98     |
> | LGCL             | 69.93 ± 0.21     | 72.91 ± 0.19     | 3.04 ± 0.36     | 2.50 ± 0.38     |
> | HiDe-Prompt      | 75.40 ± 0.27     | 78.88 ± 0.04     | 3.15 ± 0.46     | 2.64 ± 0.16     |
> | PGP              | 69.71 ± 0.15     | 72.77 ± 0.07     | 3.36 ± 0.23     | 2.85 ± 0.25     |
> | EvoPrompt        | 77.27 ± 0.40     | 81.67 ± 0.18     | 1.79 ± 0.31     | 1.41 ± 0.32     |
> | CPrompt          | 78.65 ± 0.00     | 82.44 ± 0.00     | 6.00 ± 0.00     | 5.49 ± 0.00     |
> | ConvPrompt       | 79.36 ± 0.08     | 82.93 ± 0.24     | 3.42 ± 0.05     | 2.36 ± 0.16     |
> | **LEAPGen-lite** | **82.44 ± 0.63** | **84.37 ± 0.90** | **0.43 ± 0.08** | **0.17 ± 0.06** |
> | **LEAPGen**      | **82.79 ± 0.32** | **85.06 ± 0.29** | **0.51 ± 0.04** | **0.18 ± 0.07** |
> |                  | ImageNet-R       |                  |                 |                 |
> | L2P              | 62.50 ± 0.51     | 67.05 ± 0.47     | 5.01 ± 0.40     | 4.41 ± 0.43     |
> | DualPrompt       | 68.59 ± 0.24     | 72.18 ± 0.20     | 4.61 ± 0.07     | 3.70 ± 0.18     |
> | CODA-P           | 73.77 ± 0.50     | 79.38 ± 1.48     | 7.94 ± 0.08     | 6.72 ± 0.79     |
> | LGCL             | 68.65 ± 0.25     | 72.57 ± 0.19     | 4.75 ± 0.33     | 3.38 ± 0.58     |
> | HiDe-Prompt      | 75.75 ± 0.40     | 79.27 ± 0.17     | 2.29 ± 0.27     | 2.33 ± 0.17     |
> | PGP              | 68.62 ± 0.14     | 72.19 ± 0.20     | 4.53 ± 0.40     | 3.63 ± 0.35     |
> | EvoPrompt        | 76.00 ± 0.26     | 80.97 ± 0.30     | 4.22 ± 0.42     | 3.59 ± 0.52     |
> | CPrompt          | 76.32 ± 0.53     | 81.50 ± 0.30     | 6.10 ± 0.75     | 5.60 ± 1.35     |
> | ConvPrompt       | 77.08 ± 0.26     | 81.47 ± 0.10     | 4.17 ± 0.04     | 3.11 ± 0.17     |
> | **LEAPGen-lite** | **82.38 ± 1.04** | **85.14 ± 0.52** | **3.01 ± 1.19** | **2.13 ± 0.60** |
> | **LEAPGen**      | **84.09 ± 0.93** | **85.54 ± 0.65** | **1.46 ± 1.25** | **2.11 ± 1.21** |
> |                  | ImageNet-R       |                  |                 |                 |
> | L2P              | 57.40 ± 0.31     | 63.33 ± 0.21     | 10.76 ± 0.45    | 7.88 ± 0.17     |
> | DualPrompt       | 65.19 ± 0.17     | 70.31 ± 0.29     | 7.30 ± 0.18     | 5.16 ± 0.34     |
> | CODA-P           | 70.55 ± 0.71     | 77.08 ± 1.02     | 8.23 ± 0.86     | 6.95 ± 0.70     |
> | LGCL             | 64.96 ± 0.67     | 70.18 ± 0.37     | 7.35 ± 0.65     | 5.05 ± 0.32     |
> | HiDe-Prompt      | -                | 81.60 ± 0.48     | -               | 2.23 ± 0.38     |
> | PGP              | 65.24 ± 0.25     | 70.36 ± 0.26     | 7.17 ± 0.21     | 5.09 ± 0.25     |
> | EvoPrompt        | 74.93 ± 0.64     | 79.92 ± 0.13     | 6.72 ± 0.90     | 5.67 ± 0.26     |
> | CPrompt          | 74.23 ± 0.17     | 79.82 ± 0.51     | 5.98 ± 0.24     | 5.54 ± 0.48     |
> | ConvPrompt       | 73.93 ± 0.36     | 78.92 ± 0.37     | 4.87 ± 0.57     | 3.57 ± 0.25     |
> | **LEAPGen-lite** | **83.67 ± 0.39** | **85.65 ± 0.33** | **1.06 ± 0.24** | **0.47± 0.14**  |
> | **LEAPGen**      | **87.03 ± 0.12** | **87.81 ± 0.48** | **2.17 ± 0.17** | **2.54 ± 0.77** |
>
> **Please Note:  The analysis of LEAGen-lite empirically proves that our method doesn't rely on GPT/LLM or complex generator networks (high number of parameters)**

---

> ### Comment · Reviewer_J5sH · 2024-11-29
>
> Thank you for these detailed experiments.
> From the revisited paper, I now clearly understand that the increase in parameters is reasonable, and comparable with other research works. These results answer my questions, and I decide to increase the rate accordingly.

---

> > ### Author Response · Authors · 2024-11-29
> > **Response to Reviewer J5sH**
> >
> > Dear Reviewer J5sH,
> >
> > Thank you for the time and effort to review our revised manuscript, and increase the score for our paper. We surely appreciate it.
> >
> > Best Regards,
> >
> > Author of Submission 6684

---

### Official Review · Reviewer_ueBy · 2024-11-04

**Soundness:** 3
**Presentation:** 3
**Contribution:** 3
**Rating:** 8
**Confidence:** 4

**Summary:**

The paper proposes a new method for prompt based continual learning. Unlike previous methods that focus on task-specific prompts or growing prompt components, this work generates prompts using language descriptions of top-k similar classes from all the previously trained tasks to an input image. The generated prompts are plugged in to the Key and Value pairs of the backbone model for the given input image. In addition, the combined encoded language descriptions of the top-k similar classes are appended to the input image embeddings. Result on CIFAR100, ImageNet-R and CUB datasets with pretrained ViT-B/16 backbone shows significant improvements across accuracy and forgetting metrics over previous works. Extensive ablation study on different components provides better insights on the proposed approach.

**Strengths:**

•	Introduction is clear in terms of discussing drawbacks of prior works and motivating the proposed approach. Contributions are listed clearly.

•	I appreciate Section 3 and Figure 1 as it gives provides brief recap of the previous SOTA methods and laid some foundation for discussing the proposed method.

•	Leveraging language descriptions and using all language descriptors to find top-k similar classes to generate the prompts is novel in the context of continual learning.

•	The strategy to incorporate inter-class loss helps the method to achieve better results.

•	Comparisons are made against various previous works and significant improvements are shown with their proposed method.

•	Appreciate the detailed ablation study to understand benefits of different components.

**Weaknesses:**

I don’t find any major weakness but have set of questions (see below) to improve the understanding. However, I find that learning with auxiliary data and inter-task loss are critical components of the method (as shown in ablation study), which are not sufficiently emphasized in the contributions.

**Questions:**

1. Between L155-160, under the context of previous task-specific approaches, it is mentioned that two different tasks could produce similar key vectors. It would make a stronger point to justify this statement with some empirical analysis.

2. What is the value of k for top-k descriptors and how many Generators are chosen for each task? Can we chose a single generator for each task? How {E_i}^t of size |C^t|gets chosen for k set of generators?

3. From the Figure 2 or its caption, it is not clear what is the component that is compared against input x to get the cosine similarity. The sentence “search space for top-k” in the figure gives some hint, but the arrows for entire blue block rising the confusion. Please clarify it in the figure or caption.

4. Why it is interesting to solve equation 3, if the task is to search for E_i in all tasks [1, 2, ….t]? Similarly, why consider K^t and why not just use {L_i}^t ?

5. Under the ablation study, what does absence of generator mean? How do you get prompts in this approach without generator?

6. Is the MLP head (classifier) also trained for each task? How can it produce list of softmax values of all classes for equation 7?

7. Please explain why inter-task loss help improve FAA and CAA? What does it tell us about the learned parameters?

8. Has the importance of inter-task loss already explored in previous continual learning papers?

Minor suggestions:

a. In Figure 1, reposition the labels “fixed”, “trainable”, and “generated” since these labels affect entire Figure, not just the green block.

b. In section 3.b, I suggest to use emphasized text style for “pool-based approach”, “task-specific prompt approach”, and “growing component approach”.

c. In Line 242, correct the typo “tem” -> “them”

d. Line 267 mentions t_hat, but eq. 3 uses notation as t, but not t_hat.

------------------------------------------------------------------------
Final review: I thank authors for their responses. I find that paper proposes a novel method to leverage language descriptors within the context of continual learning. Authors have conducted thorough experimental evaluation and demonstrated notable improvements across multiple datasets. I read the fellow reviewers concerns and I find that authors have sufficiently addressed the major concerns and revised their manuscript accordingly. In particular, regarding the utilization of GPT, authors have justified that their method does not necessarily rely on compute heavy GPTs or LLMs, and can also work without GPT-generated descriptors. Overall, I believe that this paper would be interesting to the research community and suggest towards acceptance.

---

> ### Author Response · Authors · 2024-11-22
> **Autrhor Response to Reviewer ueBy for Q1-Q5**
>
> Q1. Between L155-160, under the context of previous task-specific approaches, it is mentioned that two different tasks could produce similar key vectors. It would make a stronger point to justify this statement with some empirical
>
> Our Response: Thank you for your question. We have added extra analysis on the similarity between task keys as presented in Appendix D1. Our analysis shows the high cosine similarity of task keys after completing all task (10 tasks) learning .
>
> Q2. What is the value of k for top-k descriptors and how many Generators are chosen for each task? Can we chose a single generator for each task? How {E_i}^t of size |C^t|gets chosen for k set of generators?
>
> Our Response: Thank you for your question.  The value of k is the same as the number of generators, and it is assigned as a user parameter. We choose k=3 for our experiment. Our sensitivity analysis (sec 5.2 point e.3) shows the performance of our method in various k. Yes, we can use a single generator as well. In our method, using a single generator means we choose k=1 which also means we choose only a single descriptor for prompt generation. How to select top-k $E_i$?
> Given a predicted task $t$, we choose top-k $E_i$ from the predicted (current task) task i.e. $E_c^t$, and from the previous task $E_c^1$, $E_c^2$,... $E_c^{t-1}$. The top-k selected descriptor embedding i.e. $E_1$, $E_2$... $E_k$ will be inputted into k generators of task $t$ i.e.  $G_1^t$, $G_2^t$... $G_k^t$. We explain the complete components and procedures in sec. 4 and Figure 2.
>
> Q3. From the Figure 2 or its caption, it is not clear what is the component that is compared against input x to get the cosine similarity. The sentence “search space for top-k” in the figure gives some hint, but the arrows for entire blue block rising the confusion. Please clarify it in the figure or caption.
>
> Our Response: Thank you for your question. We compare x to $L_c^i$ that is associated with $E_c^i, i \in [1,2,...t]$. We have adjusted the font in Figure 2 so that the string "search space for top-k" and symbol $L_c^i$ have the same size. We set both of them in italics for better cohesion between them.
>
> Q4. Why it is interesting to solve equation 3, if the task is to search for E_i in all tasks [1, 2, ….t]? Similarly, why consider K^t and why not just use {L_i}^t ?
>
> Our Response: Thank you for your question.
>
> - We search for top k $E_i$ as the input for prompt generators. Along with the $E_i$ from the expected class, we look for other k-1 descriptors that are closest to the expected class which may be located in previous tasks, not in the same task. For example, descriptors of class "pear" are closer to the descriptors of class "apple" than to the descriptors of class "palm tree". Class "apple" is located in the previous task, while class "palm tree" is in the same task as class "pear". Thus, it is better to select other k-1 descriptors from previous tasks. Why not search from all tasks? In that case, our method will always use the same (latest) generators that might be incompatible (un-optimal) for all tasks. Besides, It will increase the chance of wrong-selected descriptors.
>
> - The use of ${L_i}^t$ only has the risk of misprediction due to similar keys from different classes from different tasks. For example, images of class 'Scottish terrier' have high similarity with images of class 'West Highland white terrier' from a different task. The use of $K^t$ only has the risk of similar key vectors from different tasks as we mentioned earlier. Thus, task prediction using  ${L_i}^t$ and $K^t$ together has less misprediction risk than using ${L_i}^t$  and $K^t$ only.
>
>
> Q5. Under the ablation study, what does absence of generator mean? How do you get prompts in this approach without generator?
>
> Our Response: Thank you for your question.  The absence of a generator means the prompt components are produced directly from selected (the top-k) descriptors i.e. $PC_i = E_i$, and the prompt is produced as $P = \Sigma_i^k PC_i = \Sigma_i^k PC_i = \Sigma_i^k  E_i$.

---

> ### Author Response · Authors · 2024-11-22
> **Autrhor Response to Reviewer ueBy for Q6-Sd**
>
> Q6. Is the MLP head (classifier) also trained for each task? How can it produce list of softmax values of all classes for equation 7?
>
> Our Response: Thank you for your question. Yes, the MLP head is trained in every task. Please note that following standard practice in continual learning, the MLP head gradually adds output neurons when entering a new task. For example, in the case of class incremental learning with 10 classes per task. In the first task, the MLP head has 10 outputs, when entering task 2, the output neurons are added by 10 (now 20 outputs), and so on. Therefore It can produce logits for all and ongoing learned classes. In equation 7, we compare all the logits to the labels, while in equation 6 (intra-task loss) we only consider the logists from currently learned classes, excluding previously learned classes.
>
>
> Q7. Please explain why inter-task loss help improve FAA and CAA? What does it tell us about the learned
>
> Our Response: Thank you for your question.  Inter-task loss improves the discriminative ability between currently learned classes and previously learned classes (from previous tasks). The newly added parameters/weights are driven to produce different representations between the currently learned classes and previously learned classes. Thus, the inter-task loss contributes to improving CAA and FAA. However, at the same time, inter-task loss increases the risk of forgetting since it changes (adjusts) model's parameters that optimal for previous learned classes.
>
> Q8. Has the importance of inter-task loss already explored in previous continual learning papers?
>
> Our Response: Thank you for your question.  Our literature study shows that inter-task loss is the default cross-entropy loss that is widely utilized for continual learning. Following HidePrompt, we emphasize the importance of intra-task and inter-task loss to improve within-task prediction and inter-task prediction. However, our method and HiDe-Prompt have different mechanisms for intra-task loss. The new joint loss function including intra-task loss, inter-task loss, cosine similarity to $K^t$ ($\mathcal{L}_t$), and cosine similarity to $L_c^t$ ($\mathcal{L}_c$) as presented in eq.10 is unique and it is one of our contributions.
>
> Sa. In Figure 1, reposition the labels “fixed”, “trainable”, and “generated” since these labels affect entire Figure, not just the green block.
>
> Our Response: Thank you for the suggestion. We have revised Figure 1, we moved the labels outside the green block.
>
> Sb. In section 3.b, I suggest to use emphasized text style for “pool-based approach”, “task-specific prompt approach”, and “growing component approach”.
>
> Our Response: Thank you for the suggestion. We have bolded the mentioned texts in our revised paper.
>
> Sc. In Line 242, correct the typo “tem” -> “them”
>
> Our Response: Thank you for the correction. We have corrected the typo in our revised manuscript.
>
> Sd. Line 267 mentions t_hat, but eq. 3 uses notation as t, but not t_hat.
>
> Our Response: Thank you for the correction. We have revised the symbol into $t$ (without hat).

---

### Author Response · Authors · 2024-11-22
**Response to All Reviewers and Summary of Updates**

We would like to express our gratitude to all the reviewers for their constructive and insightful comments on our paper. We are honored that all the reviewers (ueBy, J5sH, UGi2, and bACr) found our work to be novel/new. We also appreciate reviewers (ueBy, and bACr) for pointing out the detailed writing of our paper as a positive point. We have revised our paper, following the reviewers' feedback. We have improved the quality and presentation of our paper with the summary of changes as follows:

(1). We added a detailed overview of our method as presented in section 4.1. to improve e better presentation and understanding of our proposed method. We emphasized the uniqueness of our method to existing SOTAs in the section.

(2). We replace our ablation study with a step-by-step scenario to improve the clarity of each component contribution, as presented in section 5.2.d.

(3). We added a new empirical experiment and analysis on the high similarity of task key (K^t) as presented in Appendix D.1. The high similarity of task keys motivates us to develop a more accurate task-id prediction mechanism rather than the conventional way.

(4).  We added a new empirical experiment and analysis on the high similarity of language embedding generated by the encoded string "the photo of class name" as used in the existing method. The high similarity between classes motivates us to develop a more discriminative way of language-guided learning rather than utilizing the embedding directly as references or loss anchors. Please see Appendix D2.

(5). We added a new experiment and analysis on the trade-offs between cost and performance that comprehensively evaluate the proposed methods and existing SOTAs in terms of performance (accuracy and forgetting), number of parameters, and running time. We also compare our proposed method with the large version of SOTAs that executes a higher number of parameters. Please see Appendix D3.

(6). We added a new experiment and analysis on the "class name" as descriptors where we have no class descriptors generated by GPT or other LLMs. Our experiment results as presented in Appendix D4. shows that our method outperforms the existing SOTAs with a significant margin in 3 datasets despite substituting class descriptors with the class name.

(7). We elaborated the detailed setting for all consolidated methods as presented in Appendix E.

(8). We revised Figures 1 and 2 for a better understanding of our paper.

(9). We improve the readability and presentation of our paper by ensuring a clear space between a text to another text, caption, figure, or table.

(10), Last but not least, We revise the writing errors and typos.

The changes w.r.t. feedback from reviewers 1(ueBy), 2(J5sH), 3(UGij), and 4 (bACr) are highlighted in red, green, orange, and violet colors respectively.

---

### Author Response · Authors · 2024-11-27
**Additional Updates**

Dear Reviewers and Area Chairs,

We added additional updates in our latest manuscript that fully address the comments/concerns of reviewers  J5sH, UGi2, and bACr as follows.

1. We added experiments and analysis of the lite version of LEAPGen i.e. LEAPGen-lite that utilizes lightweight Conv1d generators and without class descriptors (it uses class names only). LEAPGen lite has far smaller parameters than ConvPrompt (12.7% of ConvPrompt #params) and is similar to Hide-Prompt parameters. Our experiment shows that LEAPGen-lite outperforms the existing SOTAs i.e. 4.7-25% FAA and 3-11% CAA in CIFAR100, 3-30% FAA and 2-26% CAA in ImageNet-R, and 6-20% FAA and 3.3-14% CAA in CUB. This evidence proves that the significant performance of our ideas (language as input for prompt generation, task-wise generators, soft task-id predictor, and learning with auxiliary) is not bounded by LLM-generated descriptors or the complexity (parameters and size) of the generators. Please kindly see the results and analysis in Tables 2,3,4, Figure 3, and Section 5.2.d.

2. We added a summarized analysis on #Parameters, Running Time, and Storage as presented in section 5.2.g, detailed in Appendix D.3. The analysis shows that LEAPGen spends moderate inference time, but lower training+inference time and total running time, than ConvPrompt and Hide-Prompt, despite having more parameters and storage. LEAPGen-lite requires far smaller cost in all aspects than ConvPrompt and smaller running time and storage than HiDe-Prompt, with just a slightly higher number of parameters i.e. 0.16M vs 0.15M. We move the analysis of LEAPGen performance on various types of generators to the Appendix for the exchanged space in the main paper.

Both additional updates are presented in brown color in our latest manuscript.

Thank you.

Best Regards

Author of Submission 6684

---

### Author Response · Authors · 2024-12-04
**Summary of Revision and Discussion**

Dear Program Chairs (PC), Senior Area Chairs (SAC), Area Chairs (AC), and Reviewers,


As the discussion phase has ended,

**First**, we thank reviewers PC, SAC, AC, and Reviewers for your effort and time in organizing the review and discussion of our paper. Through these processes, we have improved the technical quality and presentation of our paper significantly.  We thank the reviewers ueBy and J5sH for reviewing our paper and/or its revision thoroughly, confirming the novelty and significance of our works.


**Second**, we have revised our paper following the reviewers' advice in both writing i.e. **layout, presentation, details, tables, figures, etc**, and technical aspects i.e. **method overview, ablation study, the trade-off between performance and cost, empirical/numerical evidence of task similarity, language embedding similarity,  setting details, etc** as we mentioned in our previous official comments.


**Third**, we would like to confirm that we have addressed all concerns and questions of the reviewers in our revised paper and responses. However, we would like to emphasize a few key points that reviewers UGi2 and/or bACr may not have reviewed thoroughly in the discussion phase, as follows:


1. **Fairness Setting** :   LEAPGen utilizes the same resources as ConvPrompt and LEAPGen-lite(no descriptors with lightweight generators) utilizes less resources than ConvPrompt and comparable resources to the other SOTAs (sec. 5.1, sec. 5.2.g, Appendix E). All methods are evaluated in the same setting and run with respective best (recommended by official paper/code) hyperparameters. Thus we believe we have satisfied the fairness aspect in our study.


2. **The Non-Mandatory Presence of GPT**  :

- **GPT is not part of our method** (sec 4.2.a, and fig. 2), It is one of the options to generate class descriptors following ConvPrompt. Our method accommodates both class descriptors (as in ConvPrompt) and class names (as in LGCL) language text. Even without GPT-generated descriptors. our method works excellently and significantly outperforms the existing SOTAs, as demonstrated by LEAPGen-lite (sec. 5.2.d) and LEAPGen-CN (Appendix D.4).

- **GPT is utilized in an online way** (through the Internet), by querying it using API and Python script. It is a common practice nowadays as in our simulation (please see https://platform.openai.com/docs/guides/). Thus we never download and save the GPT model in our storage. LEAPGen indeed saves the descriptors embedding (numerical vector), but not the GPT model.  The embedding adds to the whole model's storage size but is still in a reasonable amount, and we have included it in our storage analysis.

- **Descriptor generation by GPT** (before training) via only query indeed takes a fair amount of time. However, despite spending extra time on it, our method still has a **lower total running time** than existing SOTAs, i.e. ConvPrompt and HidePrompt. (sec. 5.2.g, Appendix D.3)

- **Descriptor impact** to the model performance has been demonstrated by the performance difference between LEAPGen and LEAPGen-CN (Appendix D.4). We also extend our analysis of the performance of LEAPGen vs ConvPrompt in various types of descriptors (Appendix D.5).


3. **LEAPGen-lite** proves the significant impacts of our ideas i.e. (1) language as input for prompt generation, (2) task-wise generators, (3) soft task-id predictor, and (4)learning with auxiliary despite without GPT-generated descriptors (only utilizes class names), and utilize least parameters, running time and storage.  (sec. 5.2.d, sec. 5.2.g, Appendix D.3)


We believe these key points **resolve all concerns** regarding the **fairness setting** and **GPT (non-mandatory) presence**, and prove the **significant impacts** of our ideas i.e.  (1) language as input for prompt generation, (2) task-wise generators, (3) soft task-id predictor, and (4) learning with auxiliary, without being bound to the presence of GPT.


I think that's all, once again thank you to all the committee, we pray that the hard work and dedication of the committee will be paid off by the success and impact of this year's ICLR, Thank you.


Best Regards,

Author of Submission 6684

---

### Meta-Review · Area_Chair_6jbB · 2024-12-20

**Metareview:**

(a) The paper proposes LEAPGen, a novel method for continual learning that generates prompts using language descriptors for classes instead of shared embeddings, with a learnable task key and class-level parameters. LEAPGen predicts the task ID and selects top-k matching descriptor embeddings to generate prompts. Experimental results on CIFAR100, ImageNet-R, and CUB show that LEAPGen outperforms state-of-the-art methods in accuracy and mitigating forgetting.

(b) Strengths: The strengths of the paper lie in its clear and well-motivated introduction, which effectively highlights limitations of prior works and outlines the proposed approach. LEAPGen introduces a novel use of language descriptors for prompt generation, leveraging richer semantic information and avoiding the need to store or replay old data, thereby reducing catastrophic forgetting. The method achieves significant improvements over state-of-the-art methods and incorporates a unique inter-class loss strategy to enhance performance. The experimental results are robust, with thorough comparisons against existing methods and detailed ablation studies to validate the contributions of individual components. Section 3 and Figure 1 provide a concise recap of prior methods, laying a solid foundation for the proposed approach.

(c) Weaknesses: The paper has several weaknesses, including a lack of clarity about key details, such as the specific Sentence Transformer used, whether it is pre-trained, and whether any fine-tuning occurs during the continual learning process. The scalability of the proposed method is a concern, as adding new generators for new tasks could significantly increase the number of parameters, leading to potential unfair comparisons with related works. The writing and structure require improvement, as the compact format affects readability, and some components, like the theorem, seem irrelevant and better suited for supplementary materials. The ablation study lacks rigor and should more systematically validate the contributions of each module starting from a standard baseline. Additionally, the use of ChatGPT for generating descriptions raises concerns about the fairness of comparisons and the source of performance gains, prompting the need for transparency and public availability of these descriptions.

(d) The most important reasons for acceptance are the solid methods and compelling performance, along with the sufficient theoretical & empirical analyses. Since this paper has received 2 negative scores (i.e., 3), the AC has checked very carefully about the details and discovered that the major reasons for low scores, i.e., the inclusion of GPT for accurate description, are well addressed by the authors. The authors have designed a lite version (LEAPGen-lite) of the proposed approach which does not rely on GPT entirely. The LEAPGen-lite still shows significantly superior performance to previous sota methods, proving the significant impacts of the ideas. In the AC's view, all the issues have been addressed during the rebuttal period. By the way, Reviewer UGi2 who scores 3 fails to participate in the rebuttal period despite reminders from the AC.

One more suggestion for the authors to improve the presentation quality: In the current form of this manuscript, the authors show 4 principles. Four principles are too many; for a highly impactful paper, one principle is sufficient—for example, the residual idea in ResNet. The AC suggests the authors reconsider the core problem this paper aims to address, propose a single principle, and then present multiple methods to support this principle. This approach would enhance the clarity of the motivation and increase the paper's overall impact.

**Additional Comments On Reviewer Discussion:**

(a) Reviewer ueBy notes no major weaknesses but raises questions to improve understanding. They highlight that the auxiliary data and inter-task loss, critical components demonstrated in the ablation study, are underemphasized in the stated contributions. Emphasizing these aspects could strengthen the presentation of the method. The authors have successfully addressed the issues.

(b) Reviewer J5sH highlights that the paper lacks clarity on key details about the pre-trained models used, such as the specific Sentence Transformer and whether fine-tuning occurs during continual learning. They also point out the absence of a clear comparison of parameter increases due to adding generators for new tasks, noting that the generators are not lightweight. This could result in significant parameter growth and potentially unfair comparisons with related works. Given the rebuttal, the reviewer clearly understands that the increase in parameters is reasonable, and comparable with other research works. He decides to increase the score to 6.

(c) Reviewer UGi2 notes that the paper's writing and overly compact structure hinder readability, suggesting an initial overview and moving the weakly relevant theorem to the supplementary materials. They find the ablation study insufficiently convincing and recommend validating modules step-by-step from a standard baseline. Concerns are raised about the fairness of experimental comparisons involving ChatGPT and the extent of performance gains attributed to accurate descriptions, with a suggestion to make ChatGPT-generated descriptions publicly available to enhance the paper's impact. Unfortunately, the reviewer UGi2 fails to participate in the rebuttal period despite reminders from the AC. The AC has checked the rebuttal and discovered that the authors have addressed the concerns accordingly.

(d) Reviewer bACr finds the comparison of prompt similarities (lines 183-184) flawed, as evaluating similarity on a letter-by-letter basis is inappropriate and fails to reflect semantic differences. They highlight concerns about the reliance on ChatGPT and a Sentence Transformer, which introduce additional resources and create unfair comparisons with other methods, limiting real-world applicability. Although the method achieves state-of-the-art performance, gains are minor on fine-grained datasets like CUB200, and the paper's layout requires improvement for better readability. After the initial rebuttal, the reviewer still has the concerns regarding the resources of pre-computations from GPT. However, the AC notes that the LEAPGen-lite proposed by the authors has already removed the GPT-generation, demonstrating that the concern regarding GPT is not an issue.

---

### Decision · Program_Chairs · 2025-01-22

Accept (Poster)